



# Simultaneously Determining Global Sensitivities of Model Parameters and Model Structure

Juliane Mai[1], James R. Craig[1], and Bryan A. Tolson[1]

[1]Department Civil and Environmental Engineering, University of Waterloo, 200 University Ave W, Waterloo, ON, N2L 3G1, Canada.

**Correspondence:** Juliane Mai (juliane.mai@uwaterloo.ca)

**Abstract.** Model structure uncertainty is known to be one of the three main sources of hydrologic model uncertainty along with input and parameter uncertainty. Some recent hydrological modeling frameworks address model structure uncertainty by supporting multiple options for representing hydrological processes. It is, however, still unclear how best to analyze structural sensitivity using these frameworks. In this work, we apply an Extended Sobol' Sensitivity Analysis (xSSA) method that oper-
ates on grouped parameters rather than individual parameters. The method can estimate not only traditional model parameter sensitivities but is also able to provide measures of the sensitivities of process options (e.g., linear vs. non-linear storage) and sensitivities of model processes (e.g., infiltration vs. baseflow) with respect to a model output. Key to the xSSA method's applicability to process option and process sensitivity is the novel introduction of process option weights in the Raven hydrological modeling framework. The method is applied to both artificial benchmark models and a watershed model built with the Raven
framework. The results show that: 1) The xSSA method provides sensitivity estimates consistent with those derived analytically for individual as well as grouped parameters linked to model structure. 2) The xSSA method with process weighting is computationally less expensive than the alternative aggregate sensitivity analysis approach performed for the exhaustive set of structural model configurations, with savings of 81.9% for the benchmark model and 98.6% for the watershed case study. 3) The xSSA method applied to the hydrologic case study analyzing simulated streamflow showed that model parameters adjust-
ing forcing functions were responsible for 42.1% of the overall model variability while surface processes cause 38.5% of the overall model variability in a mountainous catchment; such information may readily inform model calibration. 4) The analysis of time dependent process sensitivities regarding simulated streamflow is a helpful tool to understand model internal dynamics over the course of the year.

## 1 Introduction

Hydrologic processes such as infiltration of water into soil or water interception by the canopy of trees are often too complex to parameterize or insufficiently understood at scales of interest to be represented in every detail in computer models. The consequence of this is that simplified conceptual or empirical models are often used to represent these physical processes; such models are typically computationally expedient and possess a relatively smaller number of parameters than continuum models based upon the Freeze and Harlan (1969) blueprint. They are also non-unique: a large number of such process algorithms can



be found in the hydrological modeling literature; non-unique model representations are similarly ubiquitous in (e.g.) ecological

models or socioeconomic systems models. The availability of different conceptualization schemes leads to a wide variety of

algorithmic options to describe phenomena within a model. The subjective decision of which processes representation should

be used in a model is complemented by other subjective decisions such as how a modeled system is discretized, how processes

may be simplified, or what time step is appropriate. The uncertainty introduced by these decisions is usually referred to as

structural model uncertainty.

Model structural uncertainty is commonly recognized (Gupta et al., 2012) as one of the three key components of hydrological

model uncertainty, along with parameter uncertainty and input (forcing) uncertainty. However, while the literature on sensitivity

analysis for model parameters is rich (Morris, 1991; Sobol', 1993; Demaria et al., 2007; Foglia et al., 2009; Campolongo et al.,

2011; Rakovec et al., 2014; Pianosi and Wagener, 2015; Cuntz et al., 2015, 2016; Razavi and Gupta, 2016a, b; Borgonovo

et al., 2017; Haghnegahdar et al., 2017), and there has likewise been a good deal of research into the influence of model input

uncertainty (Baroni and Tarantola, 2014; Abily et al., 2016; Schürz et al., 2019), sensitivity to model structural uncertainty has

received far less attention. With model structure we refer to various process conceptualizations within a model rather than, for

example, model discretization. One major reason for difficulties in addressing model structural uncertainty is due, in part, to the

historical inflexibility of environmental and hydrological models which readily allow the user to perturb parameters or input

forcings via input files, but are often constrained to a hard-coded model structure or a generally fixed model structure with a

relatively small number of options. However, the recent advent of flexible hydrological modeling frameworks such as FUSE

(Clark et al., 2008), SUPERFLEX (Fenicia et al., 2011), SUMMA (Clark et al., 2015), or Raven (Craig et al., 2020) enables

manipulation of model structure in addition to parameters and inputs. They afford a sufficient number of degrees of freedom in

model structure to start to explore model sensitivity to structural choices, and the interplay between model structures.

To date, there have been limited attempts to simultaneously estimate model parameter, input, and structural sensitivities.

Baroni and Tarantola (2014) introduced a sensitivity method based on a finite set of individual models. We will refer from now

on to this method as the *Baroni method*. While the approach may be generally applicable to arbitrary structural differences, in

their testing, they varied only in how the model was internally discretized (i.e., in the number of soil layers). The number of

and sensitivity to parameters did not change when moving between model structures, and the method only derives sensitivity

of hyper-parameters such as, for example, the group of soil parameters making it difficult to pinpoint the key parameters the

model predictions are most sensitive to. A major limitation of this method is that individual parameters can only be associated

to one form of uncertainty and hence the method limits the groups that can be defined. The Baroni method will be contrasted

to the method developed here to examine this limitation in more detail. The method introduced

Günther et al. (2019) applied the Baroni method to determine the sensitivity of a multi-physics snowpack model regard-

ing model parameters, forcing data, and 32 distinct model structures, but individual model parameter sensitivities were not

determined. Schürz et al. (2019) proposed a comprehensive sensitivity analysis regarding alternative model inputs, climate

scenarios, and model setups, where the model setups varied in the number of sub-basins and hydrologic response units (but not

process representation). The analysis, based on pre-sampled behavioral parameter sets and 7000 model combinations, could as-

sess the relative impact of the different sources of uncertainty, but could not be used to examine the linkages between different





forms of uncertainty. Similar to the Baroni method, parameters were treated in an aggregate fashion which made it impossible to attribute the parameter sensitivity to a certain parameter or model component.

Van Hoey et al. (2014) is one of the few studies that explicitly examined the sensitivity of a model to changes in process representation, estimating sensitivities of parameters of various model structures with two or three alternatives per process, e.g., linear vs. non-linear storage; with or without an interflow process. A computationally expensive sensitivity analysis was
performed for each individual model analyzing the results by pairwise visual comparison of the alternatives, leading to $N$ sensitivity estimates for each parameter conditional on which of the $N$ model structures it is based on. It remains unclear on how to aggregate these $N$ estimates to derive a global overall sensitivity for all parameters.

Francke et al. (2018) proposed a similar method that was only capable of distinguishing between binary model components (i.e., a model feature/ enhancement is either present or not). Although Pfannerstill et al. (2015) did not explicitly focus on
modeling frameworks, they studied the sensitivity of parameters regarding individual model processes verifying if a process output is behavioral.

Dai et al. (2017) proposed a so-called process sensitivity metric which is based on Sobol' sensitivities. They enable the derivation of a countable set of process options for each of the model processes and derive an overall sensitivity through model averaging.

In all cases above, the resultant sensitivity metrics may be useful for (e.g.) differentiating between the magnitude of model sensitivity to structure versus that of parameters. However, these methods cannot be used to provide insight into the sensitivity of individual model structural choices, nor can they be used to disentangle the complex relations between model parameter and structure sensitivities or the interplay between interacting model structures. It is therefore difficult to use such methods to identify preferred model structures to inform the process of model calibration. In addition, none of the methods that derive
sensitivities across multiple model structures recognize the fact that model parameters may be present or absent conditional upon the model structure. Lastly, the above mentioned methods are generally computationally expensive, are only available for a small number of process parameterization options, and only determine the sensitivity of the parameterization in general without providing insight into what is causing this sensitivity by analyzing, for example, the sensitivity of individual parameterization options or individual parameters. While potentially useful for some applications, the available approaches have either
not been applied of do not allow for such an in-depth analysis of model structure and hence might provide only limited support for improvements in hydrologic modeling.

Here, we propose a new technique, the Extended Sobol' Sensitivity Analysis (xSSA) method, and apply it to models whose structure can vary continuously between discrete process options for estimating hydrologic fluxes. The proposed method is based on the existing concept of grouping parameters when applying the Sobol' method (Sobol and Kucherenko, 2004; Saltelli
et al., 2008; Gilquin et al., 2015). This concept has been applied by Dai et al. (2017) to derive the process sensitivities of processes with several alternative options by using model averaging of the discrete set of options. To our knowledge, the method of grouping parameters to derive sensitivities of parameters, process options, and processes without the explicit necessity of model averaging has not yet been applied. The xSSA method is made uniquely possible due to a special property of the Raven hydrologic modeling framework (Craig et al., 2020), whereby hydrologic fluxes (e.g., infiltration, runoff, or baseflow) may be



calculated via the weighted average of simulated fluxes generated by individual process algorithms; other flexible models may be revised to accommodate such analysis. As will be demonstrated below, the xSSA is uniquely capable of simultaneously providing global sensitivities of parameters, process algorithms (e.g., the Green and Ampt (1911) infiltration method), and hydrologic processes (e.g., infiltration).

The xSSA method allows us to efficiently estimate not only the global sensitivity of model parameters independent of the
chosen model structure, but also to evaluate the sensitivity of alternative model process options (e.g., that of different snowmelt algorithms), and the sensitivity of hydrological process components (e.g., snowmelt vs. infiltration). We here pose these as four distinct sensitivity metrics:

A.  Conditional parameter sensitivity: *Which model parameter is most influential given a certain model structure?*
    For example, which model parameter is most important in the HBV model? (This is the traditional Sobol' metric)

B.  Unconditional parameter sensitivity: *Which model parameter is most influential independent of model option choice?*
    For example, which model parameter is overall the most influential given all possible model structures (available in the modeling framework)?

C.  Process option sensitivity: *Which of the available options for a process in a modeling framework is the most sensitive?*
    For example, which choice of the infiltration process description has the largest impact on the simulated streamflow in
my catchment of interest?

D.  Process sensitivity: *Which model process or component is most influential upon model results?*
    For example, is infiltration more important than the handling of snow melt? Or, is the simulated streamflow more sensitive to infiltration or evaporation?

Below, we define these metrics explicitly and introduce the xSSA methodology for calculating them. The xSSA method is
tested using two artificial benchmark model to check for consistency between analytical and numerically derived sensitivity index estimates. The proposed method is also compared to the existing Baroni method revealing limitations that can be resolved using the xSSA method. The xSSA method is then applied to a hydrologic modeling case study using the Raven hydrologic modeling framework, demonstrating the insights that may be gained through the simultaneous in-depth analysis of model parameters and model structure to improve hydrologic modeling practices. The method is demonstrated to be more efficient
than a conventional approach whereby the standard Sobol' method is repeatedly applied to distinct model structures as in the study by Van Hoey et al. (2014), in addition to providing more useful information regarding model sensitivities.

## 2  Materials & Methods

The section will first introduce the models and their setups (Sec. 2.1) used to test and validate the proposed Extended Sobol' Sensitivity Analysis (xSSA) method as here applied to determine model structure and parameter sensitivities. We will briefly
revisit the traditional method of Sobol' that is so far primarily used to obtain model parameter sensitivities (sensitivity metric





A) and introduce the extensions which support sensitivity estimates for model process options (sensitivity metric C) and model processes (sensitivity metric D; all in Sec. 2.2). Finally, we present the experiments used to test the proposed method and address the research questions raised in the introduction (Sec. 2.3).

## 2.1   Models and Setup

This section will briefly introduce the three test cases used to demonstrate the functioning of the xSSA. The first two test cases are artificial benchmark models where the sensitivity index values can be derived analytically (Sec. 2.1.1). We use two benchmark models to demonstrate limitations of available methods and to show that the proposed xSSA method is converging to all analytical values. The third model is a real world example using a hydrologic model that allows for flexible model structures, i.e. the hydrologic modeling framework Raven (Sec.2.1.2). The watershed being modeled is described in the last

section (Sec. 2.1.3).

### 2.1.1   Artificial Benchmark Model Setups

The artificial benchmark models are employed to demonstrate that the proposed method is capable of deriving the sensitivities of not only individual model parameters but also of grouped parameters linked to individual process options (e.g., the linear baseflow algorithm) or processes regardless of available options (e.g., baseflow). The benchmark models are further used to

demonstrate limitations of existing methods that were previously used to analyze model structure sensitivities. We use two hypothetical models here: One model where each model parameter is only used in distinct processes and process options (disjoint-parameter benchmark; Fig. 1A) and a more advanced benchmark model where parameters are shared between several processes and process options (shared-parameter benchmark; Fig. 1B). The latter is assumed to be more realistic as model parameters such as, for example, the thickness of the upper soil layer can appear in multiple processes (e.g., evaporation,

quickflow, infiltration, and percolation). The two benchmark models are both assumed to consist of three processes: $A$, $B$, and $C$ as well as $D$, $E$, and $F$. The model output $f(\boldsymbol{x})$, a function of model parameters $\boldsymbol{x}$, is defined by

$$f_{\text{disjoint}}(\boldsymbol{x}) \quad = \quad A \cdot B + C \tag{1}$$

$$f_{\text{shared}}(\boldsymbol{x}) \quad = \quad D \cdot E + F \ . \tag{2}$$

The product of processes $A$ ($D$) and $B$ ($E$) is intended to mimic non-linear coupling of model processes while the addition of

$C$ ($F$) is intended to resemble linear process coupling. Each of the three processes is assumed to allow for multiple process options. For the disjoint-parameter benchmark model, the process $A$ is set to have 2 options ($A_1$, $A_2$), $B$ has 3 options ($B_1$,





$B_2$, $B_3$), and $C$ has two options ($C_1$, $C_2$):

$$A_1 = \sin(x_1) \tag{3}$$

$$A_2 = 1 \tag{4}$$

$$B_1 = 1 + bx_2^4 \tag{5}$$

$$B_2 = 1 + bx_3^2 \tag{6}$$

$$B_3 = x_4 + bx_5 \tag{7}$$

$$C_1 = a\sin^2(x_6) \tag{8}$$

$$C_2 = 1 + bx_7^4 \tag{9}$$

For the shared-parameter benchmark model, the process $D$ is set to have 2 options ($D_1$, $D_2$), $E$ has 3 options ($E_1$, $E_2$, $E_3$), and $F$ has two options ($F_1$, $F_2$):

$$D_1 = \sin(x_1) \tag{10}$$

$$D_2 = x_1 + x_2^2 \tag{11}$$

$$E_1 = 1 + bx_2^4 \tag{12}$$

$$E_2 = 1 + bx_3^2 \tag{13}$$

$$E_3 = x_4 + bx_5 \tag{14}$$

$$F_1 = a\sin^2(x_6) \tag{15}$$

$$F_2 = 1 + bx_7^4 + x_3^2 \tag{16}$$

This allows for $2 \times 3 \times 2 = 12$ individual models for the disjoint-parameter benchmark and 12 individual models for the shared-
parameter setup using seven model parameters $x_1$ to $x_7$ that are all sampled uniformly from the range $[-\pi, \pi]$. By design, not all model parameters are used in each of the 12 models. The number of "active" parameters for the shared-parameter setup ranges from 3 (e.g., $D_1 \cdot E_1 + F_1$) to 6 (e.g., $D_2 \cdot E_3 + F_2$). For the disjoint-parameter setup each parameter appears in exactly one process option. In the shared-parameter setup, parameter $x_1$ is used in two process options of the same process ($D_1$ and $D_2$). Parameter $x_2$ and $x_3$ are used in multiple process options of different processes. $x_2$ is present in process options $D_2$ and
$E_1$ to evaluate the behavior of sensitivities for multiplicative parameters and $x_3$ is present in process options $E_2$ and $F_2$ to check for additive behavior. A schematic of the model options and associated model parameters are shown in Figure 1A and B.

The model that is built using the first process options

$$
\begin{aligned}
f_{\text{disjoint}}(\boldsymbol{x}) &= A_1 \cdot B_1 + C_1 \\
&= \sin(x_1) \cdot (1 + bx_2^4) + a\sin^2(x_6) \\
&= D_1 \cdot E_1 + F_1 = f_{\text{shared}}(\boldsymbol{x})
\end{aligned}
\tag{17}
$$





resembles the Ishigami-Homma function (Ishigami and Homma, 1990) which is a common benchmark function in sensitivity analysis studies (Homma and Saltelli, 1996; Cuntz et al., 2015; Stanfill et al., 2015; Pianosi and Wagener, 2015, 2018; Mai and Tolson, 2019). The Ishigami-Homma parameters $a$ and $b$ are fixed at 2.0 and 1.0, respectively.

The Sobol' sensitivity indexes of all 12 model configurations can be derived analytically following closely the description in Saltelli et al. (2008) (p. 179–182). They are listed for the shared-parameter benchmark model in Table B1 of the Appendix B.

A reasonable approach for evaluating the sensitivity of 12 individual models involves choosing exactly one process option for each process in Eq. 2, e.g. $D = D_1$, $E = E_2$ and so on. This can be generalized by choosing a weighted sum of all available process options to represent a process, e.g. $D = w_1 D_1 + w_2 D_2 + \ldots$. The sum of weights $w_i$ per process is assumed to be 1. In case of the shared-parameter benchmark example, Eq. 2 is therefore changing to

$$f_{\text{shared}}(\boldsymbol{x}, \boldsymbol{w}) \quad = \quad (w_{d1} D_1 + w_{d2} D_2) \cdot (w_{e1} E_1 + w_{e2} E_2 + w_{e3} E_3) + (w_{f1} F_1 + w_{f2} F_2) \tag{18}$$

where

$$
\begin{aligned}
w_{d1} + w_{d2} &= 1 \\
w_{e1} + w_{e2} + w_{e3} &= 1 \\
w_{f1} + w_{f2} &= 1 \;.
\end{aligned}
$$

The 12 individual models can be obtained when the weights are set accordingly, e.g. the Ishigami-Homma function can be obtained by setting $w_{d1} = w_{e1} = w_{f1} = 1$. However, given that weights can take on non-integer values, we now have an infinite number of model structures rather than 12. The same can be constructed for the disjoint-parameter model setup.

For sampling the continuum of all process options, the weights need to be independently and identically distributed (iid). Therefore, random numbers $r_i$ are sampled from the uniform distribution, $\mathcal{U}[0,1]$, and transformed into the weights following
the approach described by Moeini et al. (2011). $N-1$ such random numbers are required for $N$ weights of competing options. The recipe on how to transform the uniformly sampled numbers $r_i$ into weights is specified in Eq. A1 of Appendix A. For the benchmark example, one requires 4 such uniform random numbers ($r_1, \ldots, r_4$) to derive the 7 weights ($w_1, \ldots, w_7$).

The approach of weighted model options hence comes at the expense of introducing additional parameters $r_i$ to derive the weights. Larger numbers of model parameters always results into an increased number of model runs needed for the sensitivity
analysis. However, using the model with weighted options, one now has to run and analyze only one generalized model structure instead of 12 fixed structures. Therefore, this approach reduces the number of required model runs provided that the model allows to derive outputs of weighted process options directly. This feature is available in the hydrologic modeling framework Raven and was the primary reason for the choice of this flexible modeling framework over others. The sensitivities of the additional model parameters (i.e., weights) can further hold interesting insights into the model structure (see Section 3).
The analytically-derived Sobol' indexes for the remaining three sensitivity metrics (B-D) can be derived using the revised model description (Eq. 18). The indexes for the shared-parameter model setup can be found in Eq.s B1 to B3 in Appendix B.





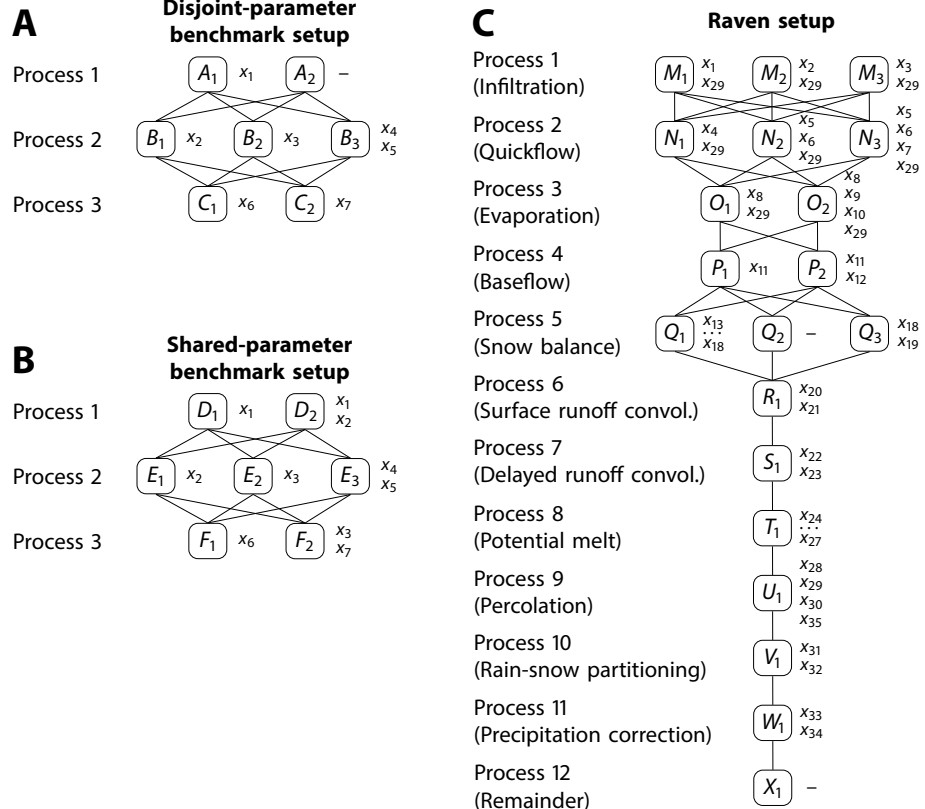

**Figure 1.** The three model setups used in this study. The first two serves as a artificial benchmark model since sensitivities of parameters, process options and processes can be derived analytically. The benchmark examples consists of three processes: $A$, $B$, and $C$ ($D$, $E$, and $F$). The three processes are connected through $A \cdot B + C$ ($C \cdot D + E$) to obtain the hypothetical model outputs. Processes $A$ ($D$) and $C$ ($E$) have two options, process $B$ ($E$) has three. The parameters $x_i$ required for each process option are listed right of the respective process options. Each parameter of the disjoint-parameter benchmark model (A) appears in exactly one process option (disjoint parameters) while in the more advanced benchmark model (B) parameters $x_1$ and $x_3$ appear in multiple process options and processes. The process formulations can be found in Eq.s 3 to 9 and Eq.s 10 to 16. (C) The third setup is used for a sensitivity analysis of the hydrologic modeling framework Raven. The three options $M_i$ are used for the infiltration process, three options $N_i$ for quickflow, two options $O_i$ for evaporation, two options $P_i$ for baseflow, and three options $Q_i$ for snow balance. All other processes needed for the model are used with one fixed option, i.e., convolution for surface and delayed runoff $R_1$ and $S_1$, respectively, potential melt $T_1$, percolation $U_1$, rain-snow partitioning $V_1$, and precipitation correction $W_1$. The remaining processes also have only one option but none of them contains tunable parameters. They are merged to a "remaining" process $X_1$. The Raven model parameters $x_1$ to $x_{35}$ are listed right of the process options. Details about the chosen process options can be found in Appendix C Table C1, and details on the model parameters and their ranges in Appendix C Table C2.





### 2.1.2 Hydrologic Modeling Framework Raven

The Raven hydrologic modeling framework developed by James R. Craig at the University of Waterloo (Craig et al., 2020) is a C++ based framework that gives users full flexibility regarding model input handling and hydrologic process description chosen for each process of the water cycle. It is platform independent, open-source and is retrievable from http://raven.uwaterloo.ca. For this study we used the released version 3.0. The Raven framework currently allows for an ensemble of about $8 \times 10^{12}$ hydrologic model configurations with, for example, 14 options for infiltration, 13 options for percolation, and 9 for baseflow handling. The overall number of model structures is hypothetical as not all processes need to be present in a model setup. For example, the sublimation process allows for 6 different options in Raven but would likely not be used to model an arid catchment. Further, other processes might appear several times. For example, convolution processes can be defined for each soil layer and hence would increase the number of possible models.

In Raven, the user defines the model as a list of hydrologic processes which move water between storage compartments corresponding to physical stores (e.g., topsoil, canopy, snowpack). The list determines the state variables, connections between stores and the parameters required. For each hydrologic process, several options of process algorithms are implemented. There are, for example, 14 infiltration process options available. Amongst others, the GR4J (Perrin et al., 2003), HMETS (Martel et al., 2017), UBC watershed model (Quick and Pipes, 2009), PRMS (Markstrom et al., 2015), HBV (Bergström, 1995), VIC (Wood et al., 1992), and VIC/ARNO (Clark et al., 2008) infiltration descriptions are implemented. All options of each process can be combined with all options of other processes. Raven can fully emulate a number of hydrologic models (GR4J, HMETS, MOHYSE, HBV-EC, and the UBC Watershed model) by choosing specific configurations of the hydrologic processes.

Raven has another unique feature relative to other modular frameworks: Rather than selecting one process option (e.g., HMETS method for estimation of infiltration runoff fluxes) one can specify multiple process options (e.g., HMETS, VIC/ARNO, and HBV) and define weights for each option (e.g., 0.4, 0.5,and 0.1, respectively). Raven then uses the weighted sum of the fluxes calculated by the process options internally. Raven is run only once and not multiple times to obtain the outputs for the multiple process options. Based on this feature, we chose Raven as model for our study. Please note that the proposed sensitivity method is applicable for any multi-model framework that allows to mix-and-match process descriptions. However, in the case of a framework without weights for process options, the application of the method would be much less efficient.

For the case study used herein, Raven is applied in lumped mode and we have chosen three different options $M_i$ for the infiltration process, three options $N_i$ for quickflow, two options $O_i$ for evaporation, two options $P_i$ for baseflow, and three options $Q_i$ for snow balance. All other processes, i.e. convolution for surface runoff $R_1$ and delayed runoff $S_1$, potential melt $T_1$, percolation $U_1$, rain-snow partitioning $V_1$, and precipitation correction $W_1$ are used with one fixed process option. The remaining processes also have only one option but none of them contains tunable parameters. They are merged to a "remaining" process $X_1$. This remaining process will never appear in the sensitivity analysis because it is constant. Details of process options can be found in Appendix C Table C1.

This selection of process options results in $3 \times 3 \times 2 \times 2 \times 3 \times 1 \times 1 \times 1 \times 1 \times 1 \times 1 \times 1 = 108$ possible models when only one option is allowed per process. When the first option of each process $M_1, N_1, O_1, P_1, Q_1, R_1, S_1, T_1, U_1, V_1, W_1$, and





$X_1$ is chosen and parameter $x_{35}$ is set to zero, the Raven setting emulates the HMETS model (Martel et al., 2017) perfectly. All other combinations are unnamed models. The Raven model is stable for all of these combinations although a check of the hydrologic realism of these models, as done by Clark et al. (2008), was not performed.

Figure 1B shows the possible combinations and associated active parameters. In total 35 model parameters are active in at least one model option. The details on model parameters and their ranges used for the sampling of parameter sets are listed in Table C2 of the appendix. An additional number of $13 (= 3 + 3 + 2 + 2 + 3)$ weights is required for the weighted model setup (similar to Eq. 18), and are also sampled using the approach described in Appendix A. Therein, $8 (= 2 + 2 + 1 + 1 + 2)$ parameters $r_i$ are sampled uniformly $\mathcal{U}[0, 1]$ and transformed into weights $w_i$.

### 2.1.3 Case Study Domain

The Salmon River catchment located in the Canadian Rocky Mountains in British Columbia is selected as the study watershed. The domain is depicted in Figure 2A and was chosen only for the purpose of demonstrating the proposed method. The catchment drains towards a Water Survey Canada (WSC) streamflow gauge station near Prince George (WSC ID 08KC001; latitude 54.09639° N, latitude -122.67972° W, elevation 606 m) and has continuous data since 1953. The 4230 km$^2$ large, low-human impacted catchment is mainly evergreen needleleaf forested (83% of whole domain) on a loamy (63%) and loamy sandy (25%) soil (Fig. 2C and Fig. 2D).

Meteorological inputs are obtained from Natural Resources Canada on an approximately $10 \times 10$ km$^2$ grid. The model is setup in lumped mode. Hence, all available forcings grid points that fall within the catchment have been aggregated. Average daily temperature and the daily sum of precipitation have been used to force the Raven model. The forcings are available from 1954 to 2010. The average annual precipitation of the Salmon River catchment is 592.7 mm over the 57 years of available data. The monthly distribution of precipitation is shown in Figure 2B where rain is highlighted as the dark blue portion of each bar and snow as the light blue portion of each bar. The basin has a dryness index (PET/P) of 0.735, which demonstrates the energy limitation of this catchment.

The lumped model was setup for the simulation period from January 1, 1989 to December 31, 2010 while the first two years were discarded as warm-up. Hence, 20 years of daily streamflow simulations were used for this study.

## 2.2 Sensitivity Analysis: Theory

In this section we briefly describe the Sobol' method (Sobol', 1993) that is traditionally used to derive model parameter sensitivities (Sec. 2.2.1). This corresponds to the sensitivity metric A mentioned in the introduction. To calculate the sensitivity metrics B, C, and D, we propose an extended version of the Sobol' method which is introduced in Sec 2.2.2.

### 2.2.1 Traditional Sobol' Sensitivity Analysis: Sensitivities of Individual Model Parameters

Traditionally, the Sobol' sensitivity analysis- as all other methods- focuses on the sensitivity of model parameters. In case of multiple models, one would typically run the analysis individually for each model and might aggregate the sensitivity index

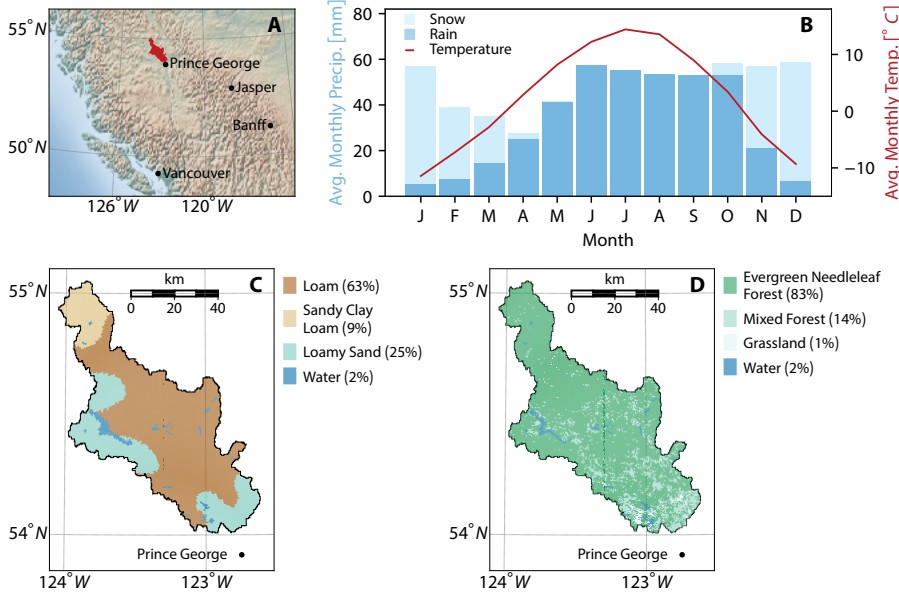

**Figure 2.** (A) Location of the Salmon River catchment (red polygon) in British Columbia, Canada. The watershed is 4230 km$^2$ and located around 700 km north of Vancouver. It is located in the Rocky Mountains with an elevation of 606 m above sea level at the streamflow gauge station of the Salmon River (08KC001). (B) The average monthly mean temperatures (red line) and average monthly precipitation is divided into rain (dark blue) and snow (light blue). Maps of (C) the four soil types based on the Harmonized World Soil Data (HWSD; 30") (Nachtergaele et al., 2010) and (D) four land cover types based on the MCD12Q1 MODIS/Terra+Aqua Land Cover (500m) (Friedl and Sulla-Menashe, 2015) of the Salmon River catchment are provided. The colors indicate different soil and land use classes.

estimates for parameters that are active in multiple models. This, however, may underestimate the sensitivity of parameters that are active in only a few models while parameters that are active in almost all models might be overestimated in their aggregated sensitivity.

We here briefly revisit the implementation of the traditional Sobol' method to emphasize the differences with the extended method we propose that will be able to handle multiple process options and derive (overall) parameter sensitivities, sensitivities of process options, and sensitivities of whole processes.

Usually two Sobol' indexes are derived: the main and total Sobol' index. The main Sobol' index $S_{x_i}$ of a parameter $x_i$ regarding a certain model output $f(\boldsymbol{x_i})$ represents the variability in the model output $V_i$ that can be achieved by changing this
parameter while keeping all other parameters at a nominal value. This impact is normalized by the overall model variability $V(f)$ that can be generated when all model parameters are varied. Therefore, the main index is derived by

$$S_{x_i} \quad = \quad \frac{V_i}{V(f)} = \frac{V[E(f|x_i)]}{V(f)} \tag{19}$$

where $V$ depicts variances and $E$ expected values. $E(f|x_i)$ is the expected model output when the model parameter $x_i$ is fixed.





Similarly, the total Sobol' index $ST_{x_i}$ for a parameter $x_i$ regarding a certain model output $f(\boldsymbol{x_i})$ is similar to the main
index but includes parameter interactions. Therefore, it is derived using the variability of model output that can be generated
by changing all parameter subsets that include parameter $x_i$. Since there might be a large number of such subsets, the total
index for parameter $x_i$ can also be viewed as 1 minus the variability that can be achieved by changing all parameters but not
parameter $x_i$ ($V_{\sim i}$) normalized by the overall possible model output variability $V_f$:

$$ST_{x_i} \quad = \quad 1 - \frac{V_{\sim i}}{V(f)} = 1 - \frac{V[E(f|x_{\sim i})]}{V(f)} \tag{20}$$

where $V$ depicts variances and $E$ expected values. $E(f|x_{\sim i})$ is the expected model output when all model parameter except
$x_i$ are fixed.

Sobol' (1993) proposed an elegant and efficient method to approximate the variances $V_i$, $V_{\sim i}$, and $V(f)$. We have used the
implementations proposed by Cuntz et al. (2015) (Appendix D therein). Unlike derivative based methods, the Sobol' index
calculations are only dependent on the model outputs but not the parameter values $x_i$.

For the numerical estimation of the indexes $S_{x_i}$ and $ST_{x_i}$, one constructs two base matrices $\mathcal{A}$ and $\mathcal{B}$ which each contain $K$
parameter sets (rows) of $N$ parameters (columns). The samples are assumed to be independent within one matrix and between
the matrices. Based on $\mathcal{A}$ and $\mathcal{B}$, a set of additional $N$ matrices $\mathcal{C}_i$ is constructed. $\mathcal{C}_i$ is a copy of $\mathcal{A}$ but column $i$ is replaced
with column $i$ of matrix $\mathcal{B}$. The model then needs to be forced with all the parameter sets; in total $K \times (N+2)$ model runs
are required where $N$ is the number of parameters and $K$ is the so-called number of reference parameter sets. $K$ needs to be
chosen to be large enough to obtain stable Sobol' indexes. This number is highly dependent on the model but $K = 1000$ seems
to be a good rule-of-thumb (Cuntz et al., 2015, 2016).

The 12 possible shared-parameter benchmark models (Eq. 2) contain 3 (4 models), 4 (5 models), 5 (2 models), or 6 (1 model)
parameters. Hence, $72\,000 \, (= 4 \times 5000 + 5 \times 6000 + 2 \times 7000 + 1 \times 8000)$ model runs would be required if $K = 1000$ reference
parameter sets would be used.

**2.2.2   Extended Sobol' Sensitivity Analysis: Sensitivities of Groups of Model Parameters**

The Sobol' method is here generalized to groups of parameters $x_G$ rather than focusing on individual parameters $x_i$. The
subscript $G$ is used here to refer to parameter groups, such that $V_G$ represents the variance of a group of parameters $x_G$, e.g.,
the set $x_G = \{x_2, x_4, x_5\}$. The calculation of the main and total Sobol' indexes is marginally changed: Instead of changing
individual parameters $x_i$ groups of parameters get changed. The derivation of the main index gets generalized to:

$$S_{x_G} \quad = \quad \frac{V_G}{V(f)} = \frac{V[E(f|x_G)]}{V(f)} \tag{21}$$

where $V$ depicts variances and $E$ expected values. $E(f|x_G)$ is the expected model output when the set of model parameters
$x_G$ is fixed. This simplifies to Eq. 19 in case the group $x_G$ contains exactly one model parameter $x_i$. Similarly, the total Sobol'
index can be generalized to:

$$ST_{x_G} \quad = \quad 1 - \frac{V_{\sim G}}{V(f)} = 1 - \frac{V[E(f|x_{\sim G})]}{V(f)} \tag{22}$$



where $V$ depicts variances and $E$ expected values. $E(f|x_{\sim G})$ is the expected model output when all model parameters except the ones in of group $x_G$ are fixed. This simplifies to Eq. 20 when the group $x_G$ contains only the parameter $x_i$. Note that the groups are not assumed to be mutually exclusive, which means that parameters can appear in multiple groups.

The numerical approximation of these indexes is similar to the traditional approach. It is again based on the two matrices $\mathcal{A}$ and $\mathcal{B}$ containing $K$ parameter sets each. Assuming that the sensitivity of $M$ groups needs to be estimated, $M$ matrices $\mathcal{C}_m$ have to be constructed where $\mathcal{C}_m$ is a copy of $\mathcal{A}$ but all columns that correspond to parameters in group $m$ are replaced by the corresponding column of $\mathcal{B}$. For example, if the group consists of parameters $x_2$, $x_4$, and $x_5$, the columns 2, 4, and 5 would be replaced by the columns 2, 4, and 5 of matrix $\mathcal{B}$. The number of model runs that need to be performed is $(M+2) \times K$ where $K$ is the number of reference parameter sets.

The Extended Sobol Sensitivity Analysis (xSSA) method can be used to derive conventional model parameter sensitivities by using groups that contain exactly one of the model parameters $x_i$ or random number $r_i$ that are required to derive weights (sensitivity metric B). It can also supply the sensitivity of process options by defining a group for each process option containing exactly the parameters of that option (sensitivity metric C) or defining a group for each process containing all parameters active in at least one of the process options to derive the sensitivity of whole model processes (sensitivity metric D).

As an example, (see Fig. 1A) the group to derive the sensitivity of process option $A_2$ of the artificial benchmark model would contain parameters $x_1$ and $x_2$. The group to determine the sensitivity of process $B$ of the benchmark model would contain parameters $x_2$, $x_3$, $x_4$, and $x_5$.

The shared-parameter benchmark model consists of 7 model parameters $x_i$ and 4 random variables $r_i$ used to derive the 7 weights $w_i$. Let's assume we used $K = 1000$ reference parameter sets for each of the three analyses. One would require $13\,000$ $(= (7+4+2) \times 1000)$ model runs to derive individual parameter sensitivities using the xSSA method. This is compared to the $72\,000$ model runs required when the conventional Sobol' sensitivity analysis method is applied to the 12 individual models. It requires $13\,000 (= (7+4+2) \times 1000)$ additional model runs to derive the sensitivity of the seven process options $A_1$, $A_2$, $B_1$, ... $C_2$ and $5000 (= (3+2) \times 1000)$ additional model runs to derive the sensitivity of the three processes $A$, $B$, and $C$. The total of $31\,000$ model runs for all three analyses thus leads to a computational cost reduction of 57% while providing additional information about process option and process sensitivity.

## 2.3 Sensitivity Analysis: Experiments

Four experiments will be performed for each of the two benchmark models to demonstrate that the proposed method is able to obtain the analytically derived values available for the the benchmark examples (Sec. 2.3.1). Another set of four experiments is performed using the Salmon River catchment model (Sec. 2.3.2). These experiments are performed to demonstrate the type of insights that can be obtained for hydrologic models using the proposed method.

### 2.3.1 Experiments Using the Benchmark Models

The artificial benchmark models are used to prove that the proposed method of Extended Sobol' sensitivity indexes and its implementation is working. They are furthermore employed to demonstrate some limitations of the existing method proposed





by Baroni and Tarantola (2014) to derive sensitivities regarding model structures. The analytically-derived values for the traditional approach analyzing the individual 12 models independently (sensitivity metric A) can be found in Appendix B

Table B1 for the shared-parameter model setup. The budget of such an analysis is $72 \times K$ with $K$ reference parameter sets as described in Sec. 2.2.1. Mai and Tolson (2019) and Cuntz et al. (2015) have demonstrated that these indexes can be obtained with a classical Sobol' analysis for the Ishigami-Homma model.

The analytically-derived values for the sensitivity metrics B to D of the shared-parameter model are available in Eq. B1-B3 of the Appendix B.

The xSSA method is tested using different budgets to show that numerical values indeed converge. The number of reference sets used are

$K = \{50, 100, 200, 500, 1000, 2000, 5000, 10000, 20000, 50000, 100000\}$. The budget to derive parameter sensitivities (sensitivity metric B) is $(11+2) \times K$ for 7 parameters $x_i$ and 4 weight deriving random numbers $r_i$. To derive sensitivities of process options (sensitivity metric C) $(11+2) \times K$ model runs are required for the 7 process options $A_1$, $A_2$, ..., $C_2$ and the 4 weight

deriving random numbers $r_i$. For the analysis of processes (sensitivity metric D) the model needs to be run $(3+2) \times K$ times to obtain sensitivities of the 3 processes $A$, $B$, and $C$.

The Baroni method also applied to the two benchmark models was setup using the same computational budgets $K$ as above. The method requires additionally the definition of the algorithmic parameter $n_i$ which denotes the number of realizations for each source of uncertainty $U_i$ analyzed ($n_i$ and $U_i$ are terms used in the original publication). The term "sources of uncertainty"

($U_i$) is used by Baroni and Tarantola (2014) to describe groups of parameters with aggregated sensitivity, and is here equivalent to the process groupings $A$, $B$, and $C$ for the disjoint-parameter benchmark model and $D$, $E$, and $F$ for the shared-parameter benchmark model. Several values for $n_i$ were tested (32, 64, 128, 256, 512, 1024). Only results for $n_i$ equal to 128 which was used in the original publication (Baroni and Tarantola, 2014) will be reported as all other results appeared to be similar.

The errors between approximated main effects $S_i^{(\text{appr})}$ and the analytically derived true indexes $S_i^{(\text{theo})}$ as well as errors

between the approximated total indexes $ST_i^{(\text{appr})}$ and its analytically derived truth $ST_i^{(\text{theo})}$ are reported for both the xSSA and the Baroni method.

### 2.3.2 Experiments Using the Raven Modeling Framework

The Salmon watershed model is analyzed using the xSSA method with $K = 1000$ reference parameter sets assuming that this number of parameter sets is large enough to derive stable results. The analysis of individual models (sensitivity metric A)

would have required $3\,258\,000$ model runs and is not performed here.

The budget for the sensitivity metric B analyzing unconditional sensitivity of parameters independent of choice of model options, results in $(43+2) \times K$ model runs for the 35 model parameters $x_i$ and 8 weight deriving random numbers $r_i$. The budget for process options is $(27+2) \times K$ model runs for the 19 process options $M_1$, $M_2$, ..., $W_1$ and 8 weight deriving random numbers $r_i$. The process $X_1$ not analyzed since it does not contain any parameters and is hence constant, resulting in a

zero sensitivity. The budget for the analysis to obtain process sensitivities is $(11+2) \times K$ for the 11 processes $M$, $N$, ..., $W$.





The sensitivities are determined for simulated streamflow $Q(t)$ for the 20 year simulation period from 1991 to 2010. The main and total Sobol' indexes $S_{x_i}(t)$ and $ST_{x_i}(t)$, respectively, are determined for each time step $t$ and are aggregated to $\overline{S_{x_i}^w}$ and $\overline{ST_{x_i}^w}$ using variance-weighted means (Cuntz et al., 2015):

$$\overline{S_{x_i}^w} \quad = \frac{\sum_{t=1}^{T} V(t) S_{x_i}(t)}{\sum_{t=1}^{T} V(t)} \quad = \frac{\sum_{t=1}^{T} V_{x_i}(t)}{\sum_{t=1}^{T} V(t)}, \tag{23}$$

$$\overline{ST_{x_i}^w} \quad = \frac{\sum_{t=1}^{T} V(t) ST_{x_i}(t)}{\sum_{t=1}^{T} V(t)} \quad = 1 - \frac{\sum_{t=1}^{T} V_{x_i}(t)}{\sum_{t=1}^{T} V(t)} \tag{24}$$

where $V(t)$ is the total variance at time step $t$.

We further analyze the time dependent behavior of process sensitivities to reveal temporal patterns in the importance of processes at different times of the year. Therefore, the total process sensitivity $ST_M(t)$, $ST_N(t)$, ... $ST_W(t)$ over the 20 year simulation period ($t = 1, \ldots, 7305$) are averaged for each process at each day of the year ($t' = 1, \ldots, 365$) resulting in $ST_M(t')$, $ST_N(t')$, ... $ST_W(t')$. Sensitivity estimates from leap days are discarded. The sum of sensitivities at each time step are normalized to 1.0 in order to ease the comparison of all time steps:

$$\widehat{ST}_\mathcal{P}(t') = \frac{ST_\mathcal{P}(t')}{\sum\limits_{\mathcal{P} \in \Omega} ST_\mathcal{P}(t')} \quad \forall \mathcal{P} \in \Omega = \{M, N, \ldots, W\} \tag{25}$$

## 3 Results and Discussion

We will present the results of the Extended Sobol' Sensitivity Analysis (xSSA) applicable not only to model parameters but also for model process options and processes. First, the xSSA method will be compared to an existing method to derive sensitivities of groups of parameters introduced by Baroni and Tarantola (2014) highlighting limitations of these existing method (Section 3.1). Second, we will present the convergence of the xSSA results regarding parameters, process options, and processes focusing on the more shared-parameter benchmark model (Sec. 3.2). The results of xSSA for the hydrologic modeling framework are presented in Sec. 3.3.

### 3.1 Benchmarking Against Analytically-Derived Solutions and An Existing Method

In this section the proposed xSSA method based on grouped parameters and weighted process options will be compared against analytically-derived Sobol' sensitivity indexes for both benchmark problems (Fig. 1A and B). The xSSA method will aslso be compared to the sensitivity analysis method introduced by Baroni and Tarantola (2014) (hereafter called Baroni method) which is also making use of grouped parameters. Weighted process options are irrelevant for the Baroni method as only one "process option" was used in their publication.

The Baroni method defines the term "sources of uncertainty" ($U_i$) to describe groups of parameters with aggregated sensitivity, and is here used to be equivalent to the process groupings $A$, $B$, and $C$ for the disjoint-parameter benchmark model and $D$, $E$, and $F$ for the shared-parameter benchmark model. The Baroni method pre-samples a defined number $n_i$ of sets for each "uncertainty source" $U_i$. Forcing datasets could be one of such source of uncertainty. In this case, the Baroni method





pre-samples $n_i$ input time series and the Sobol' method would use the ID of the time series $(1 \ldots n_i)$ as the hyper-parameter to derive the sensitivity of the inputs. In the case of their proposed example, the sources of uncertainty are distinct and the parameterizations of the "sources of uncertainty" are disjoint. The question is how the Baroni method would be applied if two competing error structures of the same forcings are supposed to be tested. For example, when both error structures shared one or more of the same statistical parameters such as the mean error and error variance, eliminating the disjoint nature of pa-

rameters that all past studies using the Baroni method implicitly assume and utilize. The same question appears in the context of process option sensitivity when a parameter (e.g., porosity) is shared between multiple alternative process options or even multiple processes (e.g., a soil evaporation process option and a percolation process option). The xSSA method is not limited in these situations by using the weighted sum of all competing model options and the definition that parameters are allowed to appear in multiple process options and even in multiple processes.

To demonstrate this major difference of the Baroni and the proposed xSSA concept, we define three groups (sources of uncertainty) as the processes $A$, $B$, and $C$ of the disjoint-parameter benchmark model (Fig. 1A). For this scenario, all the parameters are disjoint and it can be shown that both methods converge to the analytically derived Sobol' indexes (Fig. 3A and 3B).

     The shared-parameter benchmark model (Fig. 1B), on the contrary, has parameters that appears in several process options of

the same process (e.g., $x_1$ in $A_1$ and $A_2$) and parameters that are involved in several process options across processes (e.g., $x_3$ in $E_2$ and $F_2$). This non-disjoint (overlapping) setting of influencing factors leads to the result that the Baroni method converges to a wrong sensitivity for some processes. This is caused by the usage of the $n_i$ pre-sampled parameter sets for each source of uncertainty (here processes). When a parameter appears in two source of uncertainty, one has to make a decision which parameter value to choose for running the model; during the Sobol' analysis (when creating the M matrices $\mathcal{C}_m$) one process

expects this parameter to stay constant while the other assumes it is getting changed. This contradiction can not be resolved. In our implementation we had chosen to use the parameter value of the last process leading to process $F$ converging to the analytically-derived correct value. Process $E$ also converges almost to the correct value but only because parameter $x_3$ which is shared between processes $E$ and $F$ is very insensitive ($ST_{x3} = 0.0045$). Process $D$ shares the highly sensitive parameter $x_2$ ($ST_{x2} = 0.77089$) with process $E$. Thus the sensitivity for process $D$ is significantly underestimated. The underestimation is

caused by the fact that the parameter value is supposed to change but it is kept constant because it gets overwritten by the value of $x_2$ of the pre-sampled set of process $E$.

     We also tested several numbers of pre-sampled parameter sets $n_i$ as this is mentioned by Baroni and Tarantola (2014) to be one factor that can influence the convergence of the method. We tested $n_i$ with 32, 64, 128, 256, 512, and 1024 and all led to the same results (results not shown).

The xSSA method does not pre-sample parameters. When one group is analyzed, all parameters contained in this group get perturbed. The $\mathcal{C}_m$ matrices can be defined without causing any contradiction or overwriting of parameter values. The method intrinsically counts repeatedly sensitivities for parameters that appear multiple times. This characteristic of this method is intentional and desirable. When the groups are defined as process options, i.e. various conceptual implementations of the same hydrologic process, parameters will be used in multiple options. Some parameters such as, for example, soil thicknesses or



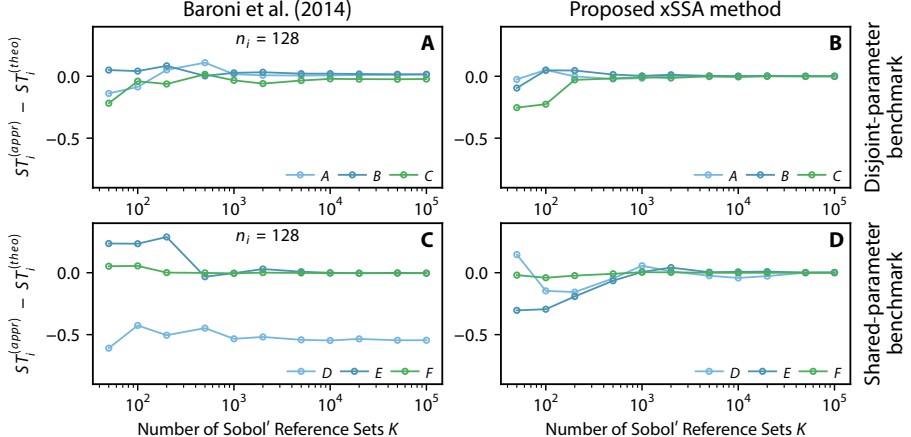

**Figure 3.** Error between approximated and analytically-derived total Sobol' sensitivity index estimates for processes of the benchmark models $ST_i^{(appr)}$ and $ST_i^{(theo)}$, respectively. The analyses are performed using increasing numbers of Sobol' references sets $K$. The errors are expected to converge to zero for increasing number of reference sets. The existing method proposed by Baroni and Tarantola (2014) (A,C) is compared to the proposed method xSSA (B, D) using the disjoint-parameter and shared-parameter benchmarking model (Fig. 1A and Fig. 1B, respectively). The model parameters are here grouped to model processes $A$, $B$, and $C$ for the disjoint-parameter benchmarking model and $D$, $E$, and $F$ for the shared-parameter benchmarking model. The main difference between these two models is that parameters of the latter model can appear in multiple processes (or groups of uncertainty) which is regarded to be more realistic. Note that the x-axis is in logarithmic scale.

porosity might even be required across processes (for example infiltration and percolation). We do not expect the process sensitivities to sum up to 1 which is anyway not achievable with non-additive models. In addition to the practical benefit of allowing for non-disjoint parameter groups, the theoretical underpinning of analytically-derived Sobol' indexes does not require this constraint as shown with the xSSA results converging towards those values.

It is notable that several publications that are considering structural sensitivities so far are limited by the disjoint definition of parameter groups (Baroni and Tarantola, 2014; Schürz et al., 2019; Francke et al., 2018; Günther et al., 2019). They would hence show a similar behavior as presented here for the example Baroni method results.

### 3.2 Extended Sobol' Sensitivity Analysis for Shared-Parameter Benchmark Setup

The shared-parameter benchmark setup is utilized to compare the xSSA derived numerical sensitivity metric values with the analytically derived, correct sensitivity metric values for all three metrics: parameters, process options, and processes. We have chosen the shared-parameter over the disjoint-parameter benchmark model here as it appears to be the more difficult model to analyze. The errors converge to zero in every analysis and hence proves that the implementation of the Extended Sobol' sensitivity analysis is coherent with the analytical theory (Figure 4).



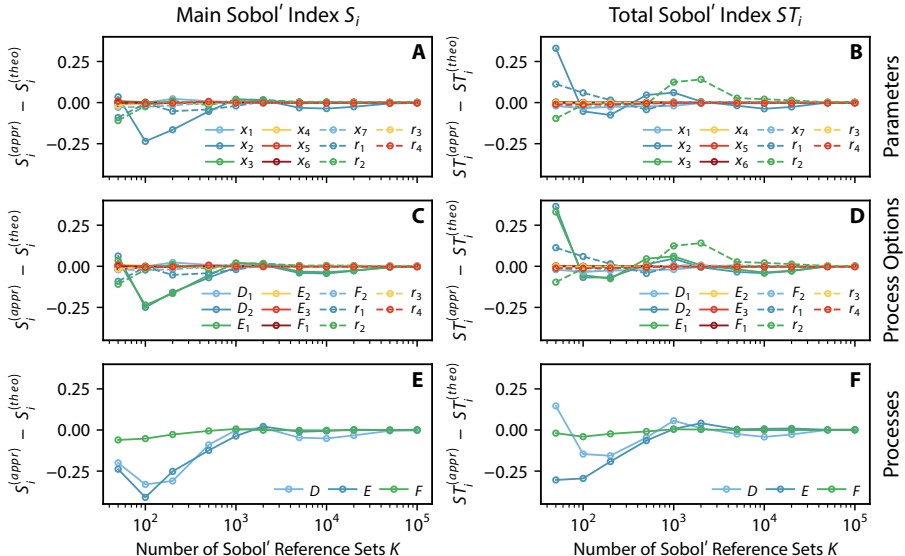

**Figure 4.** Error between xSSA approximated and analytically-derived Sobol' sensitivity index estimates $\mathcal{S}_i^{(appr)}$ and $\mathcal{S}_i^{(theo)}$, respectively. The errors are derived for the main Sobol indexes $S_i$ and the total Sobol indexes $ST_i$. The analyses were performed for (A,B) model parameters, (C,D) process options, and (E,F) processes of the shared-parameter benchmark model (Figure 1B, Eq. 2). The analyses are performed using increasing numbers of Sobol' references sets $K$. Note that the x-axis is in logarithmic scale. Figure 4F is the same as Figure 3D.

It holds for all three analyses that the model parameters/ options/ processes with the largest sensitivities converge slowest. For example, model parameter $x_2$ and weight generating random number $r_2$ have much higher sensitivities than other parameters
analyzed and need about 5000 model runs to obtain an error below 0.1 (Fig. 4A and B). Similarly, Figure Fig. 4C and 4D show that for the most influential process options ($D_2$, $E_1$) and most influential variable ($r_2$) converge slowest ($S_{D_2} = 0.43$, $S_{E_1} = 0.38$, $S_{r_2} = 0.06$, $ST_{D_2} = 0.80$, $ST_{E_1} = 0.77$, $ST_{r_2} = 0.32$). Process F's sensitivity estimates converge faster than the other two as is consistent with the fact that this process is the least sensitive (Fig. 4E and F).

It can be noted that the weight generating random numbers $r_i$ show high interaction effects- which makes sense since they
always couple at least two parameters or process options- and hence tend to converge slower due to the higher sensitivity. In this study we are primarily interested in the model parameter, process option and process sensitivities and hence suggest that a number of $K = 1000$ model runs is sufficient to derive useful sensitivity estimates.

### 3.3 Extended Sobol' Sensitivity Analysis Applied to Hydrologic Modeling Framework

Each subsection here focuses on one experiment performed using the hydrologic modeling framework Raven. The subsec-
tions will address the results of the unconditional parameter sensitivities (Sec. 3.3.1), the sensitivities of the process options (Sec. 3.3.2) and the processes (Sec. 3.3.3). All these sensitivity indexes are presented as variance-weighted aggregates over time such that one index per model parameter, option, or process can be analyzed. The last subsection focuses on temporally





varying process sensitivities over the course of the year (Sec. 3.3.4) as shown to be of importance previously (Dobler and Pappenberger, 2012; Herman et al., 2013; Günther et al., 2019; Bajracharya et al., 2020).

It is important to note that the results of the analysis for the hydrologic framework are the product of an iterative process where the intermediate results are not quantitatively reported on. Qualitatively, we wish to emphasize that the intermediate xSSA-based results helped us to improve the modeling framework by identifying sources of model instabilities and non-intuitive model results. It was especially helpful to have estimates for aggregated model compartments, i.e., process options and processes. We strongly believe that this kind of analysis will help to analyze the hydrologic realism of models since the

estimates are easier to interpret and to compare to experience and known evidence.

### 3.3.1    (Unconditional) Parameter Sensitivity

The variance-weighted main and total Sobol' sensitivity estimates $\overline{S_{x_i}^w}$ and $\overline{ST_{x_i}^w}$ are shown in Figure 5. The sensitivities are unconditional since the estimates are averaging over all possible model structures through the weighted sum of all analyzed model options as it is described in Eq. 18. The analysis of model parameters (Fig. 5A) shows that the most important ones are

$x_{24}$ to $x_{27}$ which are all associated with the potential melt (process $T$) that handles the melting of snowpack until it is gone. The quickflow parameters $x_5$ (maximum release rate from topsoil) and $x_6$ (baseflow rate exponent $n$ of topsoil) are sensitive as well. Parameters of medium sensitivity are $x_8$ (PET correction factor), $x_{16}$ (degree day refreeze factor), $x_{18}$ (refreeze factor), $x_{19}$ (maximum snow liquid saturation), $x_{29}$ (thickness of topsoil), $x_{31}$ (temperature of rain-snow transition), and $x_{34}$ (snow correction factor).

Besides that, the most influential parameters are the weight generating random variables associated to processes that are most sensitive (indicated by same color in Fig. 5A), i.e., $r_3$, $r_4$, $r_7$. This is intuitive since switching processes may cause large variability in the model outputs and hence shifting their weighted averages is also likely to lead to large variability.

     A sensitivity analysis regarding model parameters is often performed prior to model calibration to identify the most sensitive and hence most sensitive parameters which are in turn the parameters that are most likely to be identifiable during calibration.

The analysis shows that 13 of the 35 parameters ($x_5$, $x_6$, $x_8$, $x_{16}$, $x_{18}$, $x_{19}$, $x_{24}$, $x_{25}$, $x_{26}$, $x_{27}$, $x_{29}$, $x_{31}$, and $x_{34}$) are responsible for 96.5% of the overall model variability (only $\overline{ST_{x_i}^w}$; 77.2% if $\overline{ST_{r_i}^w}$ included). All other parameters are unlikely to be identifiable during model calibration using streamflow measurements alone. Independent of the model structure selected, these model parameters are negligible and thus could be fixed at default values for the Salmon River catchment over the 20-year simulation period.

This analysis helps to identify the most important parameters independent of model structure and therefore helps to identify main sources of parametric uncertainty in models despite structural configuration, presuming that individual structures are equally viable. It likewise determines non-identifiable parameters- as a traditional sensitivity analysis does- with respect to streamflow.





### 3.3.2 Process Option Sensitivity

The results of the sensitivity analysis of model parameters are consistent with the analysis of process options (Fig. 5B) where all model parameters used in a process option are varied together rather than individually. The analysis of the process option sensitivities identifies options as most sensitive that contain model parameters that have been previously determined to be sensitive.

The potential melt process- the algorithm used to determine incoming melt energy- is used in this study only with one
process option (POTMELT_HMETS) and it is still the hydrologic process that the simulated streamflow is most sensitive to (orange bar). The three infiltration options are all equally sensitive (light blue bars). Same holds for the two options of the evaporation process (dark blue bars) which are slightly more influential than the infiltration processes (light blue bars). The quickflow options BASE_VIC and BASE_TOPMODEL (medium blue bars) are the second-most sensitive after the potential melt. The quickflow option BASE_LINEAR_ANALYTIC however is much less sensitive. The two baseflow options
(dark green bars) as well as the convolution options for surface and delayed runoff (yellow and light green bar) exhibit almost no influence with sensitivity metrics near zero ($\overline{ST_G^w} < 0.0017$ with $G \in \{P_1, P_2, R_1, S_1\}\}$). The only percolation option (PERC LINEAR; light red bar), rain-snow partitioning option (RAINSNOW_HBV; medium red bar), and precipitation correction (RAINSNOW_CORRECTION; dark red bar) are showing medium sensitivities similar to the ones of the infiltration options. The SNOBAL_HBV option is the most sensitive among the three snow balance options. SNOBAL_HMETS is
slightly less sensitive while SNOBAL_SIMPLE_MELT has a zero sensitivity. The latter serves as a consistency check of the implementation. The zero sensitivity is expected since the SNOBAL_SIMPLE_MELT option does not require any parameters (see Tab. C1). Model outputs of such options do not change for different model runs and hence have a zero variance which leads to a zero Sobol' index. Another interesting result is the high sensitivity of the weight generating random numbers associated with the snow balance options ($r_7$ and $r_8$). The sensitivity of these parameters is caused by the fact that they are
responsible for the "mixing" of the outputs of the model options. In this case they are mixing a process that is always the same (SNOBAL_SIMPLE_MELT) and two options that can vary significantly with parameter choice (SNOBAL_HBV and SNOBAL_HMETS). Hence, the weighting of these processes can perturb the model output drastically and is hence yielding a high sensitivity of $r_7$ and $r_8$.

In summary, it can be deduced that the potential melt, the quickflow options BASE_VIC and BASE_TOPMODEL, and the
evaporation options are most influential upon modeled streamflow. The interpretation and use of this process option sensitivity is open, and depends upon the purpose of the sensitivity analysis. As an example of interpretation, we can consider whether or not we wish to maximize the flexibility of our models in calibration, and if so, we may wish to discard insensitive processes. The three infiltration options are equally sensitive and hence equally appropriate. The choice of the quickflow option will therefore not influence the model performance.

This analysis of process options allows, for the first time, to objectively compare model process options by mix-and-matching all of them through the approach of a weighted mean of all outputs. It can assist the setup of models by guiding choices of process options and hence guides model structure decisions depending on the purpose of the model built.





### 3.3.3 Process Sensitivity

The sensitivity analysis of the eleven processes (Fig. 5C) consistently identifies potential melt $T$ (orange bar) to be the dominat-

ing process for the Salmon River catchment. Technically, potential melt as well as rain-snow partitioning $V$ and precipitation correction $W$ are handling inputs to the hydrologic system and can hence be regarded to quantify input uncertainties or, in other words, adjustments of forcing functions. The three processes are responsible for 42.1% of the overall model variability in this catchment. Note that the process weights $r_i$, unlike for parameter and process option sensitivities, are not explicitly included in the process sensitivity results in Fig. 5C (unlike Fig.s 5A and 5B). The weights are part of the parameters that get grouped

for each process to assess its sensitivity.

Processes associated to the surface are quickflow $N$ and snow balance $Q$ (medium blue and medium green bar, respectively) which are the second and third-most influential processes. These two processes control together about 38.5% of the model variability. The strong impact of these processes (together with the input adjustments) highlights the importance of snow and melting processes in this mountainous, energy-limited catchment.

The soil-related processes of infiltration $M$, evaporation $O$, and percolation $U$ show a medium sensitivity (19.2% of total model variability). This demonstrates that soil and surface processes are of secondary importance for streamflow prediction, but they may gain importance if the uncertainty of the snow and melting processes can be reduced, i.e. by narrowing parameter ranges during calibration.

Baseflow $P$, and the convolution of the surface $R$ and delayed runoff $S$ have almost no influence on the simulated streamflow

(control on 0.2% of overall variability). These three processes demonstrate that subsurface and routing processes are not important in this catchment over the 20-year simulation period at a daily time step. This in turn leads to the conclusion that even if, for example, additional ground water observations were available, it would not help to reduce model streamflow prediction uncertainty.

This model structure-based sensitivity analysis can help to guide model development by targeting the dominant model

processes. It derives a high-level sensitivity of the main model components, i.e. processes. It reveals, for the first time, the sensitivity of model processes independent of model structure chosen and hence is one step towards sensitivity analyses regarding model structure using a true model ensemble by mix-and-matching a variety of model process options.

### 3.3.4 Process Sensitivity Over Time

The previous analysis estimated the time-aggregated sensitivities $\overline{S_{\mathcal{P}}^w}$ and $\overline{ST_{\mathcal{P}}^w}$ of model processes $\mathcal{P}$ (Fig. 5C) which might

mask interesting patterns in the temporal sensitivities of streamflow to the eleven processes. We therefore augment the analysis by calculating the normalized total Sobol' sensitivity indexes $\widehat{ST}_{\mathcal{P}}(t')$ of each process $\mathcal{P}$ at every day of the year $t'$ (Fig. 6). Each value displayed is an average over 20 values- one for each year of the 20 year simulation period. The figure also shows the weights $V(t)$ (Fig. 6 black line) that were used to derive the variance-weighted total Sobol' indexes (Eq. 24) previously discussed using Fig. 5C. The weights are generally higher during the high-flow freshet period (mid Feb to mid May) and are

close to zero for the rest of the year.





Infiltration (light blue) has an almost constant but minor sensitivity throughout the whole year. Quickflow (medium blue) is most of the time the dominating process- especially in summer- but not during the high-flow melt season. Evaporation (dark blue) is consistently responsible for about 35.4% of the sensitivity during summer (Jun to Oct) and is, expectedly, less important during winter. Snow balance (medium green) and potential melt (orange) are important as long as snow is present

(Nov to May). Potential melt is about twice as influential than snow balance process. Percolation (light red) is almost constant in its sensitivity but nearly negligible. Baseflow (dark green) and the convolution of the surface runoff as well as the delayed runoff (light green and yellow) are not even visible in the graph and have negligible sensitivities throughout the whole year.

These results highlight the importance of the weighting procedure when deriving the aggregated sensitivities shown in Fig. 5. Potential melt (orange) is the most sensitive process when using variance-weighted aggregates due to its dominant influence in

the high-flow season. The arithmetic mean of all time-dependent sensitivities $S_i(t)$ and $ST_i(t)$ would have certainly resulted in a much higher sensitivity of the quickflow process which is not as important during the melting period but is responsible for 41.7% of the model variability during summer (Jun to Oct). The same holds for the evaporation process, which is highly sensitive in summer but not during the melting season.

## 4 Conclusions

The traditional method to derive sensitivity index estimates for model parameters conditional on a fixed model structure is of limited applicability when the model is allowed to vary in its structure. First, the number of model runs can be massive when each model is analyzed independently. Second, the analysis derives a unique model-dependent sensitivity index for each parameter but no overall parameter sensitivity across the ensemble. Third, aggregated sensitivities of model processes or the sensitivity of a set of process options may lead to more useful insights than analyzing individual model parameters.

In this work we introduce two new concepts. The first is the idea of formulating the model ensemble using weighted sums of process option outputs for each process. This converts the countable, discrete model ensemble space into an infinite, continuous model space. The method of weighted process options is shown to significantly reduce number of model runs required to run a sensitivity analysis based on model parameters. For the shared-parameter benchmark model 81.9% fewer model runs are required (A: 72 000 vs B: 13 000). For the hydrologic model example, the reductions is greater than 98.6% (A: 3 258 000 vs B:

45 000). The method of weighted process options derives unconditional sensitivities of the model parameters independent on the model structure.

The second key contribution here is the application of the conventional Sobol' sensitivity analysis method based on grouped of parameters and interpreting these groups as process options and hydrologic processes. The Extended Sobol' Sensitivity Analysis (xSSA) method uses these groups of parameters to perturb them simultaneously rather than individually, allowing

to simultaneously perform analyses for model parameters, model process options and model processes. While grouping of parameters is not a new concept for Sobol' analyses they have to our knowledge not yet been interpreted in the context of model structural sensitivity assessment. The method was successfully tested using two artificial benchmark models based upon the Ishigami-Homma function. The estimated sensitivity indexes are proven to converge against the analytically derived Sobol'



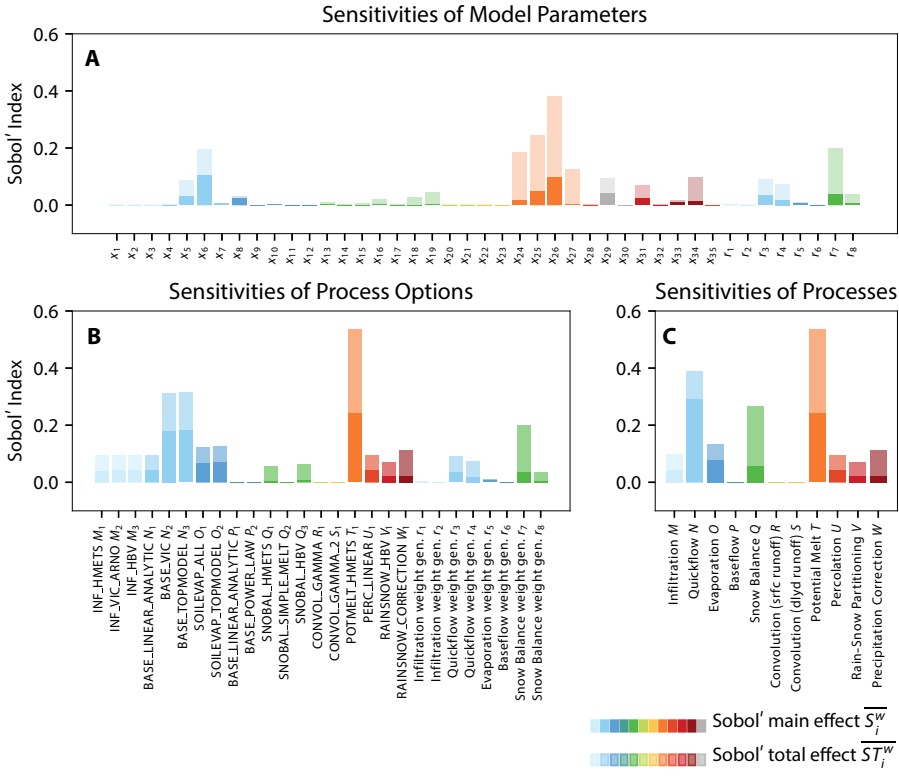

**Figure 5.** Results of the Sobol' sensitivity analysis of the hydrologic modeling framework Raven. (A) The sensitivities of 35 model parameters (see Table C2) and 8 parameters $r_i$ that are used to determine the weights of process options are estimated. The Sobol' sensitivity index estimates are determined also for (B) 19 process options and (C) the 11 processes. The different colors indicate the association of parameters and process options to the eleven processes. Parameters $x_{29}$ and $x_{30}$ are associated with several process options and are not colored but gray. The Sobol' main and total effects are shown (dark and light colored bars, respectively). All sensitivity index estimates shown are originally time-dependent and are aggregated as variance-weighted averages (Eq. 23 and 24). The average weights over the course of the year are shown in Figure 6.



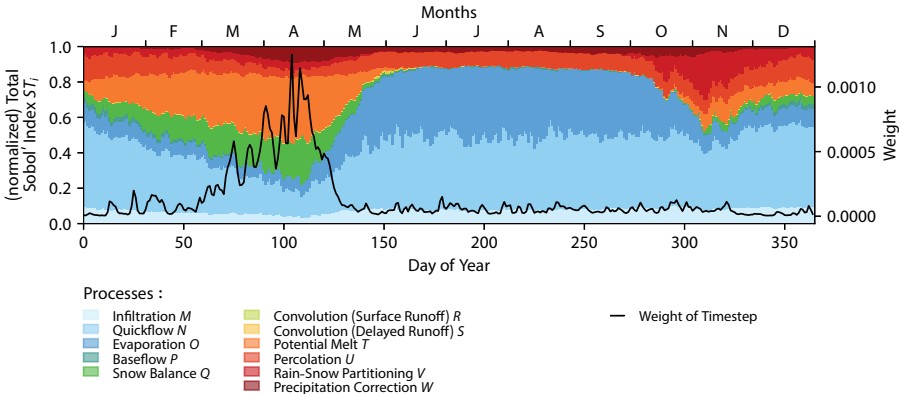

**Figure 6.** Results of the Sobol' sensitivity analysis of the hydrologic modeling framework Raven. The sensitivities of eleven processes are shown as their averages per day of the year (colored bars). The simulation period is 1991 to 2010. The sensitivities are normalized such that they sum up to 1.0 at every day of the year (Eq. 25). The sensitivities are variance-weighted averages (Eq. 24). The (average) weight of each day of the year is shown as a black line. The weights for every time step are determined by the average simulated discharge at this time step ($V(t)$ in Eq. 24). The time-aggregated sensitivity index estimates of the eleven processes are shown in Figure 5C.

sensitivity indexes for model parameters, process options, and processes. The xSSA method is shown to resolve limitations

of an existing method that also derives sensitivities of groups of parameters but that cannot handle overlapping parameter groupings.

The Extended Sobol' sensitivity analysis method was also applied to a hydrologic modeling framework that supports the representation of internal model fluxes using a weighted sum of fluxes calculated from individual process algorithms/ options. The sensitivity analysis of the hydrologic modeling framework used here identified potential melt process and other surface

processes as the most influential processes regarding streamflow in a mountainous, energy-limited, and snow-dominated catchment while all subsurface and routing processes were insensitive. These information helps to guide further model development, model calibration, and can inform the incorporation of additional observations to reduce model uncertainty. Three processes (potential melt, rain-snow partitioning, precipitation correction) handle solely inputs to the hydrologic system and can hence be attributed as input uncertainty or, in other words, model components adjusting forcing functions.

The presented methods of weighted process options and the application of the Extended Sobol' sensitivity analysis method is presenting a simultaneous analysis of model structure, model parameters, and forcing adjustments in a frugal way consistent with known methods based on the Sobol' method.

*Code and data availability.*  The code and data used for this analysis will be made available on GitHub (https://github.com/julemai/xSSA) upon publication of the manuscript.





**Appendix A: Generating Weights**

The sampling strategy introduced by Moeini et al. (2011) can be summarized as follows: For each set of weights $N+1$ needed, first generate a vector of random numbers $(r_1, r_2, ..., r_N)$ from a uniform distribution between $0$ and $1$. The corresponding vector of weights $(w_1, w_2, ..., w_{N+1})$ can then be calculated using:

$$
\begin{aligned}
w_1 &= S_N(r_1) \\
w_2 &= (1-w_1)S_{N-1}(r_2) \\
w_3 &= (1-w_1-w_2)S_{N-2}(r_3) \\
&\vdots \\
w_j &= \left(1-\sum_{i=1}^{j}w_i\right)S_{N-j+1}(r_j) \\
&\vdots \\
w_{N+1} &= \left(1-\sum_{i=1}^{N}w_i\right)
\end{aligned}
\tag{A1}
$$

with

$$
S_j = 1-(1-r)^{\frac{1}{j}} .
\tag{A2}
$$

This sampling leads to the following CDF $F_N$ and PDF $f_N$ for each of the $N+1$ weights $w_i$:

$$
\begin{aligned}
F_N(w_i) &= 1-(1-w_i)^N \\
f_N(w_i) &= N\cdot(1-w_i)^{N-1} .
\end{aligned}
\tag{A3}
$$
$$
\tag{A4}
$$

Python and R implementations of the sampling algorithm of Eq. A1 are freely available on https://github.com/julemai/PieShareDistribution.





**Appendix B: Analytically-derived Sobol' Sensitivities of Shared-Parameter Benchmark Model**

The main Sobol' indexes $S$ and total Sobol' indexes $ST$ can be calculated analytically for the shared-parameter benchmark
model (Eq. 2, Eq. 10-16, Eq. 18, and Figure 1B). There are 4 sets of indexes depending on which sensitivity metric should be
answered:

A. Conditional parameter sensitivity $\{S_{x_1}^n, S_{x_2}^n, \dots S_{x_N}^n, ST_{x_1}^n, ST_{x_2}^n, \dots ST_{x_N}^n\}$ (Table B1)
   (traditional approach; dependent on model choice)
   *Is parameter $x_1$ of one model more important than model parameter $x_2$ of another model?*

B. Unconditional parameter sensitivity $\{S_{x_1}, S_{x_2}, \dots S_{x_N}, S_{r_i}, ST_{x_1}, ST_{x_2}, \dots ST_{x_N}, ST_{r_i}\}$ (Eq. B1)
   (proposed approach; independent of model option choice)
   *Is model parameter $x_1$ or $x_2$ more influential?*

C. Process option sensitivity $\{S_{D_1}, S_{D_2}, \dots S_{F_2}, S_{r_i}, ST_{D_1}, ST_{D_2}, \dots ST_{F_2}, ST_{r_i}\}$ (Eq. B2)
   (proposed approach)
*Is the process option $D_1$ or $D_2$ more sensitive?*

D. Process sensitivity $\{S_D, S_E, S_F, ST_D, ST_E, ST_F\}$ (Eq. B3)
   (proposed approach)
   *Is process $D$ or $E$ more important?*

The analytically-derived sensitivity indexes of sensitivity metric A are given in Table B1 using Ishigami-Homma parameters
$a = 2.0$ and $b = 1.0$. The values are given for each of the 12 models that can be built using the different process options of the
artificial benchmark model.

The analytically-derived results of the overall parameter sensitivities (independent of the model options chosen), the sensi-
tivities of process options, and sensitivities of processes are listed in Eq. B1, Eq. B2, and Eq. B3, respectively. The parameters
$r_i$ therein are the random variables required to derive the weights $w_i$ of the process options according to Eq. 18 using the
sampling strategy described in Appendix A. $r_1$ is used to derive the weights $w_{d1}$ and $w_{d2}$, $r_2$ and $r_3$ are used to derive the





weights $w_{e1}$, $w_{e2}$ and $w_{e3}$, and $r_4$ is used to derive the weights $w_{f1}$ and $w_{f2}$.

$$
\begin{bmatrix} S_{x_1} \\ S_{x_2} \\ S_{x_3} \\ S_{x_4} \\ S_{x_5} \\ S_{x_6} \\ S_{x_7} \end{bmatrix} = \begin{bmatrix} 0.02298 \\ 0.38064 \\ 0.00222 \\ 0.00023 \\ 0.00023 \\ 0.00003 \\ 0.03928 \end{bmatrix}, \begin{bmatrix} S_{r_1} \\ S_{r_2} \\ S_{r_3} \\ S_{r_4} \end{bmatrix} = \begin{bmatrix} 0.05163 \\ 0.05770 \\ 0.00043 \\ 0.01006 \end{bmatrix}, \begin{bmatrix} ST_{x_1} \\ ST_{x_2} \\ ST_{x_3} \\ ST_{x_4} \\ ST_{x_5} \\ ST_{x_6} \\ ST_{x_7} \end{bmatrix} = \begin{bmatrix} 0.07533 \\ 0.77089 \\ 0.00450 \\ 0.00103 \\ 0.00103 \\ 0.00004 \\ 0.05237 \end{bmatrix}, \begin{bmatrix} ST_{r_1} \\ ST_{r_2} \\ ST_{r_3} \\ ST_{r_4} \end{bmatrix} = \begin{bmatrix} 0.26086 \\ 0.31784 \\ 0.00264 \\ 0.02333 \end{bmatrix} \tag{B1}
$$

$$
\begin{bmatrix} S_{D_1} \\ S_{D_2} \\ S_{E_1} \\ S_{E_2} \\ S_{E_3} \\ S_{F_1} \\ S_{F_2} \end{bmatrix} = \begin{bmatrix} 0.02298 \\ 0.42889 \\ 0.38064 \\ 0.00222 \\ 0.00046 \\ 0.00003 \\ 0.04149 \end{bmatrix}, \begin{bmatrix} S_{r_1} \\ S_{r_2} \\ S_{r_3} \\ S_{r_4} \end{bmatrix} = \begin{bmatrix} 0.05163 \\ 0.05770 \\ 0.00043 \\ 0.01006 \end{bmatrix}, \begin{bmatrix} ST_{D_1} \\ ST_{D_2} \\ ST_{E_1} \\ ST_{E_2} \\ ST_{E_3} \\ ST_{F_1} \\ ST_{F_2} \end{bmatrix} = \begin{bmatrix} 0.07533 \\ 0.80441 \\ 0.77089 \\ 0.00450 \\ 0.00207 \\ 0.00004 \\ 0.05687 \end{bmatrix}, \begin{bmatrix} ST_{r_1} \\ ST_{r_2} \\ ST_{r_3} \\ ST_{r_4} \end{bmatrix} = \begin{bmatrix} 0.26086 \\ 0.31784 \\ 0.00264 \\ 0.02333 \end{bmatrix} \tag{B2}
$$

$$
\begin{bmatrix} S_D \\ S_E \\ S_F \end{bmatrix} = \begin{bmatrix} 0.61237 \\ 0.60821 \\ 0.06485 \end{bmatrix}, \begin{bmatrix} ST_D \\ ST_E \\ ST_F \end{bmatrix} = \begin{bmatrix} 0.87597 \\ 0.86055 \\ 0.06697 \end{bmatrix} \tag{B3}
$$





**Table B1.** Analytically-derived Sobol' indexes $S_i$ and $ST_i$ of the shared-parameter benchmark model with 3 processes $D$, $E$, and $F$ that allow for multiple process descriptions (Eq. 2 and Eq.s 10-16). For example, process $D$ has two options $D_1$ and $D_2$. The model output $f(\boldsymbol{x})$ is assumed to be $f(\boldsymbol{x}) = D \cdot E + F$ to mimic additive and multiplicative model structures. The model $f(\boldsymbol{x}) = D_1 \cdot E_1 + F_1$ corresponds to the Ishigami-Homma function (Ishigami and Homma, 1990). The parameters $a$ and $b$ of the Ishigami-Homma function are set to 2.0 and 1.0, respectively. A dash $(-)$ in the table indicates that the parameter is not active in the according model.

| n | Model | | | $S_{x_1}^n$ | $S_{x_2}^n$ | $S_{x_3}^n$ | $S_{x_4}^n$ | $S_{x_5}^n$ | $S_{x_6}^n$ | $S_{x_7}^n$ | $ST_{x_1}^n$ | $ST_{x_2}^n$ | $ST_{x_3}^n$ | $ST_{x_4}^n$ | $ST_{x_5}^n$ | $ST_{x_6}^n$ | $ST_{x_7}^n$ |
|---|---|---|---|---|---|---|---|---|---|---|---|---|---|---|---|---|---|
| 1 | $D_1$ | $E_1$ | $F_1$ | 0.383 | 0.000 | – | – | – | 0.001 | – | 0.999 | 0.616 | – | – | – | 0.001 | – |
| 2 | $D_1$ | $E_1$ | $F_2$ | 0.171 | 0.000 | 0.007 | – | – | – | 0.548 | 0.445 | 0.274 | 0.007 | – | – | – | 0.548 |
| 3 | $D_1$ | $E_2$ | $F_1$ | 0.656 | – | 0.000 | – | – | 0.036 | – | 0.964 | – | 0.309 | – | – | 0.036 | – |
| 4 | $D_1$ | $E_2$ | $F_2$ | 0.013 | – | 0.012 | – | – | – | 0.968 | 0.019 | – | 0.019 | – | – | – | 0.968 |
| 5 | $D_1$ | $E_3$ | $F_1$ | 0.000 | – | – | 0.000 | 0.000 | 0.132 | – | 0.868 | – | – | 0.434 | 0.434 | 0.132 | – |
| 6 | $D_1$ | $E_3$ | $F_2$ | 0.000 | – | 0.013 | 0.000 | 0.000 | – | 0.983 | 0.005 | – | 0.013 | 0.002 | 0.002 | – | 0.983 |
| 7 | $D_2$ | $E_1$ | $F_1$ | 0.024 | 0.937 | – | – | – | 0.000 | – | 0.063 | 0.976 | – | – | – | 0.000 | – |
| 8 | $D_2$ | $E_1$ | $F_2$ | 0.024 | 0.926 | 0.000 | – | – | – | 0.012 | 0.062 | 0.964 | 0.000 | – | – | – | 0.012 |
| 9 | $D_2$ | $E_2$ | $F_1$ | 0.145 | 0.382 | 0.224 | – | – | 0.001 | – | 0.213 | 0.561 | 0.472 | – | – | 0.001 | – |
| 10 | $D_2$ | $E_2$ | $F_2$ | 0.052 | 0.138 | 0.138 | – | – | – | 0.583 | 0.077 | 0.202 | 0.227 | – | – | – | 0.583 |
| 11 | $D_2$ | $E_3$ | $F_1$ | 0.000 | 0.000 | – | 0.237 | 0.237 | 0.003 | – | 0.144 | 0.379 | – | 0.498 | 0.498 | 0.003 | – |
| 12 | $D_2$ | $E_3$ | $F_2$ | 0.000 | 0.000 | 0.010 | 0.043 | 0.043 | – | 0.810 | 0.026 | 0.068 | 0.010 | 0.090 | 0.090 | – | 0.810 |





**Appendix C: Details on Raven Process Options and Parameters**

The Raven hydrologic modeling framework (Craig et al., 2020) has been employed for this study. We used the released version 3.0 of Raven. The process options $M_1$, $M_2$, …, $X_1$ selected for this study are listed in Table C1). The model parameters active in the individual process options are given in that table as well. In total 35 model parameters are used in at least one of the model options. The valid ranges and parameter descriptions are given in Table C2. Further details about the process option

implementation and the parameters can be found in the Raven documentation (Craig, 2019).





**Table C1.** Processes and process options used for the Raven setup. In total $3 \times 3 \times 2 \times 2 \times 3 \times 1 \times 1 \times 1 \times 1 \times 1 \times 1 \times 1 = 108$ models are possible. The first option of each process $M_1$, $N_1$, $O_1$, $P_1$, $Q_1$, $R_1$, $S_1$, $T_1$, $U_1$, $V_1$, $W_1$, and $X_1$ resemble the HMETS model if parameter $x_35$ is set to zero. All other combinations are artificial models. All process options, however, are used in different hydrologic models. The model parameters active in each option are listed as well. The ranges and a description of the parameters can be found in Table C2.

| Process | Process option | | Parameters active |
|---|---|---|---|
| *Processes with multiple options:* | | | |
| Infiltration | $M_1$ | INF_HMETS | $\{x_1, x_{29}\}$ |
| " | $M_2$ | INF_VIC_ARNO | $\{x_2, x_{29}\}$ |
| " | $M_3$ | INF_HBV | $\{x_3, x_{29}\}$ |
| Quickflow | $N_1$ | BASE_LINEAR_ANALYTIC | $\{x_4, x_{29}\}$ |
| " | $N_2$ | BASE_VIC | $\{x_5, x_6, x_{29}\}$ |
| " | $N_3$ | BASE_TOPMODEL | $\{x_5, x_6, x_7, x_{29}\}$ |
| Soil evaporation | $O_1$ | SOILEVAP_ALL | $\{x_8, x_{29}\}$ |
| " | $O_2$ | SOILEVAP_TOPMODEL | $\{x_8, x_9, x_{10}, x_{29}\}$ |
| Baseflow | $P_1$ | BASE_LINEAR_ANALYTIC | $\{x_{11}\}$ |
| " | $P_2$ | BASE_POWER_LAW | $\{x_{11}, x_{12}\}$ |
| Snow balance | $Q_1$ | SNOBAL_HMETS | $\{x_{13}, \ldots, x_{18}\}$ |
| " | $Q_2$ | SNOBAL_SIMPLE_MELT | – |
| " | $Q_3$ | SNOBAL_HBV | $\{x_{18}, x_{19}\}$ |
| *Processes with single option:* | | | |
| Convolution (surface runoff) | $R_1$ | CONVOL_GAMMA | $\{x_{20}, x_{21}\}$ |
| Convolution (delayed runoff) | $S_1$ | CONVOL_GAMMA_2 | $\{x_{22}, x_{23}\}$ |
| Potential melt | $T_1$ | POTMELT_HMETS | $\{x_{24}, x_{25}, x_{26}, x_{27}\}$ |
| Percolation | $U_1$ | PERC_LINEAR | $\{x_{28}, x_{29}, x_{30}, x_{35}\}$ |
| Rain-snow partitioning | $V_1$ | RAINSNOW_HBV | $\{x_{31}, x_{32}\}$ |
| Precipitation correction | $W_1$ | RAINSNOW_CORRECTION | $\{x_{33}, x_{34}\}$ |
| *Processes with single option but no tunable parameter combined to process $V_1$:* | | | |
| Extraterr. Shortwave Gener. | $X_1$ | SW_RAD_NONE | – |
| Snow-rain partitioning | $X_1$ | RAINSNOW_DATA | – |
| Potential evapotranspiration | $X_1$ | PET_OUDIN | – |
| In-catchment routing | $X_1$ | ROUTE_DUMP | – |
| In-channel routing | $X_1$ | ROUTE_NONE | – |





Table C2: The model parameters $x_i$ used for the Raven setup. The parameters are uniformly distributed in the range given. The process option shows where the corresponding parameter is active. The Raven table and parameter name can be used to locate the parameter in the Raven setup files. A three-layer soil model was used here with the third (groundwater) layer being of infinite depth. The TOPSOIL is the upper soil layer while PHREATIC is the lower soil layer. The three Raven parameters FIELD_CAPACITY TOPSOIL, SNOW_SWI_MAX, and MAX_MELT_FACTOR are derived using a sampled parameter ($x_{10}$, $x_{14}$, and $x_{25}$) and SAT_WILT TOPSOIL, SNOW_SWI_MIN, and MIN_MELT_FACTOR, respectively, to make sure that one parameter is always larger than the other. The baseflow coefficients BASEFLOW_COEFF TOPSOIL and PHREATIC are derived from parameters $x_4$ and $x_{11}$ to allow for a logarithmic sampling.

| Param. | Range | Unit | Proc. Opt. | Raven table | Parameter name |
|---|---|---|---|---|---|
| *Infiltration:* | | | | | |
| $x_1$ | $[0.0, 1.0]$ | - | $M_1$ | LandUseParameterList | HMETS_RUNOFF_COEFF |
| $x_2$ | $[0.1, 3.0]$ | - | $M_2$ | SoilParameterList | B_EXP TOPSOIL |
| $x_3$ | $[0.5, 3.0]$ | - | $M_3$ | SoilParameterList | HBV_BETA TOPSOIL |
| *Quickflow:* | | | | | |
| $x_4$ | $[-5.0, -1.0]$ | 1/d | $N_1$ | SoilParameterList | BASEFLOW_COEFF TOPSOIL = $10.0^{x_4}$ |
| $x_5$ | $[0.0, 100.0]$ | mm/d | $N_2, N_3$ | SoilParameterList | MAX_BASEFLOW_RATE TOPSOIL |
| $x_6$ | $[0.5, 2.0]$ | - | $N_2, N_3$ | SoilParameterList | BASEFLOW_N TOPSOIL |
| $x_7$ | $[5.0, 10.0]$ | m | $N_3$ | TerrainClasses | TOPMODEL_LAMBDA |
| *Evaporation:* | | | | | |
| $x_8$ | $[0.0, 3.0]$ | - | $O_1, O_2$ | SoilParameterList | PET_CORRECTION TOPSOIL |
| $x_9$ | $[0.0, 0.05]$ | frac | $O_2$ | SoilParameterList | SAT_WILT TOPSOIL |
| $x_{10}$ | $[0.0, 0.45]$ | frac | $O_2$ | SoilParameterList | FIELD_CAPACITY TOPSOIL = SAT_WILT TOPSOIL + $x_{10}$ |
| *Baseflow:* | | | | | |
| $x_{11}$ | $[-5.0, -2.0]$ | 1/d | $P_1, P_2$ | SoilParameterList | BASEFLOW_COEFF PHREATIC = $10.0^{x_{11}}$ |
| $x_{12}$ | $[0.5, 2.0]$ | - | $P_2$ | SoilParameterList | BASEFLOW_N PHREATIC |
| *Snow balance:* | | | | | |
| $x_{13}$ | $[0.0, 0.1]$ | frac | $Q_1$ | GlobalParameter | SNOW_SWI_MIN |
| $x_{14}$ | $[0.01, 0.3]$ | frac | $Q_1$ | GlobalParameter | SNOW_SWI_MAX = SNOW_SWI_MIN + $x_{14}$ |
| $x_{15}$ | $[0.005, 0.1]$ | 1/mm | $Q_1$ | GlobalParameter | SWI_REDUCT_COEFF |
| $x_{16}$ | $[-5.0, 2.0]$ | °C | $Q_1$ | LandUseParameterList | DD_REFREEZE_TEMP |
| $x_{17}$ | $[0.0, 1.0]$ | - | $Q_1$ | LandUseParameterList | REFREEZE_EXP |





Table C2 – *Continued from previous page*

| Param. | Range | Unit | Proc. Opt. | Raven table | Parameter name |
|---|---|---|---|---|---|
| $x_{18}$ | $[0.0, 5.0]$ | mm/d/°C | $Q_1, Q_3$ | LandUseParameterList | REFREEZE_FACTOR |
| $x_{19}$ | $[0.0, 0.4]$ | frac | $Q_3$ | GlobalParameter | SNOW_SWI |
| *Convolution (surface runoff):* | | | | | |
| $x_{20}$ | $[0.3, 20.0]$ | - | $R_1$ | LandUseParameterList | GAMMA_SHAPE |
| $x_{21}$ | $[0.01, 5.0]$ | - | $R_1$ | LandUseParameterList | GAMMA_SCALE |
| *Convolution (delayed runoff):* | | | | | |
| $x_{22}$ | $[0.5, 13.0]$ | - | $S_1$ | LandUseParameterList | GAMMA_SHAPE2 |
| $x_{23}$ | $[0.15, 1.5]$ | - | $S_1$ | LandUseParameterList | GAMMA_SCALE2 |
| *Potential melt:* | | | | | |
| $x_{24}$ | $[1.5, 3.0]$ | mm/d/°C | $T_1$ | LandUseParameterList | MIN_MELT_FACTOR |
| $x_{25}$ | $[0.0, 5.0]$ | mm/d/°C | $T_1$ | LandUseParameterList | MAX_MELT_FACTOR = MIN_MELT_FACTOR + $x_{25}$ |
| $x_{26}$ | $[-1.0, 1.0]$ | °C | $T_1$ | LandUseParameterList | DD_MELT_TEMP |
| $x_{27}$ | $[0.01, 0.2]$ | 1/mm | $T_1$ | LandUseParameterList | DD_AGGRADATION |
| *Percolation:* | | | | | |
| $x_{28}$ | $[0.00001, 0.02]$ | 1/d | $U_1$ | SoilParameterList | PERC_COEFF TOPSOIL |
| $x_{35}$ | $[0.0, 0.02]$ | 1/d | $U_1$ | SoilParameterList | PERC_COEFF PHREATIC |
| *Rain-snow partitioning:* | | | | | |
| $x_{31}$ | $[-3.0, 3.0]$ | °C | $V_1$ | GlobalParameter | RAINSNOW_TEMP |
| $x_{32}$ | $[0.5, 4.0]$ | °C | $V_1$ | GlobalParameter | RAINSNOW_DELTA |
| *Precipitation correction:* | | | | | |
| $x_{33}$ | $[0.8, 1.2]$ | - | $W_1$ | Gauge | RAINCORRECTION |
| $x_{34}$ | $[0.8, 1.2]$ | - | $W_1$ | Gauge | SNOWCORRECTION |
| *Soil model:* | | | | | |
| $x_{29}$ | $[0.0, 0.5]$ | m | $M_{1,2,3}, N_{1,2,3}$ $O_{1,2,3,4}, U_1$ | SoilProfiles | thickness TOPSOIL |
| $x_{30}$ | $[0.0, 2.0]$ | m | $U_1$ | SoilProfiles | thickness PHREATIC |





*Author contributions.* The Extended Sobol' Sensitivity Analysis (xSSA) analyzes model parameters, process options and processes. The latter two are computed using grouped parameters. The computationally frugal method hence analyzes model structure based on weighted process options rather than discrete sets of models. It is successfully tested against known benchmarks and on Raven hydrologic modeling

framework.

*Competing interests.* The authors declare that they have no conflict of interest.

*Acknowledgements.* This research was undertaken thanks in part to funding from the CANARIE research software funding program (project RS-332). The work was made possible by the facilities of the Shared Hierarchical Academic Research Computing Network (SHARCNET; www.sharcnet.ca) and Compute/Calcul Canada. All codes, examples, and data used for this study can be found on GitHub (https://github.

com/julemai/xSSA) upon publication.



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
