# Peer review of "Simultaneously Determining Global Sensitivities of Model Parameters and Model Structure"

_Hydrology and Earth System Sciences, 2020_

## Referee Comment (RC1) · Anonymous Referee #1 · 3 Jul 2020

Summary The authors introduce a novel sensitivity analysis method, called Extended Sobol' Sensitivity Analysis (xSSA), that advances upon existing procedures in several ways: (1) it can provide insight into the sensitivity of individual model structure choices; (2) it can clarify the relation between parameter and structure sensitivity; (3) it can account for cases where model parameters are present or absent in different model structures; and (4) it is much faster than alternative methods. The main novelty of xSSA is that it estimates parameter/process sensitivity inside a flexible modelling framework (Raven), which allows the sensitivity estimates to be re-combined through weighting. On any given timestep, thesimualted states and fluxes can thus be based on multiple different parametrizations of the same process, depending on how the weights are set.

The authors test xSSA against two cases where analytical estimates of sensitivities can be derived (one case where each parameter only occurs once in all possible flux parametrizations, and one case where parameters are shared between multiple flux parametrizations), and in a real-world application of the Raven framework in a single watershed. They find that xSSA converge to the analytical solutions in both test cases, while current methods are only able to converge in the first test case. The real-world test case is used to showcase why process-based SA can be useful. I've read this paper with great interest. Model structure uncertainty is receiving considerable attention and this extension of existing SA methods to take advantage of modern multi-model frameworks is a welcome and timely contribution. Overall, the paper is easy to read but I have outlined various comments that can help the authors clarify their message. In general I think all the required information is there but some polishing would make the manuscript much more accessible for readers who are not so well-versed in Sobol' SA and Raven as the authors are.

General comments

The results section relies heavily on understanding of the Raven functions. It would be very helpful if the authors expand on the model description in section 2.1.2 or appendix C, by including the actual equations or descriptions of each parametrization.

The results section relies on an understanding of each process to interpret model sensitivities. It is not entirely clear to me what each process includes. Can the authors clarify this by briefly explaining what each process in Figure 1C and further figures includes? For example, how does process 8 (potential melt) relate to process 5 (snow balance)? I don't think these explanations need to be very long, but it would be good if they include a bit more detail then the 1-3 words they currently get.

I would encourage the authors to be careful with words such as "appropriate" and "important" in the manuscript. To some of the community, "appropriate process representations" might mean "process representations that are an accurate mathematical

description of the real-world". To the authors (I believe) this instead means "equally sensitive, so equally good choices" (e.g. L538). Similarly, "important" seems synonymous with "high sensitivity" in this manuscript, but I don't think having high sensitivity over an arbitrarily wide parameter range necessarily dictates importance for matching a specific set of observations. Therefore I would strongly recommend the authors to go through the manuscript and either define such words clearly, or avoid ambiguity by being more specific in each case where such words are used (e.g. change "soil and surface processes are of secondary importance for streamflow prediction, . . ." to "simulations are less sensitive to soil and surface processes, . . ."; L556).

Line by line comments

L32. It might be more accurate to refer to "input (forcing) uncertainty" as "data uncertainty" or "observational uncertainty" to acknowledge that uncertainties are also present in model evaluation data such as streamflow observations. See e.g. McMillan et al. (2012, 10.1002/hyp.9384).

L47. It would be helpful to the reader if the authors could summarize the Baroni method in one or two sentences.

L53. "The method introduced . . ." some text is missing here.

L94. This special Raven property is a bit unclear to me. What dimension are the simulated fluxes weighted over? Is this a weighted average across multiple parameter sets, model structures, something else? – If this property is critical to the functioning of xSSA I think it should be explained in more detail here. Perhaps an example can be added.

L103. I appreciate what the authors are going for, but "unconditional parameter sensitivity" is too broad a statement. The answers to questions A-D will be conditional on the catchment(s) being considered. It would be good to acknowledge that somewhere in lines 99-103.

L205. I'm somewhat confused about this statement. One does not need to run 12 fixed model structures but instead needs to run a single flexible structure that contains all the options that are present in the 12 models. How does this reduce the number of computations required? As far as I understand, it's still the same elements that are being tested. If the authors mean that all elements can be tested independently (implying that if and how they are connected to other elements can be ignored), than why would they need to be part of a model structure at all? Why not test each element in isolation and recombine the results through the proposed weighting? This could result in even further computational savings in cases where the same parametrization can be used in multiple processes (quite common in bucket models, possibly also in physics-based models that discretize snow/soil into multiple layers).

L212. Caption of Figure 1. "The three processes are connected through A.B+C (C.D+E) ... " Text in the brackets should read (D.E+F).

L212. Caption of Figure 1."Processes A (D) and C (E) ... " (E) should be (F).

L213. Which numerical scheme does Raven use to solve its model equations?

L262. "forcings" > "forcing"?

L269. Why were only 20 years of data used if 56 are available? Wouldn't more data give a more complete assessment because a wider range of conditions is (likely) covered?

L308. It took me a while to figure out that these numbers are: # of models x (# of parameters +2 x K), mainly because the order of operations is reversed compared to L307 (which gives # of parameters first and # of models second) and because the operation K x (N+2) from L303 has already been completed. I'd suggest to clarify this.

L350. The authors use analytically derived Sobol' scores for their shared-parameter model setup. Can these derivations be made part of the appendices or can the authors provide a reference to a paper that provides these?

L358. Single-sentence paragraphs look strange. Suggest to merge with preceding paragraph.

L364, L366. I was under the impression that the shared-parameter models was being tested. Why do these sentences refer to parametrizations A, B and C instead of D, E and F?

L429-441. I find this section difficult to follow, in part because it was not clear to me that the Baroni method uses a regular Sobol' approach. The only mentions of Sobol' so far (I believe) have been in relation to xSSA and the mention of Sobol' analysis on L434 threw me off. I'll repeat my earlier comment that a brief description of the Baroni method would be very helpful in understanding these results.

L435. "This contradiction cannot be resolved." Is it part of the Baroni method to include a single parameter twice? In my (admittedly limited) experience with the regular Sobol' method, one would include any parameter only once, regardless of how many times it occurs in the model processes being considered. This would mean that processes cannot be assessed individually if they share a parameter (which the authors already mention) but getting into this situation in the first place requires that one is looking to investigate processes, not parameters. I think the authors can make their reasoning stronger by repeating here that investigating process sensitivity requires a different approach then parameter sensitivity in cases where parameters are shared between processes.

L451. It might be good to add a reference to sensitivities of non-additive models not summing to 1. I seem to recall this is discussed in Saltelli et al. (2008) for example.

L461. "hence" > "this"?

L499. Suggest to delete "and hence most sensitive"

L509. It might be instructive to adapt the x-axis in Figure 5B, so that it shows which parameters (x-axis in 5A) are included in each process option in 5B. This could clarify

whether process sensitivities can be traced back to particular parameters.

L515. "Same" > "The same"

Figure 5. It might be worthwhile to change the orientation of these plots so that the Sobol' scores are on the x-axis and the parameters/parametrizations/processes are on the y-axis, so that these are easier to read. I currently need to tilt my head back and forth to read the results in 3.3.2 and compare them to the axes in Figure 5.

L525. "The latter serves as a consistency check of the implementation." Can the authors clarify what they mean here? – upon reading further, it might make sense to swap this sentence with the one immediately after it.

L527. I admit I'm a bit confused that model outputs of process representations do not change in different model runs. Because this process representation is connected to the rest of the model, and there are changes in the contributions of other processes as a result of different parameter values, wouldn't it be expected that the model states change as well, and as a consequence, that the contribution of this particular process to overall simulations changes too? Without knowing with SNOBAL_SIMPLE_MELT actually does, I assume that even if it has a constant melt rate, it is still constrained by snow availability and thus cannot produce a time-invariant flux. I would expect such a case (no parameters in a given process, but influenced by other parameters by virtue of being part of a bigger model) as showing in a 0 Sobol' main effect, but a non-zero Sobol' total effect. Can the authors clarify this?

L538. "The three infiltration options are equally sensitive and hence equally appropriate." Logically, only one or none of these infiltration options is appropriate (in the sense of accurately representing the real world). I also doubt that high sensitivity automatically indicates high appropriateness. I suggest to rephrase this sentence.

L538. "quickflow" > should this be "infiltration"?

L545. Can it be said that rain-snow partitioning is a forcing correction function? It does

not change the water balance, only the phase and thus by extent, the timing of liquid water availability.

L556. "This demonstrates that soil and surface processes are of secondary importance for streamflow prediction, ..." Is this true? As far as I understand, the SA only shows that impact of parameter changes on the variability of the simulations. I don't think relatively low sensitivity automatically indicates low importance for accurate streamflow simulation, because (1) no simulations have been compared to observations; (2) parameters ranges might be wider during this SA than their "real" range of values and thus much of this variability might occur in regions of the model output space that are far away from the observations. I would recommend slightly more careful phrasing, like used in L559.

L677. These are not author contributions.

---

## Short Comment (SC1) · 23 Aug 2020

Dear Juliane, dear Authors,

I have really appreciated that your study has been motivated by, among others, one of my papers (Baroni and Tarantola, 2014). I also think your manuscript can be a nice contribution to the literature, but I leave to the official Reviewers to judge with more specific comments. When reading this preprint, however, I found the need to add this short comment to clarify the terminology. I hope this will also help to strengthen your work.

[Figure]

Sincerely, Gabriele Baroni

Comments

In this study, you have introduced the use of weights that can take on non-integer values to account for model structures in the analysis. As you correctly cited I used, in contrast, discrete integer values in Baroni and Tarantola (2014). However, I find important to underline also here that this "trick" has been previously used. I think that I had properly acknowledged that in my paper and I paste the reference below for sake of clarity:

"The use, at step 5 of the framework, of a discrete scalar factor of the size of the realizations generated, enables us to extend the GSA also to non-scalar sources of uncertainty. This approach was introduced by Crosetto and Tarantola (2001), who proposed the use of a sensitivity analysis of a binary input to 'switch' the uncertainties of a rainfall intensity map on and off at the same rate (i.e. for N/2 runs, the switch is set to off and for the remaining N/2 runs it is set to on), allowing their relative importance to be determined. The same approach was then improved by Lilburne et al. (2003) and Lilburne and Tarantola (2009) who explicitly introduced the discrete uniform distribution associated to the different realizations of each specific source of uncertainty as considered in this framework."

Additional discussion on the use of discrete random variables can be found also in:

Plischke, Elmar, Emanuele Borgonovo, and Curtis L. Smith. "Global Sensitivity Measures from given Data." European Journal of Operational Research 226, no. 3 (May 1, 2013): 536–50. https://doi.org/10.1016/j.ejor.2012.11.047

For this reason, I found misleading to read in your manuscript that you compare your xSSA method with "Baroni method". Instead, I suggest using something like "continuous weights method" vs. "discrete values method". In my opinion this would better describe what you are comparing.

---

## Referee Comment (RC2) · Anonymous Referee #2 · 3 Sep 2020

The manuscript focuses on Sensitivity Analysis (SA) of hydrological models. It introduces a more general version of the well-known Sobol method, designed to operate on groups of parameters instead of on individual parameters.

Overall I enjoyed reading the manuscript - its on a topical area and the methods described are sound. I appreciate this work on mathematical model analysis, and the idea of grouped parameter sensitivity is novel at least in hydrology as far as I know. With multi-model/flexible frameworks such as RAVEN and others, analysis of their sensitivity would benefit from such "grouped" analysis.

I have the following concerns with the current manuscript form:

[Figure]

1. The algorithms are not explained in a sufficiently clear way. For example, for the description of Sobol method on lines 300-307, and the description of the xSSA method on lines 324-330, are in my opinion not sufficient for a paper presenting a mathematical method.

Yes, I could probably translate the description there into a procedure / pseudocode, but: first I would not be quite sure if I got it right, and second I (respectfully) suggest the onus is on the authors to provide such an un-ambigous description. Appendix B is helpful to a degree, but seems to use a different notation to the main text (where are the matrices A and B and Cm?).

2. Terms such "uncertainty, "sensitivity", "influence", "importance" are being used in a pretty loose, seemingly interchangeable way. For example, the paragraph on lines 31-40, which starts with "uncertainty" and then immediately switches to "sensitivity". Then line 104 mentions "sensitive/influential/important" parameters. Are these referring to the same characteristic? Similar confusing usage then carries through later in the manuscript.

I suggest the terminology should be much tighter to avoid confusion. Given the mathematically demanding topic, I would suggest giving clear definitions of the various concepts (with links to existing literature where appropriate), and avoiding the alternation of these terms in the remainder of the presentation. There are useful and interesting ideas on lines 100-115, but these are already using the terms above in a way I found unnecessarily confusing because its not clear which terms are used synonymeously and which are not.

The current literature review is heavily focused on sensitivity analysis - which is appropriate given the topic. But if the connection to uncertainty is to be made, I would say the literature review of the latter is currently rudimentary at best.

3. The aims and key contributions of the study seem to drift over the course of the manuscript/presentation. For example the Introduction is focused on sensitivity analysis (and to some extent uncertainty) - but in the Conclusions the contribution #1 is listed as formulating model ensembles as weighted sums of process options, with Sensitivity Analysis then being contribution #2.

I think the coherence between the introduction / aims and contributions could be improved, so that there is a clearer set of aims, appropriate background given on each aim, and then a clear set of conclusions that match those aims.

A clearer vision of the contributions could also help improve the structure of the manuscript, by putting the important contributions much earlier. This would avoid the multiple forward references to the proposed method and its properties before its actual description is given - e.g., see lines 235-237, which are not really that meaningful before seeing how the xSSA method operates. The new XSS method in Section 2.2.2 comes after several quite detailed sections on models and case studies - and it was not immediately apparent that this is the main advance being presented.

4. Some lack of clarity in how important new concepts are defined

Eg, is the sensitivity to groups of parameters taken as sensitivity to processes? Or is that something different? Please check wording across manuscript.

Line 115-122 - I suggest this summary of findings would work better in Abstract + Conclusions. It would also help being clearer in the wording on the comparisons that are being made. Is "conventional" approach the SSA or the Baroni method?

Line 278, where it is pointed out that a traditional single-parameter SA analysis could produce groupped-sensitivity analysis by aggregating results for individual parameters? In a paper advocating the new "groupped-SA" method - should such comparison receive priority to show the advantages of the new method. The hypothetical scenario where sensitivity is underestimated (line 279) - is this common in practice? As this goes to the motivation for the new method, I think it could receive more attention.

Line 352 "limitations of existing Baroni method" - as this comparison is important in

this paper - would seem preferable to describe the Baroni method in appropriate detail before discussing its limitations.

Line 535: "it can be deduced that the potential melt, the quickflow options BASE_VIC and BASE_TOPMODEL, and the evaporation options are most influential upon modeled streamflow". Here the lack of clarity on what is meant by "influential" can cause confusion to a reader. Especially sensitivity to a specific option for a process (eg, BASE_VIC for quickflow) - normally sensitivity is to a range of possible values for a decision - here it is to a single specific value? I don't quite follow this.

Section 3.3 - nice sections. Would be improved by providing clearer definitions of sensitivity, influential processes, uncertainty, etc (see earlier comment). Current usage is unnecessarily loose and confusing here.

===========

Many these comments focus on presentation , but given the technically demanding nature of the work, a more targetted presentation would make it easier to digest by an interested reader.

Other comments

1. Line 4: "apply" or "develop"?

2. Line 24 - what is "they" referring to? Also what does "non-unique" refer to here? Is this with regard to many models co-existing in the literature? Or non-uniqueness in their inversion when estimating parameters? I think some clarity would be useful here

3. Line 27 - are these decisions always subjective? Surely there exist studies where model decisions are developed according to sensible strategies?

4. "Sensitivity to model structural uncertainty" - I think studies such as McMillan et al 2010, Clark et al 2010 and other have investigated this?

5. "recent" - with references back to 2008 is this still recent?

[Figure]

6. Baroni method - seems an important method in the context of this work. I think it would be helpful to provide the gist of that method at least in an Appendix, in the way that is is applied here.

It is also a little unclear from the abstract that a comparison to this method is made. Eg line 13 "alternative" - if this is Baroni's method - should this be "existing" method? To avoid a confusion the reference algorithm should be clearly described.

7. line 49 - "did not change when moving between model structures" - is this for different hydrological models? or models from across multiple disciplines?

8. line 50 - what are "hyper-parameters"?

9. line 52 - not entirely clear what "form" refers to here. I found the entire sentence a bit confusing when trying to understand exactly what its trying to say

10. line 53 - "the method introduced ..." - is an incomplete sentence?

11. line 55 - "individual" - maybe clarify that the previous study assessed ONLY combined sensitivities? This is not clear from the current wording. And I thought that combined sensitivities are an advance rather than individual sensitivities? So why is that a limitation of the previous work?

12. line 62 - "sensitivité of a model" - is this for model simulations? or model parameters? or both? See comment about making sure the key concepts are clearly defined

13. line 78-79 - "it is therefore ..." - i think these ideas on the utility of SA should be introduced earlier in the presentation, to provide a stronger motivation and a practical context for the work.

14. line 88 - this property "structure can vary continuously" / "weighted average". I found this aspect quite interesting in the work. The statement below that xSSA "is made uniquely possible" to RAVEN - do you mean it can only be used by RAVEN? This seems strange as multi-model ensembles where each model has a weight are

fairly common (e.g., see the "model averaging" literature).

15. line 96 - "uniquely"?

16. line 105 Metric B - very interesting concept. but without some elaboration seems potentially ill-defined. Eg, how do you determinne if a parameter appearing in different model structures is "the same parameter"?

17. line 120: "conventional approach" - is this the Baroni method? If so best to name it. Also it was referred to as "alternative" in the Abstract

18. Section 2 - consider splitting into several sections and place in order of relevance to the contributions of the paper

19. line 145 - see earlier comment - how do you know it the "same" parameter? It seems a relevant discussion point

20. line 169 / eqn 16 - how do you "decide" in a modelling context what is a shared parameter? say is x3 in eq 16 the same as x3 as in eqn 13? Is this considered determined purely by the choice made by the modeler regarding the parameters to calibrate?

21. line 235 - 236 - I think these are discussion points - would work better in Discussion rather than forward references here - at this point of the paper the new method is not described yet!

22. Line 312 - would help clarify here that this is approach is new and introduced in this work. And as mentioned earlier - I think it would benefit from being given more prominence in the paper.

23. line 318 - "depicts"?

24. line 409 - "hereafter called Baroni method" - already said this earlier on line 48 - but still referring to this method by multiple names

25. Appendix A - an extra 1-2 sentences that refer to where in the main text are these weights used would be helpful here

26. Appendix B - I am confused why this seems duplicated in the Intro and the Appendix. If this is new - would seem better somewhere in the Theory and then Discussed, where it can be discussed in appropriate detail.

Figures

1. Figure 2 - the bleue font in panel B is quite hard to read

2. Figure 5 (and others to various extents) - could be more generous with fontsize, as many labels etc are virtually illegible

---

## Author Comment (AC1) · 29 Sep 2020

**Reply to Anonymous Referee #1**

*Review received and published: 3 July 2020*

Dear Reviewer,

Thanks a lot for your thorough review and the valuable suggestions. We will reply below in detail to your comments. Your comments are *italic*; our replies are highlighted **bold**. The **line numbers in red** are referring to the revised draft.

Best regards,
Julie, James, and Bryan

*Summary*

*The authors introduce a novel sensitivity analysis method, called Extended Sobol' Sensitivity Analysis (xSSA), that advances upon existing procedures in several ways:*

1. *it can provide insight into the sensitivity of individual model structure choices;*

2. *it can clarify the relation between parameter and structure sensitivity;*

3. *it can account for cases where model parameters are present or absent in different model structures; and*

4. *it is much faster than alternative methods.*

*The main novelty of xSSA is that it estimates parameter/process sensitivity inside a flexible modeling framework (Raven), which allows the sensitivity estimates to be recombined through weighting. On any given timestep, the simulated states and fluxes*

*can thus be based on multiple different parametrizations of the same process, depending on how the weights are set. The authors test xSSA against two cases where analytical estimates of sensitivities can be derived (one case where each parameter only occurs once in all possible flux parametrizations, and one case where parameters are shared between multiple flux parametrizations), and in a real-world application of the Raven framework in a single watershed. They find that xSSA converge to the analytical solutions in both test cases, while current methods are only able to converge in the first test case. The real-world test case is used to showcase why process-based SA can be useful. I've read this paper with great interest. Model structure uncertainty is receiving considerable attention and this extension of existing SA methods to take advantage of modern multi-model frameworks is a welcome and timely contribution. Overall, the paper is easy to read but I have outlined various comments that can help the authors clarify their message. In general I think all the required information is there but some polishing would make the manuscript much more accessible for readers who are not so well-versed in Sobol' SA and Raven as the authors are.*

**Thanks a lot for your interest and your positive evaluation of our manuscript.**

*General comments*

*The results section relies heavily on understanding of the Raven functions. It would be very helpful if the authors expand on the model description in section 2.1.2 or appendix C, by including the actual equations or descriptions of each parametrization.*

**We agree that this is a very valuable information. The details can all be found in the Raven documentation (Craig, 2020). We copied the according information and provide this now in the Supplementary Material. We decided to not include this in the manuscript or appendix to not artificially inflate the manuscript and distract from the actual message of the work. We attached the Supplementary Material at the end of this response letter for reference.**

*The results section relies on an understanding of each process to interpret model sen-*

*sitivities. It is not entirely clear to me what each process includes. Can the authors clarify this by briefly explaining what each process in Figure 1C and further figures includes? For example, how does process 8 (potential melt) relate to process 5 (snow balance)? I don't think these explanations need to be very long, but it would be good if they include a bit more detail then the 1-3 words they currently get.*

**Thanks for this really good suggestion. As mentioned above, we now include the description of the processes and options used in the Supplementary Material. We also added a flowchart of the model structure used here to the Supplementary Material (Figure S1). The processes and functions are labeled according to the usage in this work (see circled labels such as $M$, $N$, etc.). This should make more clear how the processes are interlinked with each other.**

*I would encourage the authors to be careful with words such as "appropriate" and "important" in the manuscript. To some of the community, "appropriate process representations" might mean "process representations that are an accurate mathematical description of the real-world". To the authors (I believe) this instead means "equally sensitive, so equally good choices" (e.g. L538). Similarly, "important" seems synonymous with "high sensitivity" in this manuscript, but I don't think having high sensitivity over an arbitrarily wide parameter range necessarily dictates importance for matching a specific set of observations. Therefore I would strongly recommend the authors to go through the manuscript and either define such words clearly, or avoid ambiguity by being more specific in each case where such words are used (e.g. change "soil and surface processes are of secondary importance for streamflow prediction, ..." to "simulations are less sensitive to soil and surface processes, ..."; L556).*

**The reviewer is correct in the interpretation of our interchangeable use of "importance", "sensitivity" and appropriateness. We agree that it might lead to too much leeway of interpretation for a reader. We went through the manuscript and hope that we reduced this ambiguity.**

**line 521 ff.** The analysis of model parameters (Fig. 5A) shows that the most sensitive ones are [...]

**line 586 ff.** The strong impact of these processes (together with the input adjustments) highlights the sensitivity of streamflow regarding snow and melting processes in this mountainous, energy-limited catchment.

**line 589 ff.** This demonstrates that soil and surface processes are of secondary sensitivity regarding streamflow. Their sensitivity may increase if the uncertainty of the snow and melting processes can be reduced, i.e. by narrowing parameter ranges during calibration.

**line 610 ff.** Evaporation (dark blue) is [...], expectedly, less sensitive during winter. Snow balance (medium green) and potential melt (orange) are sensitive as long as snow is present (Nov to May).

*Line by line comments*

*L32. It might be more accurate to refer to "input (forcing) uncertainty" as "data uncertainty" or "observational uncertainty" to acknowledge that uncertainties are also present in model evaluation data such as streamflow observations. See e.g. McMillan et al. (2012).*

**We agree. We only focus in this work on the input uncertainty but at this part of the introduction it should be made clear that data uncertainty is the third source of uncertainty. We have made the following adjustment:**

line 32 ff. **Model structural uncertainty is commonly recognized (e.g., Gupta et al., 2012) as one of the three key components of hydrological model uncertainty, along with parameter uncertainty (Evin et al., 2014, among many more) and data (e.g., input forcing or observational) uncertainty (e.g., McMillan et al., 2012).**

*L47. It would be helpful to the reader if the authors could summarize the Baroni method in one or two sentences.*

**We agree. We rewrote major parts of that paragraph in the introduction and hope that the additional details given are now more helpful to follow the line of arguments. Following another reviewer's suggestion we also renamed the "Baroni method" with "discrete values method (DVM)" throughout the whole manuscript.**

line 50 ff. **To date, there have been limited attempts to simultaneously estimate model parameter, input, and structural sensitivities. One notable attempt is introduced by Baroni and Tarantola (2014) using a Sobol' sensitivity analysis based on grouped parameter. In that study, groups of soil and crop parameters, the number of soil layers, and a group of parameters to perturb inputs are investigated. These groups of parameters are pre-sampled**

[Figure]

**and a finite set of parameters for each of the four groups is chosen and each set is enumerated. The sensitivity analysis is then based on those enumerated sets. This means, rather than sampling each individual parameter like in a classic Sobol' analysis, an integer for each group acting as a hyper-parameter is sampled. The model is then run with the associated pre-sampled parameter set. While the approach may be generally applicable to arbitrary structural differences, in their testing, Baroni and Tarantola (2014) varied only in how the model was internally discretized (i.e., in the number of soil layers). The soil and crop parameters were always used for the same soil and crop process. The major limitation of this method is, however, that individual parameters need to be mutually exclusive and can only be associated to one type of uncertainty. The method hence limits the groups that can be defined, for instance, overlapping group definitions are not possible. The method will be referred to as "discrete values method (DVM)" in the following and will be contrasted to the method developed here to examine this limitation in more detail.**

*L53. "The method introduced ..." some text is missing here.*

**We deleted this. It was a remainder of a previous version. We are sorry about that.**

*L94. This special Raven property is a bit unclear to me. What dimension are the simulated fluxes weighted over? Is this a weighted average across multiple parameter sets, model structures, something else? – If this property is critical to the functioning of xSSA I think it should be explained in more detail here. Perhaps an example can be added.*

**The weighted average is for the estimates of the different process options. Let's assume there are three infiltration options. The first derives an infiltration of 1.0 [mm/d], the next 1.5 [mm/d], and the third 2.0 [mm/d]. The model would pro-**

ceed with an estimate for infiltration of 1.35 [mm/d] ($= 1.0 \times 0.5 + 1.5 \times 0.3 + 2.0 \times 0.2$) if the weights are 0.5, 0.3, and 0.2, respectively. This is performed for each process with multiple options at each time step (basically any time infiltration needs to be obtained during the simulation).

We added the following additional explanation and hope this is more clear now:

line 105 ff. [...] may be calculated via the weighted average of simulated fluxes generated by individual process algorithms; other flexible models may be revised to accommodate such analysis. The weighted averaging means that at each time step each option chosen for a process would derive an estimate for the flux, in [mm/d], and the weighted average of these estimates would be used for the next step.

*L103. I appreciate what the authors are going for, but "unconditional parameter sensitivity" is too broad a statement. The answers to questions A-D will be conditional on the catchment(s) being considered. It would be good to acknowledge that somewhere in lines 99-103.*

Absolutely. We made the following adjustments and hope that it is now more clear that we indeed mean only "unconditional" regarding model structure and nothing beyond this.

line 111 ff. The xSSA method allows us to efficiently estimate not only the global sensitivity of model parameters independent and hence unconditional of the chosen model structure [...]

> [A.]Unconditional parameter sensitivity: *Which model parameter is most influential independent and hence unconditional of model option choice?*
> For example, which model parameter is overall the most influential

**given all possible model structures (available in the modeling frame-work)?**

*L205. I'm somewhat confused about this statement. One does not need to run 12 fixed model structures but instead needs to run a single flexible structure that contains all the options that are present in the 12 models. How does this reduce the number of computations required? [...]*

**The runtime of running the 12 models independently would be only the same as the runtime of the single flexible structure if the time it takes to read inputs, to initialize the model, and to write model outputs would be negligible. This is certainly true for the two benchmark models but is not the case in most hydrologic and land-surface models. Most of these models do, for example, usually not allow the users to reduce the amount of model outputs written. Raven is highly optimized regarding I/O and initialization. The runtimes for three individual models of the 108 Raven models are 51.786s ($M_1 - N_1 - O_1 - P_1 - Q_1$), 52.695s ($M_2 - N_1 - O_1 - P_1 - Q_3$), and 51.985s ($M_2 - N_1 - O_1 - P_2 - Q_3$) each for 100 runs while the runtime of the single model with the flexible structure is 53.342s for 100 runs. This yields runtime savings of about 99% for using the flexible model structure with weights over running the individual models:**

$$\left[1 - \frac{53.342}{(51.786 + 52.695 + 51.985)/3 \times 108}\right] \times 100\% = 99.05\%$$

*[...] As far as I understand, it's still the same elements that are being tested. If the authors mean that all elements can be tested independently (implying that if and how they are connected to other elements can be ignored), than why would they need to be part of a model structure at all? Why not test each element in isolation and recombine the results through the proposed weighting? This could result in even further*

*computational savings in cases where the same parametrization can be used in multiple processes (quite common in bucket models, possibly also in physics-based models that discretize snow/soil into multiple layers).*

**This is true but we think it will be pretty unlikely to beat the runtime improvement of 99% as shown above with the approach suggested by the reviewer.**

*L212. Caption of Figure 1. "The three processes are connected through A.B+C (C.D+E) ..." Text in the brackets should read (D.E+F).*
*L212. Caption of Figure 1. "Processes A (D) and C (E) ..." (E) should be (F).*

**Thanks for spotting this. This is resolved now.**

**Figure 1 caption.** **The three processes are connected through** $A \cdot B + C$ **(**$D \cdot E + F$**)**
   **to obtain the hypothetical model outputs. Processes** $A$ **(**$D$**) and** $C$ **(**$F$**) have**
   **two options, process** $B$ **(**$E$**) has three.**

*L213. Which numerical scheme does Raven use to solve its model equations?*

**We would like to refer to the publication that introduced Raven (Craig et al., 2020) (end of Section 3.2 therein) for details on all numerical schemes supported in Raven. The default is the Ordered Series approach which was used here. Besides that Raven supports the explicit Euler and iterative predictor–corrector method for solving a set of ODEs (Snowdon, 2010). Additional details can also be found in the Raven manual (Craig, 2020). We have added the following information to the manuscript:**

**line 259 ff.** **For the case study used herein, Raven is applied in lumped mode and**
   **the models are solved using the ordered series numerical scheme defined**
   **in Craig et al. (2020, (end of Section 3.2 therein)).**

*L262. "forcings" → "forcing"?*

**Done.**

*L269. Why were only 20 years of data used if 56 are available? Wouldn't more data give a more complete assessment because a wider range of conditions is (likely) covered?*

**This is probably true. We however think that a 20 years simulation period is already covering a wide range of conditions. The reduction of the simulation period from 56 to 22 years (including the two years used for warm-up) was mainly to reduce the runtime of the whole analysis. The 22-year setup took about 22 hours and would have been 56 hours for the full period. We decided that the gain in results does not justify the longer runtime. A 20-year simulation period is indeed a very long period used for sensitivity studies: Markstrom et al. (2016) used 3 years of warmup and 11 years of simulations, Mendoza et al. (2015) used 2 years of warmup and only 6 years of simulations, and Cuntz et al. (2016) used 16 years of simulations.**

*L308. It took me a while to figure out that these numbers are: # of models x (# of parameters +2 × K), mainly because the order of operations is reversed compared to L307 (which gives # of parameters first and # of models second) and because the operation K × (N+2) from L303 has already been completed. I'd suggest to clarify this.*

**Thanks for pointing us to this. We have made the following adjustment and hope it is easier to follow now.**

line 331 ff. **Out of the 12 possible shared-parameter benchmark models (Eq. 2) there are 4 models that contain 3 parameters, 5 models contain 4 parameters, 2 models consist of 5 parameters, and 1 model has 6 parameters. Hence, 72 000 ($= 4 \times (3+2) \times 1000 + 5 \times (4+2) \times 1000 + 2 \times (5+2) \times 1000 + 1 \times (6+2) \times 1000$) model runs would be required if $K = 1000$ reference parameter sets would be used.**

*L350. The authors use analytically derived Sobol' scores for their shared-parameter model setup. Can these derivations be made part of the appendices or can the authors provide a reference to a paper that provides these?*

**We are deriving these values following the example provided in Saltelli et al. (2008) in example 5 described on page 179 and following. We added this to the manuscript:**

line 386 ff. **All analytically derived indexes are obtained by following the descriptions in Saltelli et al. (2008, page 179 ff).**

*L358. Single-sentence paragraphs look strange. Suggest to merge with preceding paragraph.*

**Done.**

*L364, L366. I was under the impression that the shared-parameter models was being tested. Why do these sentences refer to parametrizations A, B and C instead of D, E and F?*

**We are sorry for that. The reviewer is absolutely right. We adjusted the text to:**

line 392 ff. **[...] model runs are required for the 7 process options $D_1$, $D_2$, ..., $F_2$ and the 4 weight deriving random numbers $r_i$. For the analysis of processes (sensitivity metric D) the model needs to be run $(3+2) \times K$ times to obtain sensitivities of the 3 processes $D$, $E$, and $F$.**

*L429-441. I find this section difficult to follow, in part because it was not clear to me that the Baroni method uses a regular Sobol' approach. The only mentions of Sobol' so far (I believe) have been in relation to xSSA and the mention of Sobol' analysis on L434 threw me off. I'll repeat my earlier comment that a brief description of the Baroni method would be very helpful in understanding these results.*

[Figure]

**Agreed. We hope the revised section in the introduction (line 50 ff.) clarifies this now.**

*L435. "This contradiction cannot be resolved." Is it part of the Baroni method to include a single parameter twice? In my (admittedly limited) experience with the regular Sobol' method, one would include any parameter only once, regardless of how many times it occurs in the model processes being considered. This would mean that processes cannot be assessed individually if they share a parameter (which the authors already mention) but getting into this situation in the first place requires that one is looking to investigate processes, not parameters. I think the authors can make their reasoning stronger by repeating here that investigating process sensitivity requires a different approach then parameter sensitivity in cases where parameters are shared between processes.*

**The point is that the Baroni method (now Discrete Values Method DVM) indeed does not investigate the sensitivity of individual parameters either. We wanted to highlight the difference to this existing method. Most parameters certainly only appear in individual groups but especially when several process options are investigated (Baroni and Tarantola (2014) did not do this) several parameters will appear in several process options. For example, porosity is likely a parameter in each process option related to soil processes. Let's say we have two process options and found that option 1 depends on parameters $x_1$, $x_2$, and $x_3$ while option 2 only depends on two parameters, again on $x_1$ and an new parameter $x_4$.**

| Group 1 | | | Group 2 | |
|---|---|---|---|---|
| $x_1$ | $x_2$ | $x_3$ | $x_1$ | $x_4$ |
| 0.1 | 5.0 | 10.0 | 0.2 | 6.0 |
| 0.2 | 2.0 | 20.0 | 0.4 | 7.0 |
| 0.3 | 3.0 | 15.0 | 0.3 | 8.0 |

**Even without knowing how exactly the Sobol' method works, the problem becomes clear when a value for parameter $x_1$ has to be picked. Is it $0.1$ or $0.2$ for**

**the first set (first row in above table)? This leads in any method to problems; not even only for the Sobol' method.**

**We followed the reviewers advise and emphasized again that a method that is applicable for shared parameters is needed when analyzing the sensitivity of process options and processes.**

line 464 ff. **This contradiction can not be resolved in a method that does not allow for shared parameters. Shared parameters occur often in several process options of the same process but also across processes and hence need to be considered when analyzing process options and processes in flexible frameworks.**

*L451. It might be good to add a reference to sensitivities of non-additive models not summing to 1. I seem to recall this is discussed in Saltelli et al. (2008) for example.*

**Yes, that is a good idea. We added two references there.**

line 481 ff. **We do not expect the process sensitivities to sum up to 1 which is anyway not achievable with non-additive models (Sobol and Kucherenko, 2004; Saltelli et al., 2008).**

*L461. "hence" → "this"?*

**Done.**

line 493 ff. **The errors converge to zero in every analysis and this proves that [...]**

*L499. Suggest to delete "and hence most sensitive"*

**Absolutely! Thanks for spotting this.**

**line 530 ff.** **A sensitivity analysis regarding model parameters is often performed prior to model calibration to identify the most sensitive parameters which are in turn the parameters that [...]**

*L509. It might be instructive to adapt the x-axis in Figure 5B, so that it shows which parameters (x-axis in 5A) are included in each process option in 5B. This could clarify whether process sensitivities can be traced back to particular parameters.*

**We in general agree with the reviewer. The information however is given in Table C1. We do not want to make the figure more complex than it already is. We however added the reference to Table C1 to the figure caption:**

**Figure 5 caption.** **[...] The Sobol' sensitivity index estimates are determined also for (B) 19 process options and (C) the 11 processes. The information which parameters are used in which process option and process can be found in Table C1. [...]**

*L515. "Same" → "The same"*

**Done.**

**line 547 ff.** **The same holds for the two options of the evaporation process (dark blue bars) [...]**

*Figure 5. It might be worthwhile to change the orientation of these plots so that the Sobol' scores are on the x-axis and the parameters/parametrizations/processes are on the y-axis, so that these are easier to read. I currently need to tilt my head back and forth to read the results in 3.3.2 and compare them to the axes in Figure 5.*

**Thanks for this suggestion. We did that and Figure 5 appears now as shown below.**

[Figure]

figure-1.pdf

**Fig. 5.** Results of the Sobol' sensitivity analysis of the hydrologic modeling framework Raven. (A) The sensitivities of 35 model parameters (see Table C2) and 8 parameters $r_i$ that are used to determine the weights of process options are estimated. The Sobol' sensitivity index estimates are determined also for (B) 19 process options and (C) the 11 processes. The different colors indicate the association of parameters and process options to the eleven processes. Parameters $x_{29}$ and $x_{30}$ are associated with several process options and are not colored but gray. The Sobol' main and total effects are shown (dark and light colored bars, respectively). All sensitivity index estimates shown are originally time-dependent and are aggregated as variance-weighted averages (Eq. 23 and 24). The average weights over the course of the year are shown in Figure 6.

*L525. "The latter serves as a consistency check of the implementation." Can the authors clarify what they mean here? – upon reading further, it might make sense to swap this sentence with the one immediately after it.*

**We are sorry that the line of arguments was a bit mixed up here. We rearranged the order and provided a bit more information. It now reads as:**

line 556 ff. **The zero sensitivity is expected since the SNOBAL_SIMPLE_MELT option does not require any parameters (see Tab. C1). Model outputs of such options do not change for different model runs and hence have a zero variance which leads to a zero Sobol' index. Such settings and parameters that are a priori known to yield zero sensitivities are beneficial in sensitivity analyses as they act as a consistency check of the implementation (Mai and Tolson, 2019).**

*L527. I admit I'm a bit confused that model outputs of process representations do not change in different model runs. Because this process representation is connected to the rest of the model, and there are changes in the contributions of other processes as a result of different parameter values, wouldn't it be expected that the model states change as well, and as a consequence, that the contribution of this particular process to overall simulations changes too? [...]*

**The process outputs of other processes might change- especially when a parameter is in the current group that also participates in other groups (e.g., parameter $x_{29}$). That process options, processes and parameters are not independent of other parts of the model can be seen through the interaction effect that is derived by $ST_i - S_i$ of the respective parameter, process option or process. The analysis however shows how much the process impacts the overall model output (here streamflow). Therefore, it is a desirable behavior that also other process outputs might change. But one knows that these differences are only caused by the change of the parameter/process option/process currently analyzed.**

*[...] Without knowing with SNOBAL_SIMPLE_MELT ($Q_2$) actually does, I assume that even if it has a constant melt rate, it is still constrained by snow availability and thus cannot produce a time-invariant flux. I would expect such a case (no parameters in a given process, but influenced by other parameters by virtue of being part of a bigger model) as showing in a 0 Sobol' main effect, but a non-zero Sobol' total effect. Can the authors clarify this?*

**The mentioned snow balance process option does not contain any parameter. The process output of SNOBAL_SIMPLE_MELT over time $t$ is $\mathbf{Q_2(t)} = Q_2(\mathbf{M_{potmelt}})$ where $M_{potmelt}$ is the potential melt at time $t$. The potential melt, i.e., the calculation of available energy at the snow surface, is another process (i.e., $T_1$) because it is used for other options and in other places of the model. In other hydrologic models the snow balance and potential melt are usually not separated but Raven allows the user to mix-and-match different approaches with each other. The snow balance $Q_2$ itself has hence no parameter $x$. The potential melt $M_{potmelt}$ is an input. This means when all parameters associated to SNOBAL_SIMPLE_MELT (means none) are changed, nothing in the model outputs $Q_2(t)$ ever changes because literally nothing is changed. The output of the SNOBAL_SIMPLE_MELT is not constant over time though. It is just does not change. That is the reason for our statement in the comment before that this is a very helpful consistency check for the implementation of the analysis. The interaction effect is zero because none of the independent variables is derived by any of the other processes and process options. The output of $Q_2$ is hence always constant even if other parts of the model are changed.**

*L538. "The three infiltration options are equally sensitive and hence equally appropriate." Logically, only one or none of these infiltration options is appropriate (in the sense of accurately representing the real world). I also doubt that high sensitivity automatically indicates high appropriateness. I suggest to rephrase this sentence.*

**We adjusted the manuscript to the following:**

**line 547 ff.** **The three infiltration options are equally sensitive and hence are all able to achieve the same amount of variability in simulated streamflow time series. This similarity is an indicator that the choice of the infiltration option will therefore not influence the model performance.**

*L538. "quickflow" → should this be "infiltration"?*

**Indeed. Thanks for spotting this. We adjusted this to:**

**line 570 ff.** **The three infiltration options are equally sensitive and hence are all able to achieve the same amount of variability in simulated streamflow time series. This similarity is an indicator that the choice of the infiltration option will therefore not influence the model performance.**

*L545. Can it be said that rain-snow partitioning is a forcing correction function? It does not change the water balance, only the phase and thus by extent, the timing of liquid water availability.*

**Yes, that is correct. We slightly adjusted the phrasing in the manuscript:**

**line 578 ff.** **Technically, potential melt $T$ as well as rain-snow partitioning $V$ and precipitation correction $W$ are handling inputs to the hydrologic system and can hence be regarded to quantify input uncertainties or, in other words, are forcing correction function and do not change the water balance within the model.**

*L556. "This demonstrates that soil and surface processes are of secondary importance for streamflow prediction, ..." Is this true? As far as I understand, the SA only shows that impact of parameter changes on the variability of the simulations. I don't think relatively low sensitivity automatically indicates low importance for accurate streamflow*

*simulation, because (1) no simulations have been compared to observations; (2) parameters ranges might be wider during this SA than their "real" range of values and thus much of this variability might occur in regions of the model output space that are far away from the observations. I would recommend slightly more careful phrasing, like used in L559.*

**We absolutely agree with the reviewer and rephrased this to:**

**line 589 ff.** **This demonstrates that soil and surface processes are of secondary sensitivity regarding streamflow. Their sensitivity may increase if the uncertainty of the snow and melting processes can be reduced, i.e. by narrowing parameter ranges during calibration.**

*L677. These are not author contributions.*

**The author contributions have been adjusted to the following:**

**line 702 ff.** **JM set up the analyses, implemented the sensitivity analysis based on groups of parameters, implemented the proper sampling of weights used in this study, wrote main parts of the manuscript, prepared all figures and tables; JRC contributed to the writing of the manuscript, implemented the weighting of process options in Raven, provided ranges for the parameters included in the analysis, helped to setup the model with the selected options and resolved inconsistencies in Raven detected by earlier versions of the sensitivity analysis, and helped with the hydrologic interpretation of the results; BAT contributed to the writing of the manuscript, provided feedback on the manuscript and the setup of all experiments including the benchmark models as well as helped with the hydrologic interpretation of the results.**

**References**

Baroni, G. and Tarantola, S.: A General Probabilistic Framework for uncertainty and global sensitivity analysis of deterministic models: A hydrological case study, Environmental Modelling & Software, 51, 26–34, 2014.

Craig, J. R.: Raven: User's and Developer's Manual v3.0, http://raven.uwaterloo.ca/files/v3.0/RavenManual_v3.0.pdf, 2020.

Craig, J. R., Brown, G., Chlumsky, R., Jenkinson, W., Jost, G., Lee, K., Mai, J., Serrer, M., Snowdon, A. P., Sgro, N., Shafii, M., and Tolson, B. A.: Flexible watershed simulation with the Raven hydrological modelling framework, Environmental Modelling & Software, p. 104728, https://doi.org/https://doi.org/10.1016/j.envsoft.2020.104728, 2020.

Cuntz, M., Mai, J., Samaniego, L., Clark, M. P., Wulfmeyer, V., Branch, O., Attinger, S., and Thober, S.: The impact of standard and hard-coded parameters on the hydrologic fluxes in the Noah-MP land surface model , Journal of Geophysical Research: Atmospheres, pp. 1–25, 2016.

Evin, G., Thyer, M., and Kavetski, D.: Comparison of joint versus postprocessor approaches for hydrological uncertainty estimation accounting for error autocorrelation and heteroscedasticity , Water Resources Research, 50, 1–26, 2014.

Gupta, H. V., Clark, M. P., Vrugt, J. A., Abramowitz, G., and Ye, M.: Towards a comprehensive assessment of model structural adequacy, Water Resources Research, 48, https://doi.org/10.1029/2011WR011044, 2012.

Mai, J. and Tolson, B. A.: Model Variable Augmentation (MVA) for Diagnostic Assessment of Sensitivity Analysis Results, Water Resources Research, 55, 2631–2651, 2019.

Markstrom, S. L., Hay, L. E., and Clark, M. P.: Towards simplification of hydrologic modeling: identification of dominant processes, Hydrology and Earth System Sciences, 20, 4655–4671, 2016.

McMillan, H., Krueger, T., and Freer, J.: Benchmarking observational uncertainties for hydrology: rainfall, river discharge and water quality, Hydrological Processes, 26, 4078–4111, 2012.

Mendoza, P. A., Clark, M. P., Barlage, M., Rajagopalan, B., Samaniego, L., Abramowitz, G., and Gupta, H.: Are we unnecessarily constraining the agility of complex process-based models?, Water Resources Research, 51, 716–728, 2015.

Saltelli, A., Ratto, M., Andres, T. H., Campolongo, F., Cariboni, J., Gatelli, D., Saisana, M., and

Tarantola, S.: Global sensitivity analysis. The primer, John Wiley & Sons, Ltd., 2008.

Snowdon, A. P.: Improved numerical methods for distributed hydrological models, Ph.D. thesis, University of Waterloo, Waterloo, Ontario, Canada., 2010.

Sobol, I. M. and Kucherenko, S. S.: Global Sensitivity Indices for Nonlinear Mathematical Models. Review, WILMOTT magazine, pp. 2–7, 2004.

[Figure]

**Supplement:**

**Simultaneously Determining Global Sensitivities of Model Parameters and Model Structure – Supplementary Material –**

Juliane Mai[1], James R. Craig[1], and Bryan A. Tolson[1]

[1]Department Civil and Environmental Engineering, University of Waterloo, 200 University Ave W, Waterloo, ON, N2L 3G1, Canada.

**Correspondence:** Juliane Mai (juliane.mai@uwaterloo.ca)

**S.1 The Raven Model with Weighted Process Options**

The Raven model allows to specify multiple options for each process such as infiltration, evaporation, and baseflow. Instead of the model using one unique parametrization and output for each process it is then deriving the weighted average of outputs for the various options (Craig et al., 2020). This is defined in the main manuscript (Eq. 18) by

$$f_{\text{shared}}(\boldsymbol{x}, \boldsymbol{w}) = (w_{d1}D_1 + w_{d2}D_2) \cdot (w_{e1}E_1 + w_{e2}E_2 + w_{e3}E_3) + (w_{f1}F_1 + w_{f2}F_2) \tag{S1}$$

where

$$
\begin{aligned}
w_{d1} + w_{d2} &= 1 \\
w_{e1} + w_{e2} + w_{e3} &= 1 \\
w_{f1} + w_{f2} &= 1 \;.
\end{aligned}
$$

where, for example, $D_1$ and $D_2$ might be two options for one process. For example, deriving infiltration is performed once using the infiltration definition of HMETS and once derived as defined in the HBV model. The infiltration outputs $D_1$ and $D_2$ are then weighted using $w_{d1}$ and $w_{d_2}$ to derive the infiltration estimate Raven will use for the remainder of the simulation. The overall flowchart of the model given all hydrologic processes involved is given in the flowchart in Fig. S1. In that flowchart the processes labeled with $M$ to $Q$ are used here with multiple options while the processes $R$ to $W$ are fixed with only one option. The processes labeled with $X$ are the ones that are also fixed at one option and this option does not contain tunable parameters. In the following we will explain briefly the process options chosen for this study that will lead to non-zero sensitivities, i.e. processes $M$ to $W$. Most of these description as well as all descriptions for processes $X$ can be found in the Raven documentation (Craig, 2020).

[Figure]

**Figure S1.** The model schematic of the models structure used in this study. The connection of storages (boxes with thick outlines), processes (hexagonal shapes), and forcing functions (diamond shapes) are shown. Some processes are simplified in this schematic (hexagonal shape with dashed outline). The labels used for the processes/functions in this study are indicated by the circled letters right of the processes and forcing functions. The five processes $M$ to $Q$ are used here with multiple options while the processes $R$ to $W$ are fixed with only one option. The processes labeled with $X$ are the ones that are also fixed at one option and this option does not contain tunable parameters. Hence the sensitivity of the processes $X$ is already prior known to be zero. The processes and options as well as the parameters active in each option are listed in Tab. C1 of the main manuscript.

**S.1.1 Infiltration Process $M$**

20   Infiltration refers to the partitioning of ponded water (the residual rainfall and/or snowmelt) between the shallow surface soil (infiltrated water) and surface water (runoff). Infiltration is typically controlled by the saturation of the soil and its hydraulic properties (e.g., hydraulic conductivity, infiltration capacity).

Infiltration always moves water from `PONDED_WATER` to `SOIL[0]` (the top soil layer), and depending upon the soil structure model specified by the `:SoilModel` command, may additionally push water to lower soil moisture stores. The remaining
25   infiltrated water is typically treated as runoff and moved to `SURFACE_WATER`.

Infiltration is limited by the availability of soil/ aquifer storage. Many of the following algorithms use the quantities of maximum soil storage ($\phi_{max}$ [mm]), maximum tension storage ($\phi_{tens}$ [mm]), and field capacity storage ($\phi_{fc}$ [mm]) in a layer, always calculated as:

$$\phi_{max} = Hn(1 - SF) \tag{S2}$$
$$\phi_{tens} = \phi_{max}(S_{fc} - S_{wilt})$$
$$\phi_{fc} = \phi_{max}S_{fc}$$

where $H$ is the soil layer thickness [mm] (in this study parameter $x_{29}$), $n$ is the porosity (soil property `POROSITY`), $SF$ is the stone fraction (soil property `STONE_FRAC`), $S_{fc}$ is the saturation at field capacity (soil parameter `FIELD_CAPACITY`), and $S_{wilt}$ is the saturation at the wilting point (soil parameter `SAT_WILT`).

**S.1.1.1   HMETS infiltration method (`INF_HMETS`) used as option $M_1$**

From the HMETS model (Martel et al., 2017):

$$M_1 = R \cdot \left(1 - \alpha \cdot \frac{\phi_{soil}}{\phi_{soil}^{max}}\right)$$

where $R$ is the rainfall/snowmelt rate [mm/d], $\alpha$ is the unitless land use parameter `HMETS_RUNOFF_COEFF` (in this study parameter $x_1$), $\phi_{soil}$ is the topsoil layer water content, and $\phi_{max}$ is the maximum soil storage [mm] calculated using equation
40   S2.

**S.1.1.2   VIC/ARNO method (`INF_VIC_ARNO`) used as option $M_2$**

The VIC/ARNO model as interpreted by (Clark et al., 2008).

$$M_2 = R \cdot \left(1 - \left(1 - \frac{\phi_{soil}}{\phi_{max}}\right)^b\right)$$

where $R$ is the rainfall/snowmelt rate [mm/d], $b$ is the soil parameter `B_EXP` (in this study parameter $x_2$), $\phi_{soil}$ is the top soil
45   layer water content [mm], and $\phi_{max}$ is the maximum topsoil storage [mm] calculated using equation S2.

**S.1.1.3   HBV method (`INF_HBV`) used as option $M_3$**

The standard HBV model approach (Bergström, 1995).

$$M_3 = R \cdot \left(1 - \left(\frac{\phi_{soil}}{\phi_{max}}\right)^\beta\right)$$

where $\beta$ is the soil parameter `HBV_BETA` (in this study parameter $x_3$), $\phi_{soil}$ is the soil layer water content [mm], and $\phi_{max}$ is
50   the maximum soil storage [mm] calculated using equation S2.

**S.1.1.4 Weighted sum of all options used for infiltration process $M$**

The combined, weighted sum of the three options is used as infiltration estimate in Raven, i.e.

$$M \quad = \quad w_1 M_1 + w_2 M_2 + w_3 M_3$$

The three weights $w_i$ are derived from the two i.i.d. parameters $r_j$ sampled uniform from the unit interval following the approach described in Appendix A of the main manuscript:

$$w_1 \quad = \quad 1 - (1 - r_1)^{\frac{1}{2}}$$
$$w_2 \quad = \quad (1 - w_1) r_2$$
$$w_3 \quad = \quad 1 - w_1 - w_2$$

**S.1.2 Quickflow Process $N$**

**S.1.2.1 Linear storage (`BASE_LINEAR_ANALYTIC`) used as option $N_1$**

A very common approach used in a variety of conceptual models. The baseflow rate is linearly proportional to storage (BASE_LINEAR_STORAGE):

$$N_1 = k \phi_{soil}$$

Where $k$ [1/d] is the baseflow coefficient (soil parameter BASEFLOW_COEFF for the TOPSOIL; in this study parameter $x_4$), and $\phi_{soil}$ is the water storage [mm] in the soil or aquifer layer (Eq. S2). The alternate version BASE_LINEAR_ANALYTIC is used here. It simulates the same condition except using a closed-form expression for integrated flux over the time step ($\Delta t$):

$$N_1 = \phi_{soil} \cdot (1 - \exp(-k\Delta t)) / \Delta t$$

The two methods are effectively equivalent for sufficiently small time steps, but the second is preferred for large values of $k$. The second was used in this study.

**S.1.2.2 VIC baseflow method (`BASE_VIC`) used as option $N_2$**

From the VIC model (Wood et al., 1992) as interpreted by (Clark et al., 2008):

$$N_2 = M_{max} \left( \frac{\phi_{soil}}{\phi_{max}} \right)^n$$

where $M_{max}$ [mm/d] is the maximum baseflow rate at saturation (soil parameter MAX_BASEFLOW_RATE; in this study parameter $x_5$), $\phi_{soil}$ is the water storage [mm] in the soil or aquifer layer, $\phi_{max}$ is the maximum soil storage capacity , and $n$ is the user-specified soil parameter BASEFLOW_N (in this study parameter $x_6$).

**S.1.2.3    VIC baseflow method (`BASE_TOPMODEL`) used as option $N_3$**

From TOPMODEL (Beven and Kirkby, 1979) as interpreted by Clark et al. (2008):

$$N_3 = M_{max} \cdot \frac{\phi_{max}}{n} \cdot \frac{1}{\lambda^n} \cdot \left( \frac{\phi_{soil}}{\phi_{max}} \right)^n$$

where $M_{max}$ [mm/d] is the maximum baseflow rate at saturation (soil parameter `MAX_BASEFLOW_RATE`; in this study parameter $x_5$), $\phi_{soil}$ is the water storage [mm] in the soil layer, $\phi_{max}$ is the maximum soil storage capacity, $\lambda$ is the mean of the power-transformed topographic index [m] (terrain parameter LAMBDA; in this study parameter $x_7$), and $n$ is the user-specified soil parameter `BASEFLOW_N` (in this study parameter $x_6$).

**S.1.2.4    Weighted sum of all options used for quickflow process $N$**

The combined, weighted sum of the three options is used as quickflow estimate in Raven, i.e.

$$N \quad = \quad w_4 N_1 + w_5 N_2 + w_6 N_3$$

The three weights $w_i$ are derived from the two i.i.d. parameters $r_j$ sampled uniform from the unit interval following the approach described in Appendix A of the main manuscript:

$$w_4 \quad = \quad 1 - (1 - r_3)^{\frac{1}{2}}$$
$$w_5 \quad = \quad (1 - w_4) r_4$$
$$w_6 \quad = \quad 1 - w_4 - w_5$$

**S.1.3    Soil Evaporation $O$**

Soil evaporation (really evapotranspiration) involves converting water from the soil layers to water vapour in the atmosphere via both evaporation and transpiration. The rate of evapotranspiration depends on soil moisture, plant type, stage of plant development and weather conditions such as solar radiation, wind speed, humidity and temperature.

Soil evaporation always moves water between `SOIL[m]` and `ATMOSPHERE` units. Which soil layers are subjected to evaporation depend on the soil structure model specified by the `:SoilModel` command and the particular evaporation algorithm. Soil evaporation is rate-limited by the availability of soil/aquifer storage (dependent on the soil thickness which is parameter $x_{29}$ in this study) and by the capacity of the atmosphere to absorb water vapour.

In all notation below, PET refers to the potential evapotranspiration determined by one of the forcing function estimators for PET (see Raven manual). In all cases, this PET is modified by the soil parameter `PET_CORRECTION` (in this study parameter $x_8$), which only modifies PET in these algorithms.

**S.1.3.1    Uncorrected evaporation algorithm (`SOILEVAP_ALL`) used as option $O_1$**

Water is removed from soil at the maximum rate until there is no water remaining:

$$O_1 = \text{PET}$$

**S.1.3.2 TOPMODEL evaporation algorithm (`SOILEVAP_TOPMODEL`) used as option $O_2$**

Soil ET is at PET if storage exceeds the tension storage, then is linearly proportional to the soil saturation:

$$O_2 = \text{PET} \cdot \min(\frac{\phi_{soil}}{\phi_{tens}}, 1)$$

where PET is the potential evapotranspiration rate [mm/d], and $\phi_{soil}$ [mm] and $\phi_{tens}$ [mm] are defined in equation S2 (contains parameters `SAT_WILT TOPSOIL` $x_9$, `FIELD_CAPACITY TOPSOIL` $x_9 + x_{10}$, and thickness of TOPSOIL $x_{29}$). The HBV model uses an additional snow correction (in this study parameter $x_8$), such that ET is zero in non-forested areas if snow depth is non-zero.

**S.1.3.3 Weighted sum of all options used for soil evaporation process $O$**

The combined, weighted sum of the two options is used as soil evaporation estimate in Raven, i.e.

$$O \quad = \quad w_7 O_1 + w_8 O_2$$

The two weights $w_i$ are derived from one parameter $r_j$ sampled uniform from the unit interval following the approach described in Appendix A of the main manuscript:

$$w_7 \quad = \quad r_5$$
$$w_8 \quad = \quad 1 - w_7$$

**S.1.4 Baseflow Process $P$**

**S.1.4.1 Linear storage (`BASE_LINEAR_ANALYTIC`) used as option $P_1$**

The same linear storage computation as described in Sec. S.1.2.1 is used here. The only difference is the baseflow coefficient $k$ [1/d] that is now the soil parameter `BASEFLOW_COEFF` for the PHREATIC soil layer (in this study parameter $x_{11}$).

$$P_1 = \phi_{soil} \cdot (1 - \exp(-k\Delta t))/\Delta t$$

**S.1.4.2 Non-linear storage (`BASE_POWER_LAW`) used as option $P_2$**

The non-linear storage is a very common approach used in a variety of conceptual models, including HBV (Bergström, 1995). The baseflow rate is non-linearly proportional to storage:

$$P_2 = k\phi_{soil}^n$$

Where $k$ [1/d] is the baseflow coefficient (soil parameter `BASEFLOW_COEFF` here for the PHREATIC soil layer; parameter $x_{11}$), and $\phi_{soil}$ is the water storage [mm] in the soil or aquifer layer, and $n$ is the user-specified soil parameter `BASEFLOW_N` (in this study parameter $x_{12}$).

**S.1.4.3 Weighted sum of all options used for baseflow process $P$**

The combined, weighted sum of the two options is used as baseflow estimate in Raven, i.e.

$$P = w_9 P_1 + w_{10} P_2$$

The two weights $w_i$ are derived from one parameter $r_j$ sampled uniform from the unit interval following the approach described in Appendix A of the main manuscript:

$$w_9 = r_6$$
$$w_{10} = 1 - w_9$$

**S.1.5 Snow Balance Process $Q$**

Snow balance algorithms are used to simulate the strongly coupled mass and energy balance equations controlling melting and refreezing of snow pack and the liquid phase in the snow pores.

Most snow balance algorithms consists of multiple coupled equations, and there are also many 'to' and 'from' compartments, depending on which algorithm is selected. 'From' compartments include SNOW (as SWE), SNOW_LIQ and SNOW_DEPTH. 'To' compartments include SNOW, ATMOSPHERE, SNOW_LIQ, SNOW_DEPTH and SURFACE_WATER. Snow balance is rate-limited by the storage in 'from' and 'to' compartments.

Most of the snowmelt algorithms that explicitly simulate liquid water content within the snowpack use the global parameter SNOW_SWI to determine the maximum possible liquid water storage of the snowpack:

$$\phi_{max}^{sl} = SWE \cdot SWI$$

where $\phi_{max}^{sl}$ [mm] is the maximum liquid water storage of the snowpack, $SWE$ is the snow water equivalent of the snowpack [mm], and $SWI$ is the global parameter SNOW_SWI, which defaults to 0.05 if not specified.

**S.1.5.1 HMETS snow balance (SNOBAL_HMETS) used as option $Q_1$**

A snowmelt model documented in Martel et al. (2017). This is a simple single layer snowmelt model with degree day freezing, which tracks liquid water content in the snowpack in addition to SWE. The refreeze rate (constrained by water availability) is given by:

$$Q_1 = K_f \cdot (T_{rf} - T_{di})^f$$

where $K_f$ is the land use property REFREEZE_FACTOR (in this study parameter $x_{18}$), $T_r f$ is the degree day refreeze factor (land use property DD_REFREEZE_TEMP; parameter $x_{16}$), and $f$ is the land use parameter REFREEZE_EXPONENT (parameter $x_{17}$). The water retention capacity (upper limit of liquid water content in snow) varies over the course of the year based upon cumulative snowmelt:

$$\text{SWI} = \max\left(\text{SWI}_{min}, \text{SWI}_{max} \cdot (1 - \alpha \cdot M_{cumul})\right)$$

where SWI$_{min}$ and SWI$_{max}$ are the land use parameters `SNOW_SWI_MIN` (parameter $x_{13}$) and `SNOW_SWI_MAX` (parameter $x_{13} + x_{14}$), $\alpha$ is the land use parameter `SWI_REDUCT_COEFF` (parameter $x_{15}$), and $M_{cumul}$ is the cumulative melt since the last period of zero snow depth.

**S.1.5.2 Simple melt (`SNOBAL_SIMPLE_MELT`) used as option $Q_2$**

The melt rate (in [mm/d]) is simply calculated by applying the potential melt rate to the snowpack until it is gone.

$$Q_2 = \begin{cases} M_{potmelt}, & \text{if } S \geq 0 \\ 0, & \text{if } S < 0 \end{cases}$$

where $M_{potmelt}$ [mm/d] is calculated using the method described in section S.1.6.2, i.e. $M_{potmelt} = T_1$. Note that the simple melt process option $Q_2$ for simulating the snow balance does not include any tunable parameter $x$.

**S.1.5.3 HBV snow balance (`SNOBAL_HBV`) used as option $Q_3$**

The HBV snow balance (Bergström, 1995) represents both melt and liquid water storage in the pore space of the snow. The melt rate is determined by the potential melt rate algorithm (`POTMELT_HBV` for true HBV emulation), while refreeze is calculated using:

$$Q_3 = M_{refreeze} = K_a \cdot \max(T_f - T, 0)$$

where $K_a$ is the land use parameter `REFREEZE_FACTOR` [mm/d/°C] (in this study parameter $x_{18}$). Meltwater fills the snow pore space first with the maximum fillable pore space determined by the global parameter `SNOW_SWI` (in this study parameter $x_{19}$) and is then allowed to overflow. All overflow percolates into `SOIL[0]` by default, but may be redirected to `PONDED_WATER` using the `:Redirect` command if desired.

**S.1.5.4 Weighted sum of all options used for snow balance process $Q$**

The combined, weighted sum of the three options is used as snow balance estimate in Raven, i.e.

$$Q \quad = \quad w_{11}Q_1 + w_{12}Q_2 + w_{13}Q_3$$

The three weights $w_i$ are derived from the two i.i.d. parameters $r_j$ sampled uniform from the unit interval following the approach described in Appendix A of the main manuscript:

$$w_{11} \quad = \quad 1 - (1 - r_7)^{\frac{1}{2}}$$
$$w_{12} \quad = \quad (1 - w_{11})r_8$$
$$w_{13} \quad = \quad 1 - w_{11} - w_{12}$$

**S.1.6 Processes with Single Options With Tunable Parameters**

185

For the following processes only one option has been used during this study for simplicity. Each option is hence theoretically weighted with 1.0 in every model run.

**S.1.6.1 Convolution Processes for Surface and Delayed Runoff ($R$ and $S$)**

190 Since convolution methods store the time history of inputs to convolution storage of a duration consistent with the longest time delay in the convolution, it is not suggested to use convolution with a time constant in days with an hourly time step. Typically the order of the time delay should be on the order of the model time step.

The below convolution methods are available. All of them perform a discrete version of the following convolution:

$$R_1 = S_1 = \int_0^\infty UH(\tau)I(t-\tau)d\tau$$

where $I(t)$ is the input flux history (in mm/d) to the convolution storage unit and $UH(t)$ is the transfer function; the area under

195 the transfer function is always equal to one to ensure mass balance. For the convolution of the surface and delayed runoff two different transfer functions have been used.

**Gamma transfer function 1 (`CONVOL_GAMMA`) used as option $R_1$**

For the convolution of the surface runoff $R$ the following transfer function is used

$$UH(t) = \frac{1}{t}\frac{(\beta t)^a}{\Gamma(a)}\exp(-\beta t)$$

200 where $a$ and $\beta$ are the land use parameters `GAMMA_SHAPE` and `GAMMA_SCALE` (in this study parameters $x_{20}$ and $x_{21}$, respectively).

**Gamma transfer function 2 (`CONVOL_GAMMA2`) used as option $S_1$**

For the convolution of the delayed runoff $S$ the following transfer function is used

$$UH(t) = \frac{1}{t}\frac{(\beta t)^a}{\Gamma(a)}\exp(-\beta t)$$

205 where $a$ and $\beta$ are the land use parameters `GAMMA_SHAPE2` and `GAMMA_SCALE2` (in this study parameters $x_{22}$ and $x_{23}$, respectively).

**S.1.6.2 Potential Melt $T$**

Potential snow melt can be estimated using a number a methods in the Raven model. To set the appropriate process in the model the RVI must include the `:PotentialMeltMethod` keyword along with the appropriate value for the method selected. The

210 method selected here is:

**Potential Melt HMETS method (`POTMELT_HMETS`) used as option $T_1$**

A revised degree day model from the HMETS model (Martel et al., 2017), which uses a degree day factor which varies with cumulative snowmelt. The degree day model is given as

$$T_1 = M_a \cdot (T - T_f)$$

215 where $T$ is the daily average temperature. $T_f$ is the melt temperature (zero by default, but can be set with the land use parameter `DD_MELT_TEMP`; in this study parameter $x_{26}$), and $M_a$ [mm/d/°C] is the degree day melt factor, calculated as a function of cumulative melt:

$$M_a = \min\left(M_a^{max}, M_a^{min} \cdot (1 + \alpha \cdot M_{cumul})\right)$$

where the following land use parameters are used: the minimum melt rate $M_a^{min}$ [mm/d/°C] (`MIN_MELT_FACTOR`; in this
220 study parameter $x_{24}$), the maximum melt rate $M_a^{max}$ [mm/d/°C] (`MAX_MELT_FACTOR`; in this study parameter $x_{24} + x_{25}$), and $\alpha$ [1/mm] is the parameter `DD_AGGRADATION` (in this study parameter $x_{27}$).

**S.1.6.3   Percolation Process $U$**

Percolation refers to the net downward flow of water from one soil/ aquifer unit to another. This process is physically driven by a moisture gradient, but this is often simplified in conceptual percolation models.
225 Percolation moves water between `SOIL[m]` or `AQUIFER` units, depending upon the soil structure model specified by the `:SoilModel` command. The user typically has to specify both the 'from' and 'to' storage compartments. Percolation is rate-limited by the availability of soil/aquifer storage and by the capacity of the receptor 'to' compartment.

**Linear Percolation (`PERC_LINEAR`) used as option $U_1$**

Percolation is proportional to soil water content:

230 $$U_1 = k\phi_{soil}$$

where $k$ [1/d] is the soil parameter `PERC_COEFF` (in this study parameter $x_{28}$ for TOPSOIL and $x_{35}$ for PHREATIC soil layer) and $\phi_{soil}$ [mm] is defined in equation S2. All parameters refer to that of the 'from' soil compartment.

**S.1.6.4   Rain-Snow Partitioning Process $V$**

If only total precipitation is specified at a gauge station or grid cell, then this total precipitation is partitioned into rain and
235 snow. All of the provided algorithms in Raven calculate the snow fraction $\alpha_s$, and rain and snow are determined from:

$$
\begin{aligned}
R &= (1 - \alpha_s)P \\
S &= \alpha P
\end{aligned}
$$

where $R$ [mm/d], $S$ [mm/d], and $P$ are rainfall, snowfall, and total precipitation rates, respectively.

**Linear approach (`RAINSNOW_HBV`) used as option $V_1$**

In these approaches, a linear transition between all snow and all rain is determined from the average daily temperature, $T_{ave}$:

$$\alpha_s = 0.5 + \frac{T_{trans} - T_{ave}}{\Delta T}$$

in the range from $T_{trans} - \Delta T/2$ to $T_{trans} + \Delta T/2$, where $T_{trans}$ is the rain/snow transition temperature (global parameter `RAINSNOW_TEMP`, [$^\circ$C]; in this study parameter $x_{31}$) and $\Delta T$ is the global parameter `RAINSNOW_DELTA` [$^\circ$C] (in this study parameter $x_{32}$). If $T_{ave}$ is outside of this temperature range, the precipitation is either all snow ($\alpha_s = 1$) or all rain ($\alpha_s = 0$), accordingly. This snow fraction is applied for the entire day.

**S.1.6.5  Precipitation Correction Process $W$**

Measured total precipitation, snow precipitation, or rain precipitation may be corrected on a gauge-by-gauge basis by using gauge-dependent rainfall and snowfall corrections to correct for observation bias. This is handled using the `:RainCorrection` and `:SnowCorrection` commands given for each gauge. The parameters used in this study are $x_{33}$ and $x_{34}$, respectively.

**S.1.7  Processes with Single Options Without Tunable Parameters**

The following processes have been fixed at single options that do not contain tunable parameters. They have been added for completeness and have been labeled as process $X_1$ which is known a priori to have a sensitivity of zero since no parameter will be perturbed during the analysis. We refer to the Raven documentation (Craig, 2020) for details on those options. The options are SW_RAD_DEFAULT for Extraterrestrial Shortwave Generation, PET_OUDIN for potential evapotranspiration, ROUTE_DUMP for in-catchment routing, and in-channel routing is switched off (ROUTE_NONE) since only lumped catchments are analyzed here.

**References**

Bergström, S.: The HBV model, in: Computer Models of Watershed Hydrology, edited by Singh, V., pp. 443–476, 1995.

Beven, K. J. and Kirkby, M. J.: A physically based, variable contributing area model of basin hydrology / Un modèle à base physique de zone d'appel variable de l'hydrologie du bassin versant, Hydrological Sciences Bulletin, 24, 43–69, 1979.

Clark, M. P., Slater, A. G., Rupp, D. E., Woods, R. A., Vrugt, J. A., Gupta, H. V., Wagener, T., and Hay, L. E.: Framework for Understanding Structural Errors (FUSE): A modular framework to diagnose differences between hydrological models, Water Resources Research, 44, https://doi.org/10.1029/2007WR006735, 2008.

Craig, J. R.: Raven: User's and Developer's Manual v3.0, http://raven.uwaterloo.ca/files/v3.0/RavenManual_v3.0.pdf, 2020.

Craig, J. R., Brown, G., Chlumsky, R., Jenkinson, W., Jost, G., Lee, K., Mai, J., Serrer, M., Snowdon, A. P., Sgro, N., Shafii, M., and Tolson, B. A.: Flexible watershed simulation with the Raven hydrological modelling framework, Environmental Modelling & Software, p. 104728, https://doi.org/https://doi.org/10.1016/j.envsoft.2020.104728, 2020.

Martel, J.-L., Demeester, K., Brissette, F., Poulin, A., and Arsenault, R.: HMETS—A Simple and Efficient Hydrology Model for Teaching Hydrological Modelling, Flow Forecasting and Climate Change Impacts, International Journal of Engineering Education, 33, 1307–1316, 2017.

Wood, E. F., Lettenmaier, D. P., and Zartarian, V. G.: A land-surface hydrology parameterization with subgrid variability for general circulation models, Journal of Geophysical Research, 97, 2717–2728, 1992.

---

## Author Comment (AC2) · 29 Sep 2020

**Reply to Anonymous Referee #2**
*Review received and published: 3 September 2020*

Dear Reviewer,

Thanks a lot for your thorough review and the valuable suggestions. We will reply below in detail to your comments. Your comments are *italic*; our replies are highlighted **bold**. The **line numbers in red** are referring to the revised draft.

Best regards,
Julie, James, and Bryan

*The manuscript focuses on Sensitivity Analysis (SA) of hydrological models. It introduces a more general version of the well-known Sobol' method, designed to operate on groups of parameters instead of on individual parameters. Overall I enjoyed reading the manuscript - its on a topical area and the methods described are sound. I appreciate this work on mathematical model analysis, and the idea of grouped parameter sensitivity is novel at least in hydrology as far as I know. With multi-model/flexible frameworks such as RAVEN and others, analysis of their sensitivity would benefit from such "grouped" analysis.*

**We thank the reviewer for this positive evaluation and are glad to hear that it was an enjoyable read.**

*I have the following concerns with the current manuscript form:*

1. *The algorithms are not explained in a sufficiently clear way. For example, for the description of Sobol' method on lines 300-307, and the description of the xSSA method on lines 324-330, are in my opinion not sufficient for a paper presenting a mathematical method. Yes, I could probably translate the description there into a procedure/ pseudocode, but: first I would not be quite sure if I got it right, and second I (respectfully) suggest the onus is on the authors to provide such an un-ambiguous description. Appendix B is helpful to a degree, but seems to use a different notation to the main text (where are the matrices A and B and Cm?).*

   **We understand that the reviewer might have been confused due to the lack of some information. The Sobol' method is however one of the most cited methods in sensitivity analysis and part of every single package providing implementations to this method. We refer to the most relevant publication, i.e. Sobol' (1993) in the section about the traditional Sobol' Sensitivity Analysis (Sec. 2.2.1) and even explain the method in that section. There is really nothing more to it than sampling a matrix $\mathcal{A}$, $\mathcal{B}$ and construction the matrices $\mathcal{C}_m$ as explained. The user would not even need to do that if using a package. The matrices are of dimension $K \times N$ with $K$ being 1000 in our experiments and $K$ being 11 to derive for example the parameter sensitivities of the shared benchmark problem. We do not think that listing those matrices would help to understand the method. We adjusted slightly the text regarding the sampling of the matrices $\mathcal{A}$ and $\mathcal{B}$ and hope that it is now more clear that those are purely sampled:**

   **line 323 ff.** **For the numerical estimation of the indexes $S_{x_i}$ and $ST_{x_i}$, one samples two base matrices $\mathcal{A}$ and $\mathcal{B}$ which each contain $K$ parameter sets (rows) of $N$ parameters (columns). The samples are assumed to be independent within one matrix and between the matrices. We used the stratified sampling of Sobol' sequences here to improve convergence speed of the derived indexes compared to a Monte-Carlo sampling.**

In Appendix B we solely list the analytical results of the shared-parameter benchmark as a reference that users could test their implementations in case this is needed. We would also like to emphasize that we are planning to make all codes used here openly accessible once this manuscript gets accepted for publication.

2. *Terms such "uncertainty", "sensitivity", "influence", "importance" are being used in a pretty loose, seemingly interchangeable way. For example, the paragraph on lines 31- 40, which starts with "uncertainty" and then immediately switches to "sensitivity". Then line 104 mentions "sensitive/influential/important" parameters. Are these referring to the same characteristic? Similar confusing usage then carries through later in the manuscript. I suggest the terminology should be much tighter to avoid confusion. Given the mathematically demanding topic, I would suggest giving clear definitions of the various concepts (with links to existing literature where appropriate), and avoiding the alternation of these terms in the remainder of the presentation. There are useful and interesting ideas on lines 100-115, but these are already using the terms above in a way I found unnecessarily confusing because its not clear which terms are used synonymously and which are not.*

We have revised the final paragraph to the introduction clarifying some general definitions for the entire manuscript. We hope that it is now more clear that we use most of the terms you listed interchangeably. We apologize that this was confusing in our first version of the draft. For example:

**line 141 ff.** We also wish to mention that the terms 'sensitive' and 'influential' are used interchangeably throughout this work.

We agree that it might lead to too much leeway of interpretation for a reader when using too many interchangeable terms. We went through the manuscript and hope that we reduced this ambiguity. Here are some examples:

**line 521 ff.** The analysis of model parameters (Fig. 5A) shows that the most sensitive ones are [...]

**line 586 ff.** The strong impact of these processes (together with the input adjustments) highlights the sensitivity of streamflow regarding snow and melting processes in this mountainous, energy-limited catchment.

**line 589 ff.** This demonstrates that soil and surface processes are of secondary sensitivity regarding streamflow. Their sensitivity may increase if the uncertainty of the snow and melting processes can be reduced, i.e. by narrowing parameter ranges during calibration.

**line 610 ff.** Evaporation (dark blue) is [...], expectedly, less sensitive during winter. Snow balance (medium green) and potential melt (orange) are sensitive as long as snow is present (Nov to May).

*The current literature review is heavily focused on sensitivity analysis - which is appropriate given the topic. But if the connection to uncertainty is to be made, I would say the literature review of the latter is currently rudimentary at best.*

We agree which the reviewer that our literature review heavily focuses on the sensitivity topic given that we try to make a contribution to tat field. There is a connection between uncertainty of a parameter (reflected in the range we associate to a parameter and we then use as search space in calibration or to derive the sensitivity of a parameter) and its sensitivity. The range of a parameter is influencing its sensitivity, i.e. if the (uncertainty) range of a parameter gets reduced its sensitivity will also get reduced. But it might be that the parameter overall is insensitive (no matter which range picked). So, it is hard to determine from the range alone a sensitivity of a parameter; hence sensitivity analyses are performed. We thought that this is an obvious connection and did not want to

distract the reader by putting too much emphasize on that topic. We have made the following adjustments in the manuscript:

**line 16 ff.** [...] such information may readily inform model calibration and uncertainty analysis.

**line 34 ff.** A key purpose of model sensitivity analysis is to inform model calibration or model uncertainty analysis so as to focus either of these analyses on only the model inputs/model structural choices the model outputs are most sensitive to.

3. *The aims and key contributions of the study seem to drift over the course of the manuscript/ presentation. For example the Introduction is focused on sensitivity analysis (and to some extent uncertainty) - but in the Conclusions the contribution #1 is listed as formulating model ensembles as weighted sums of process options, with Sensitivity Analysis then being contribution #2. I think the coherence between the introduction/ aims and contributions could be improved, so that there is a clearer set of aims, appropriate background given on each aim, and then a clear set of conclusions that match those aims.*

We totally agree with that and apologize that the key contributions had not been clearly stated in the introduction. We rephrased the following paragraph to match the order and wording we use in the conclusions. We hope that our key contributions are now more clear to the reader.

**line 96 ff.** Two main contributions of this work are to (A) reformulate a hydrologic modeling framework so that it can define model structure by weighting or blending of discrete model process options continuously for simulating process level hydrologic fluxes and (B) to propose a technique, the Extended Sobol' Sensitivity Analysis (xSSA) method, based on the existing concept of grouping parameters when applying the Sobol' method (Sobol and Kucherenko, 2004; Saltelli et al., 2008; Gilquin et al., 2015) to derive the sensitivity of a model prediction (here streamflow) to model structural choices.

*A clearer vision of the contributions could also help improve the structure of the manuscript, by putting the important contributions much earlier. This would avoid the multiple forward references to the proposed method and its properties before its actual description is given- e.g., see lines 235-237, which are not really that meaningful before seeing how the xSSA method operates. The new xSSA method in Section 2.2.2 comes after several quite detailed sections on models and case studies - and it was not immediately apparent that this is the main advance being presented.*

We indeed started our initial draft with the proposed reversed order (first explaining the xSSA method including the grouping and weights of process options and then explaining the benchmark models, study domain and Raven). The problem then is that most readers will have trouble to understand what we mean with process options and processes (in a real-world) example. We therefore decided to explain first all the "vocabularies" (i.e., benchmarks, hydrologic model, process options and processes) before getting to how a sensitivity analysis would work without defining groups (Sec. 2.2.1) leading to the version that is then based on groups, i.e. xSSA (Sec. 2.2.2). This is then followed by all the experiments we intent to run (Sec. 2.3). We made a few adjustments in the introductory paragraph for the Material & Methods. We are hoping that this will help the reader to navigate through this long section and maybe directly go to the section with the major contribution:

**line 144 ff.** The section will first introduce the models and their setups (Sec. 2.1) used to test and validate the proposed Extended Sobol' Sensitivity Analysis

(xSSA) method as here applied to determine model structure and parameter sensitivities. In section 2.2, we will briefly revisit the traditional method of Sobol' that is so far primarily used to obtain model parameter sensitivities (sensitivity metric A; Sec. 2.2.1) before we introduce the major contribution of this work (Sec. 2.2.2) which supports sensitivity estimates for model process options (sensitivity metric C) and model processes (sensitivity metric D) besides the sensitivities of model parameters (sensitivity metric B). Finally, we present the experiments used to test the proposed method and address the research questions raised in the introduction (Sec. 2.3).

4. *Some lack of clarity in how important new concepts are defined. E.g., is the sensitivity to groups of parameters taken as sensitivity to processes? Or is that something different? Please check wording across manuscript.*

   We have added a final paragraph to the introduction clarifying some general definitions for the entire manuscript. We hope that it is now more clear that all sensitivities we derive are regarding streamflow in this study. The method however is not at all limited to streamflow but can be applied to any model output of interest.

   **line 99 ff.** [...] applying the Sobol' method (Sobol and Kucherenko, 2004; Saltelli et al., 2008; Gilquin et al., 2015) to derive the sensitivity of a model prediction (here streamflow) to model structural choices.

   **line 134 ff.** We propose a method for estimating how sensitive a simulated model output is to groups of parameters. We have chosen here streamflow as this model output as it is the fundamental and most important and common output variable in hydrologic studies. The sensitivities of the groups of parameters is hence obtained regarding streamflow. The groups defined here are either individual parameters (metric B) or the set of parameters that is used in an individual process option (metric C) or all parameters used in any available process option for a modelled process (metric D). We acknowledge that the definition of these groups is subjective and has been chosen here to demonstrate a novel approach of how to evaluate process and process option sensitivities, i.e., how sensitive is the simulated streamflow regarding the choice of a specific infiltration process description or how sensitive is the simulated streamflow regarding infiltration in general.

   The abstract also highlights for the two results 3) and 4) that results are regarding streamflow. But we understand that it should have been more emphasized in the manuscript.

- *Line 115-122 - I suggest this summary of findings would work better in Abstract + Conclusions. It would also help being clearer in the wording on the comparisons that are being made. Is "conventional" approach the xSSA or the Baroni method?*

   The paragraph the reviewer is pointing out here is indeed explaining the outline of the study and the major analyses that will be undertaken. The reviewer is right that the last sentence is containing a result/outcome and has hence been removed:

   **removed:** "The method is demonstrated to be more efficient than a conventional approach (see metric A) whereby the standard Sobol' method is repeatedly applied to distinct model structures as in the study by Van Hoey et al. (2014), in addition to providing more useful information regarding model sensitivities."

   This efficiency improvement had been mentioned in the abstract:

 **2) The xSSA method with process weighting is computationally less expensive than the alternative aggregate sensitivity analysis approach performed for the exhaustive set of structural model configurations, with savings of 81.9% for the benchmark model and 98.6% for the watershed case study.**

and the conclusions:

 **The method of weighted process options is shown to significantly reduce number of model runs required to run a sensitivity analysis based on model parameters. For the shared-parameter benchmark model 81.9% fewer model runs are required (A: $72\,000$ vs B: $13\,000$). For the hydrologic model example, the reduction is greater than 98.6% (A: $3\,258\,000$ vs B: $45\,000$).**

already in the previous version of the manuscript. Besides removing the sentence claiming already results, the paragraph the reviewer mentions, remains unchanged and we hope that the reviewer agrees that an outline of the major components of this study is helpful to get the reader prepared for the remaining part of the manuscript.

- *Line 278, where it is pointed out that a traditional single-parameter SA analysis could produce grouped-sensitivity analysis by aggregating results for individual parameters? In a paper advocating the new "grouped-SA" method - should such comparison receive priority to show the advantages of the new method. The hypothetical scenario where sensitivity is underestimated (line 279) - is this common in practice? As this goes to the motivation for the new method, I think it could receive more attention.*

We agree. We think that the approach of estimating parameter sensitivities for each individual enumerated model (like the 12 theoretical models) and then using the average of all the sensitivities per parameter (basically the mean of each column in Table B1) would be the most obvious way to come up with a sensitivity for each parameter across multiple models. We derived these average sensitivities $\overline{\mathcal{S}_{x_i}^n}$ of the seven parameters of the shared-parameter benchmark (mean of each column in Table B1) and compare them to the parameter sensitivities $\mathcal{S}_{x_i}$ derived with the weighted model approach (Eq. B1):

$$\begin{bmatrix} \overline{S_{x_1}^n} \\ \overline{S_{x_2}^n} \\ \overline{S_{x_3}^n} \\ \overline{S_{x_4}^n} \\ \overline{S_{x_5}^n} \\ \overline{S_{x_6}^n} \\ \overline{S_{x_7}^n} \end{bmatrix} = \begin{bmatrix} 0.1223 \\ 0.2978 \\ 0.0506 \\ 0.0699 \\ 0.0699 \\ 0.0288 \\ 0.6506 \end{bmatrix}, \begin{bmatrix} \overline{ST_{x_1}^n} \\ \overline{ST_{x_2}^n} \\ \overline{ST_{x_3}^n} \\ \overline{ST_{x_4}^n} \\ \overline{ST_{x_5}^n} \\ \overline{ST_{x_6}^n} \\ \overline{ST_{x_7}^n} \end{bmatrix} = \begin{bmatrix} 0.3238 \\ 0.5052 \\ 0.1321 \\ 0.2562 \\ 0.2562 \\ 0.0288 \\ 0.6506 \end{bmatrix} \text{ and } \begin{bmatrix} S_{x_1} \\ S_{x_2} \\ S_{x_3} \\ S_{x_4} \\ S_{x_5} \\ S_{x_6} \\ S_{x_7} \end{bmatrix} = \begin{bmatrix} 0.0230 \\ 0.3806 \\ 0.0022 \\ 0.0002 \\ 0.0002 \\ 0.0000 \\ 0.0393 \end{bmatrix}, \begin{bmatrix} ST_{x_1} \\ ST_{x_2} \\ ST_{x_3} \\ ST_{x_4} \\ ST_{x_5} \\ ST_{x_6} \\ ST_{x_7} \end{bmatrix} = \begin{bmatrix} 0.0753 \\ 0.7709 \\ 0.0045 \\ 0.0010 \\ 0.0010 \\ 0.0000 \\ 0.0524 \end{bmatrix}$$

The sensitivities of the parameters in both approaches vary drastically. For example, the most sensitive parameter when using the 12 enumerate models is parameter $x_7$ while the most sensitive parameter with the proposed approach is parameter $x_2$. The huge differences are mostly explained with the fact that in the first approach not all parameters are active all the time. Another reason is the interaction between parameters. Even though we have these results, we choose to keep this out of the paper so as to not complicate the discussion.

- *Line 352 "limitations of existing Baroni method" - as this comparison is important in this paper - would seem preferable to describe the Baroni method in appropriate detail before discussing its limitations.*

We now provide a more detailed description of this method in the introduction and hope this helps to understand the method itself, and makes the comparison

to the proposed method easier. The method is now referred to as the Discrete Values Method based on a comment of another reviewer.

**line 50 ff.** To date, there have been limited attempts to simultaneously estimate model parameter, input, and structural sensitivities. One notable attempt is introduced by Baroni and Tarantola (2014) using a Sobol' sensitivity analysis based on grouped parameter. In that study, groups of soil and crop parameters, the number of soil layers, and a group of parameters to perturb inputs are investigated. These groups of parameters are pre-sampled and a finite set of parameters for each of the four groups is chosen and each set is enumerated. The sensitivity analysis is then based on those enumerated sets. This means, rather than sampling each individual parameter like in a classic Sobol' analysis, an integer for each group acting as a hyper-parameter is sampled. The model is then run with the associated pre-sampled parameter set. While the approach may be generally applicable to arbitrary structural differences, in their testing, Baroni and Tarantola (2014) varied only in how the model was internally discretized (i.e., in the number of soil layers). The soil and crop parameters were always used for the same soil and crop process. The major limitation of this method is, however, that individual parameters need to be mutually exclusive and can only be associated to one type of uncertainty. The method hence limits the groups that can be defined, for instance, overlapping group definitions are not possible. The method will be referred to as "discrete values method (DVM)" in the following and will be contrasted to the method developed here to examine this limitation in more detail.

- *Line 535: "it can be deduced that the potential melt, the quickflow options BASE_VIC and BASE_TOPMODEL, and the evaporation options are most influential upon modeled streamflow". Here the lack of clarity on what is meant by "influential" can cause confusion to a reader. Especially sensitivity to a specific option for a process (eg, BASE_VIC for quickflow) - normally sensitivity is to a range of possible values for a decision - here it is to a single specific value? I don't quite follow this.*

The reviewer is right that (parameter) sensitivity is regarding the range of possible parameter ranges. The sensitivity of process options is now the sensitivity of the model output (here streamflow) based on a range of different settings of these process options. This means that a process option is picked and a range of different parameterizations for this process option is tested and evaluated which impact (sensitivity) is has on the model output (here streamflow). Compared to the standard parameter sensitivity where the impact of one parameter is evaluated, the proposed method evaluated the impact of a group of parameters. It could technically be any group of parameters but we decided to define each group as either the parameters that belong to a certain process option and compare those groups' sensitivities or to group all parameters of an entire process to estimate the impact (sensitivity) of this group (process) on the streamflow.

In the paragraph the reviewer mentions, we identify that potential melt, two quickflow options and the evaporation options have the most impact on the simulated streamflow time series. Means that if the parameterization of any of those process options is modified the simulated streamflow would change more than with any other change made in any of the other process options.

- *Section 3.3 - nice sections. Would be improved by providing clearer definitions of sensitivity, influential processes, uncertainty, etc (see earlier comment). Current usage is unnecessarily loose and confusing here.*

We agree with the reviewer that using too many of those terms interchangeably

might lead to confusion for the readers and might lead to too much leeway of interpretation. We went through the manuscript and hope that we reduced this ambiguity.

**line 521 ff.** The analysis of model parameters (Fig. 5A) shows that the most sensitive ones are [...]

**line 586 ff.** The strong impact of these processes (together with the input adjustments) highlights the sensitivity of streamflow regarding snow and melting processes in this mountainous, energy-limited catchment.

**line 589 ff.** This demonstrates that soil and surface processes are of secondary sensitivity regarding streamflow. Their sensitivity may increase if the uncertainty of the snow and melting processes can be reduced, i.e. by narrowing parameter ranges during calibration.

**line 610 ff.** Evaporation (dark blue) is [...], expectedly, less sensitive during winter. Snow balance (medium green) and potential melt (orange) are sensitive as long as snow is present (Nov to May).

*Many these comments focus on presentation , but given the technically demanding nature of the work, a more targeted presentation would make it easier to digest by an interested reader.*

*Other comments*

1. *Line 4: "apply" or "develop"?*

   We still use apply here since using grouped parameters for a Sobol' analysis has been used in the past. The groups just never have been based on process options and processes. The part of xSSA that is new is the usage of weights to have several process options active in parallel. To avoid confusion and not ovre-selling our new approach, we prefer to stick with "apply".

2. *Line 24 - what is "they" referring to? Also what does "non-unique" refer to here? Is this with regard to many models co-existing in the literature? Or non-uniqueness in their inversion when estimating parameters? I think some clarity would be useful here*

   It refers to the conceptual models that are non-unique (as it depends on the concept and simplifications made by the individual modeler defining the conceptual model). We rephrased the beginning of the sentence to the following:

   **line 24 ff.** The model descriptions are also non-unique as they depend on the modelers simplifications and choices made during the model conceptualization. A large number of non-unique process algorithms can be found [...]

3. *Line 27 - are these decisions always subjective? Surely there exist studies where model decisions are developed according to sensible strategies?*

   We think that these decisions are made subjectively in most of the cases. In case a modeler picks one hydrologic model (based on experience, availability, expertise in a research group, etc), the choice of the process options is fixed (and hence subjective) from the start as the process option/conceptualization of that model would be chosen. Nearly, nobody would ever question that set of process algorithms (unless the focus of the study is to improve that model). In case a modeling framework is used the modeler would be faced from the beginning of which process algorithm to use. We are not aware of any study that is proposing an objective selection of the most appropriate process definition.

4. *"Sensitivity to model structural uncertainty" - I think studies such as McMillan et al. (2010); Clark et al. (2011) and other have investigated this?*

**We added those citations.**

5. *"recent" - with references back to 2008 is this still recent?*

   **Probably not all of them are "recent" but some are. We anyway just removed that word.**

6. *Baroni method - seems an important method in the context of this work. I think it would be helpful to provide the gist of that method at least in an Appendix, in the way that is is applied here.*

   **We totally agree. We have added a brief description about the Baroni method (now called "discrete values method (DVM)") to the introduction:**

   **line 50 ff.** **To date, there have been limited attempts to simultaneously estimate model parameter, input, and structural sensitivities. One notable attempt is introduced by Baroni and Tarantola (2014) using a Sobol' sensitivity analysis based on grouped parameter. In that study, groups of soil and crop parameters, the number of soil layers, and a group of parameters to perturb inputs are investigated. These groups of parameters are pre-sampled and a finite set of parameters for each of the four groups is chosen and each set is enumerated. The sensitivity analysis is then based on those enumerated sets. This means, rather than sampling each individual parameter like in a classic Sobol' analysis, an integer for each group acting as a hyper-parameter is sampled. The model is then run with the associated pre-sampled parameter set. While the approach may be generally applicable to arbitrary structural differences, in their testing, Baroni and Tarantola (2014) varied only in how the model was internally discretized (i.e., in the number of soil layers). The soil and crop parameters were always used for the same soil and crop process. The major limitation of this method is, however, that individual parameters need to be mutually exclusive and can only be associated to one type of uncertainty. The method hence limits the groups that can be defined, for instance, overlapping group definitions are not possible. The method will be referred to as "discrete values method (DVM)" in the following and will be contrasted to the method developed here to examine this limitation in more detail.**

   *It is also a little unclear from the abstract that a comparison to this method is made. Eg line 13 "alternative" - if this is Baroni's method - should this be "existing" method? To avoid a confusion the reference algorithm should be clearly described.*

   **The "alternative" method mentioned in the abstract does not refer to the Baroni method but to the method "performed for the exhaustive set of structural model configurations". We do not mention the comparison to the Baroni method in the abstract as we want to focus on the novelty of using process weights in the abstract and the results accompanied with this novelty. The comparison to the Baroni method is solely to highlight that there might be limitations in the existing approaches that are overcome by the proposed method.**

7. *line 49 - "did not change when moving between model structures" - is this for different hydrological models? or models from across multiple disciplines?*

   **This does not appear anymore since we rewrote the section about the Baroni method (see reply to comment above) and hope that it is less confusing now.**

8. *line 50 - what are "hyper-parameters"?*

   **Hyper-parameters can be for example multipliers that are applied to all parameters in a group (e.g., porosity parameters of all soil types) rather than analyzing**

every individual parameter independently. The manuscript now explains them a bit more through:

> **line 54 ff.** This means, rather than sampling each individual parameter like in a classic Sobol' analysis, an integer for each group acting as a hyper-parameter is sampled. The model is then run with the associated pre-sampled parameter set.

9. *line 52 - not entirely clear what "form" refers to here. I found the entire sentence a bit confusing when trying to understand exactly what its trying to say*

We meant "type [of uncertainty]". We changed that in this paragraph (rewritten section about the Baroni method) and throughout the whole manuscript.

> **line 58 ff.** The major limitation of this method is, however, that individual parameters need to be mutually exclusive and can only be associated to one type of uncertainty.

10. *line 53 - "the method introduced ..." - is an incomplete sentence?*

Resolved. This paragraph was rewritten entirely to add a better description of the Baroni method (see reply to comment 6).

11. *line 55 - "individual" - maybe clarify that the previous study assessed ONLY combined sensitivities? This is not clear from the current wording. And I thought that combined sensitivities are an advance rather than individual sensitivities? So why is that a limitation of the previous work?*

Yes, correct. Only the sensitivity of the model regrading all parameters (parameter uncertainty) was determined. But it is not clear which parameter(s) are causing this sensitivity. We do not think (and also do not state) that this is a disadvantage. It might just limit the insights of such an analysis as we mention in the last sentence of that paragraph: "Similar to the discrete values method, parameters were treated in an aggregate fashion which made it impossible to attribute the parameter sensitivity to a certain parameter or model component."

12. *line 62 - "sensitivity of a model" - is this for model simulations? or model parameters? or both? See comment about making sure the key concepts are clearly defined*

We mean the sensitivity of a model output here. We explain in the second half of that sentence that Van Hoey et al. (2014) is looking at parameter and structural sensitivities. We added "output" to the text now and hope it is less confusing.

> **line 71 ff.** Van Hoey et al. (2014) is one of the few studies that explicitly examined the sensitivity of a model output to changes in process representation, estimating sensitivities of parameters of various model structures with two or three alternatives per process, e.g., linear vs. non-linear storage; with or without an interflow process.

13. *line 78-79 - "it is therefore ..." - i think these ideas on the utility of SA should be introduced earlier in the presentation, to provide a stronger motivation and a practical context for the work.*

We think that introducing first the current status of the work in the field of structural sensitivity analysis is a better order. Making the claim that we are limited in the way we can analyze structural uncertainty without having any of these terms and approaches introduced, seems to be much more confusing. We hope that the reviewer agrees- especially since we explain some concepts now (hopefully) more clear in the paragraphs before.

14. *line 88 - this property "structure can vary continuously" / "weighted average". I found this aspect quite interesting in the work. The statement below that xSSA "is made uniquely possible" to RAVEN - do you mean it can only be used by RAVEN? This seems strange as multi-model ensembles where each model has a weight are fairly common (e.g., see the "model averaging" literature).*

    **The multi-model averaging community would always average the final model outputs (here streamflow). Raven allows the user to get an internal average of, for example, the amount of snow melting in each time step before this amount of meltwater is then used to derive anything else in the model (e.g., the amount of water infiltrating in this time step). To date this is (to our knowledge) only possible in Raven but could be implemented in any model that is allowing for several process options.**

15. *line 96 - "uniquely"?*

    **We hope that our response to the comment above (#14) resolves this issue and makes clear that this feature is indeed very unique.**

16. *line 105: Metric B - very interesting concept. but without some elaboration seems potentially ill-defined. Eg, how do you determine if a parameter appearing in different model structures is "the same parameter"?*

    **The reviewer is totally right that it is in the eye of the modeler (or person setting up the sensitivity analysis) to treat a parameter that appears in several process options as "the same". We treated the parameters the same if they have the same units (and therefore ranges that would be tested) and would be treated the same in a model analysis and interpretation of results. One example would be the depth of the top soil layer (in [mm]). Several processes and process options use this parameter and there is certainly no ambiguity of how to interpret- for example- the optimal value of this parameter. Other examples might be the temperature where snow is melting or the porosity of a soil type. Of course there might be parameters where this is less obvious but the person setting up the study could handle those parameters separately, e.g. soil depth method 1 and soil depth method 2. The method however gives us the huge opportunity to evaluate what some model parameters actually mean and if some parameters across conceptualizations are comparable.**

17. *line 120: "conventional approach" - is this the Baroni method? If so best to name it. Also it was referred to as "alternative" in the Abstract*

    **No, we refer here to the traditional/conventional/alternative approach as used for Metric A. We agree that this is confusing and rephrased the sentence to:**

    **line 113 ff.** **We here pose these as four distinct sensitivity metrics:**

    A. **Conditional parameter sensitivity:** *Which model parameter is most influential given a certain model structure?*
       **For example, which model parameter is most influential in the HBV model? (This is the traditional Sobol' metric. This conventional approach would test all possible models and derive parameter sensitivities conditional on the model tested.)**
    B. **...**

    **We also made an adjustment at another place in the introduction and hope that this strengthens the understanding of the link between the proposed method and the "traditional" method:**

    **line 100 ff.** **To our knowledge, the method of grouping parameters to derive sensitivities of parameters, process options, and processes without the explicit**

necessity of averaging parameter sensitivities after deriving them for individual models (referred to as conventional/ traditional sensitivity analysis) has not yet been applied.

We also would like to highlight that the section explaining this method is called "Traditional Sobol' Sensitivity Analysis".

18. *Section 2 - consider splitting into several sections and place in order of relevance to the contributions of the paper*

We find it very hard to make that clearer than it is right now. We previously tried to organize the Material and Methods following the metrics A-D we introduce but that leads to a significant duplication of information. On the other hand the contributions are plenty-fold and will likely not be able to be condensed in one paragraph- even though everything comes together in Section 2.2. The introductory paragraph for section 2 is meant to help the reader to navigate through the section. We hope that our earlier response regarding the best structure of the Methods section (major comment #3) also helps to make this more clear.

19. *line 145 - see earlier comment - how do you know it the "same" parameter? It seems a relevant discussion point*

Yes, it is. Please see our reply above (comment 16).

20. *line 169/ eqn 16 - how do you "decide" in a modeling context what is a shared parameter? Say is $x_3$ in eq 16 the same as $x_3$ as in eqn 13? Is this considered determined purely by the choice made by the modeler regarding the parameters to calibrate?*

We hope our explanation above (comment 16) helps to clarify this. The modeler has always the option to define these parameters separately for each process option but in a lot of cases it is obvious that these parameters are shared.

21. *line 235-236 - I think these are discussion points - would work better in Discussion rather than forward references here - at this point of the paper the new method is not described yet!*

Agreed. We remove the following sentence from the Material & Methods: "However, in the case of a framework without weights for process options, the application of the method would be much less efficient." We discuss the improvement of efficiency in the Conclusions.

22. *Line 312 - would help clarify here that this is approach is new and introduced in this work. And as mentioned earlier - I think it would benefit from being given more prominence in the paper.*

We added the following to that paragraph:

**line 338 ff.** Although the grouping of parameters has previously been used (Sobol and Kucherenko, 2004; Saltelli et al., 2008; Gilquin et al., 2015), it is- to our knowledge- the first time they have been used to group parameters of process options in the context of examining model structure sensitivity.

23. *line 318 - "depicts"?*

We think the reviewer points to line 320. There it says "[...] where $V$ depicts variances and $E$ expected values." We do not know exactly what the reviewer means with his/her comment and do not know what to adjust.

24. *line 409 - "hereafter called Baroni method" - already said this earlier on line 48 - but still referring to this method by multiple names*

We now refer to this method consistently as the "discrete values method (DVM)".

**Besides we think that it might be helpful to introduce this abbreviation again at the beginning of the results- just as a quick reminder for the readers.**

25. *Appendix A - an extra 1-2 sentences that refer to where in the main text are these weights used would be helpful here*

    **We agree. We added the following as an introduction in Appendix A:**

    **line 659 ff.** **In this work we define a model that is using the weighted average of a set of process options instead of choosing one fixed process option (Eq. 18). This is enabling to analyze several model structures at the same time by either setting weights to 0 or 1 (which selects exactly one option) or any weight in between which leads to the weighted average of those process option outputs.**

    **The sampling of such weights needs to lead to independent and identically (not necessarily uniform) distribution for each of the weights $w_i$.**

26. *Appendix B - I am confused why this seems duplicated in the Intro and the Appendix. If this is new - would seem better somewhere in the Theory and then Discussed, where it can be discussed in appropriate detail.*

    **We are sorry that this duplication led to confusion. We initially thought that the repetition of the four metrics would be convenient for the reader. We removed this now and hope the readers will remember those metrics (A-D) from the introduction. We also highlight the four different metrics in bold font in the Appendix B now. We hope that further improves the readability.**

*Figures*

1. *Figure 2 - the blue font in panel B is quite hard to read*

   **We are using a darker blue shade now for the bars and the left y-axis label.**

2. *Figure 5 (and others to various extents) - could be more generous with fontsize, as many labels etc are virtually illegible*

   **We increased the fontsize of the tick labels in Figure 5 and the fontsize used in the legend of Figure 6. Figure 5 is now also "rotated" by 90 degrees. That should make it much easier to read all labels.**

**References**

Baroni, G. and Tarantola, S.: A General Probabilistic Framework for uncertainty and global sensitivity analysis of deterministic models: A hydrological case study, Environmental Modelling & Software, 51, 26–34, 2014.

Clark, M. P., Kavetski, D., and Fenicia, F.: Pursuing the method of multiple working hypotheses for hydrological modeling, Water Resources Research, 47, 5468–16, 2011.

Friedl, M. and Sulla-Menashe, D.: MODIS/Terra+Aqua Land Cover Type Yearly L3 Global 500 m SIN Grid [data set], NASA EOSDIS Land Processes DAAC 10, 2015.

Gilquin, L., Prieur, C., and Arnaud, E.: Replication procedure for grouped Sobol' indices estimation in dependent uncertainty spaces, Information and Inference A Journal of the IMA, 4, 354–379, 2015.

McMillan, H., Freer, J., Pappenberger, F., Krueger, T., and Clark, M.: Impacts of uncertain river flow data on rainfall-runoff model calibration and discharge predictions, Hydrological Processes, 47, 1270–1284, 2010.

[Figure]

Figure 2: (A) Location of the Salmon River catchment (red polygon) in British Columbia, Canada. The watershed is 4230 km² and located around 700 km north of Vancouver. It is located in the Rocky Mountains with an elevation of 606 m above sea level at the streamflow gauge station of the Salmon River (08KC001). (B) The average monthly mean temperatures (red line) and average monthly precipitation is divided into rain (dark blue) and snow (light blue). Maps of (C) the four soil types based on the Harmonized World Soil Data (HWSD; 30") (Nachtergaele et al., 2010) and (D) four land cover types based on the MCD12Q1 MODIS/Terra+Aqua Land Cover (500m) (Friedl and Sulla-Menashe, 2015) of the Salmon River catchment are provided. The colors indicate different soil and land use classes.

[Figure]

Figure 5: Results of the Sobol' sensitivity analysis of the hydrologic modeling framework Raven. (A) The sensitivities of 35 model parameters (see Table C2) and 8 parameters $r_i$ that are used to determine the weights of process options are estimated. The Sobol' sensitivity index estimates are determined also for (B) 19 process options and (C) the 11 processes. The information which parameters are used in which process option and process can be found in Table C1. The different colors indicate the association of parameters and process options to the eleven processes. Parameters $x_{29}$ and $x_{30}$ are associated with several process options and are not colored but gray. The Sobol' main and total effects are shown (dark and light colored bars, respectively). All sensitivity index estimates shown are originally time-dependent and are aggregated as variance-weighted averages (Eq. 23 and 24). The average weights over the course of the year are shown in Figure 6.

Nachtergaele, F., van Velthuizen, H., Verelst, L., Batjes, N. H., Dijkshoorn, K., van Engelen, V. W. P., Fischer, G., Jones, A., Montanarella, L., Petri, M., Prieler, S., Shi, X., Teixera, E., and Wiberg, D.: The harmonized world soil database, in: 19th World Congress of Soil Science, Soil Solutions for a Changing World, Brisbane, Australia, 1-6 August 2010, pp. 34–37, 2010.

Saltelli, A., Ratto, M., Andres, T. H., Campolongo, F., Cariboni, J., Gatelli, D., Saisana, M., and Tarantola, S.: Global sensitivity analysis. The primer, John Wiley & Sons, Ltd., 2008.

Sobol', I. M.: Sensitivity analysis for non-linear mathematical models, Mathematical Modeling & Computational Experiment (Engl. Transl.), 1, 407–414, 1993.

Sobol, I. M. and Kucherenko, S. S.: Global Sensitivity Indices for Nonlinear Mathematical Models. Review, WILMOTT magazine, pp. 2–7, 2004.

Van Hoey, S., Seuntjens, P., van der Kwast, J., and Nopens, I.: A qualitative model structure sensitivity analysis method to support model selection, Journal of Hydrology, 519, 3426–3435, 2014.

---

## Author Comment (AC3) · 29 Sep 2020

**Reply to Short Comment of Gabriele Baroni**
*Review received and published: 23 Aug 2020*

Dear Gabriele,

Thanks a lot for your suggestion. We will reply below in detail to your comments. Your comments are *italic*; our replies are highlighted **bold**. The line numbers in red are referring to the revised draft.

Best regards,
Julie, James, and Bryan

*Dear Juliane, dear Authors,*
*I have really appreciated that your study has been motivated by, among others, one of my papers (Baroni and Tarantola, 2014). I also think your manuscript can be a nice contribution to the literature, but I leave to the official Reviewers to judge with more specific comments. When reading this preprint, however, I found the need to add this short comment to clarify the terminology. I hope this will also help to strengthen your work.*
*Sincerely,*
*Gabriele Baroni*

*Comments*

*In this study, you have introduced the use of weights that can take on non-integer values to account for model structures in the analysis. As you correctly cited I used, in contrast, discrete integer values in Baroni and Tarantola (2014). However, I find important to underline also here that this "trick" has been previously used. I think that I had properly acknowledged that in my paper and I paste the reference below for sake of clarity:*

*"The use, at step 5 of the framework, of a discrete scalar factor of the size of the realizations generated, enables us to extend the GSA also to non-scalar sources of uncertainty. This approach was introduced by Crosetto and Tarantola (2001), who proposed the use of a sensitivity analysis of a binary input to 'switch' the uncertainties of a rainfall intensity map on and off at the same rate (i.e. for N/2 runs, the switch is set to off and for the remaining N/2 runs it is set to on), allowing their relative importance to be determined. The same approach was then improved by Lilburne et al. (2003) and Lilburne and Tarantola (2009) who explicitly introduced the discrete uniform distribution associated to the different realizations of each specific source of uncertainty as considered in this framework."*

*Additional discussion on the use of discrete random variables can be found also in Plischke et al. (2013).*

*For this reason, I found misleading to read in your manuscript that you compare your xSSA method with "Baroni method". Instead, I suggest using something like "continuous weights method" vs. "discrete values method". In my opinion this would better describe what you are comparing.*

**We agree with the reviewer that this method was used before and the introduced as a novel framework for sensitivity analyses by Baroni and Tarantola (2014). To avoid confusion and acknowledge also previous attempts to use binary or discrete random variables for sensitivity analyses we now use the term "discrete values method" (DVM) throughout the manuscript instead of "Baroni method". We however continue to refer to the proposed method as the xSSA method as it comprises not only of the "continuous weights" but also the grouping of variables. We made the following adjustment in the introduction:**

**line 50 ff.** To date, there have been limited attempts to simultaneously estimate model parameter, input, and structural sensitivities. One notable attempt is introduced by Baroni and Tarantola (2014) using a Sobol' sensitivity analysis based on grouped parameter. In that study, groups of soil and crop parameters, the number of soil layers, and a group of parameters to perturb inputs are investigated. These groups of parameters are pre-sampled and a finite set of parameters for each of the four groups is chosen and each set is enumerated. The sensitivity analysis is then based on those enumerated sets. This means, rather than sampling each individual parameter like in a classic Sobol' analysis, an integer for each group acting as a hyper-parameter is sampled. The model is then run with the associated pre-sampled parameter set. While the approach may be generally applicable to arbitrary structural differences, in their testing, Baroni and Tarantola (2014) varied only in how the model was internally discretized (i.e., in the number of soil layers). The soil and crop parameters were always used for the same soil and crop process. The major limitation of this method is, however, that individual parameters need to be mutually exclusive and can only be associated to one type of uncertainty. The method hence limits the groups that can be defined, for instance, overlapping group definitions are not possible. The method will be referred to as "discrete values method (DVM)" in the following and will be contrasted to the method developed here to examine this limitation in more detail.

We also replaced "Baroni method" everywhere else in the manuscript by "discrete values method (DVM)".

**References**

Baroni, G. and Tarantola, S.: A General Probabilistic Framework for uncertainty and global sensitivity analysis of deterministic models: A hydrological case study, Environmental Modelling & Software, 51, 26–34, 2014.

Crosetto, M. and Tarantola, S.: Uncertainty and sensitivity analysis: tools for GIS-based model implementation, International Journal of Geographical Information Science, 15, 415–437, 2001.

Lilburne, L. and Tarantola, S.: Sensitivity analysis of spatial models, International Journal of Geographical Information Science, 23, 151–168, 2009.

Lilburne, L. R., Webb, T. H., and Francis, G. S.: Relative effect of climate, soil, and management on risk of nitrate leaching under wheat production in Canterbury, New Zealand, Australian Journal of Soil Research, 41, 699–709, 2003.

Plischke, E., Borgonovo, E., and Smith, C. L.: Global sensitivity measures from given data, European Journal of Operational Research, 226, 536–550, 2013.

---

## Author Response (AR1)

Dear Jim,

thanks a lot for your positive feedback. We are glad you liked our reply to the reviewers comments. We will reply below to your comments. Your comments are *italic*; our replies are highlighted **bold**. The **line numbers in red** are referring to the final draft (without tracked changes).

We want to note that some of the line numbers we indicated in our replies to the reviewers might not be correct anymore due to our adjustment in **line 467 ff.** requested below.

Thanks for handling our manuscript.

Best regards,
Julie, James, and Bryan

*Comments to the Author:*

*Dear Authors, this paper has generated 2 very good and detailed reviews that I believe can help improve the readability of the manuscript and it's clarity. You have done an excellent job of responding to these comments, which I should point out the reviewers were very enthusiastic about your manuscript and it's publication, and I have no problem with almost everything that you plan to do as final changes and to complete this process. I thank everyone involved for the time and diligence on evaluating this manuscript.*

**Thanks Jim. We are glad to hear that.**

*I will just make a couple of super quick recommendations:*

1. *Can you in the supp information on Raven just note that it is extracted from the paper you identify - I think this is great to have as a supp - I think it just needs clarifying where it has come from if copied at the start of that supp*

   **We added the following section at the beginning of the manuscript to make clear that these are supplements for the HESS manuscript.**

   **line 1 ff. (supplements)** **The following material is supplementary to the manuscript: Mai, J., Craig, J. R., and Tolson, B. A.: Simultaneously Determining Global Sensitivities of Model Parameters and Model Structure, Hydrol. Earth Syst. Sci. Discuss., *https://doi.org/10.5194/hess-2020-215*, in review, 2020.**

   **We also rephrased the note that most of the supplements can be found in the Raven manual [Craig, 2020]:**

   **line 23 ff. (supplements)** **The following description of all processes and process options is copied from the Raven documentation [Craig, 2020] and is provided here for the convenience of the reader.**

2. *Whilst you respond well to reviewer #2 point number 16 I note that there are no suggested changes within the manuscript, but I think it's an important point to clarify to confirm what your assumptions are on 'the same' parameter within your analyses.*

   **We probably did not highlight strongly enough in the reply to the reviewer that this is already mentioned multiple times in the draft.**

   **line 165 ff.** **The latter is assumed to be more realistic as model parameters such as, for example, the thickness of the upper soil layer can appear in multiple processes (e.g., evaporation, quickflow, infiltration, and percolation).**

**line 465 ff.** Shared parameters occur often in several process options of the same process but also across processes and hence need to be considered when analyzing process options and processes in flexible frameworks.

**line 481 ff.** When the groups are defined as process options, i.e. various conceptual implementations of the same hydrologic process, parameters will be used in multiple options. Some parameters such as, for example, soil thicknesses or porosity might even be required across processes (for example infiltration and percolation).

We added the following to the results and discussion to emphasize this:

**line 467 ff.** In this methodology, it is at the user's discretion which parameters are grouped and how they are grouped. A defining characteristic of the xSSA approach is that it can support any grouping of parameters, though interpretation of xSSA results benefits from meaningfully assigning group membership.

*I have simply noted publish subject to minor revisions to clarify all the changes proposed are completed as simply a check by me. There is no barrier to final publication on the strength of the authors responses provided and the positive enthusiasm the reviewers have for the manuscript.*

**That makes sense. Thanks a lot for acknowledging the potential of this manuscript.**

**References**

[revised manuscript text omitted]

$$A_2 \quad = \quad 1 \tag{4}$$

$$B_1 \quad = \quad 1 + bx_2^4 \tag{5}$$

$$B_2 \quad = \quad 1 + bx_3^2 \tag{6}$$

190    $$B_3 \quad = \quad x_4 + bx_5 \tag{7}$$

$$C_1 \quad = \quad a\sin^2(x_6) \tag{8}$$

$$C_2 \quad = \quad 1 + bx_7^4 \tag{9}$$

For the shared-parameter benchmark model, the process $D$ is set to have 2 options ($D_1$, $D_2$), $E$ has 3 options ($E_1$, $E_2$, $E_3$), and $F$ has two options ($F_1$, $F_2$):

195    $$D_1 \quad = \quad \sin(x_1) \tag{10}$$

$$D_2 \quad = \quad x_1 + x_2^2 \tag{11}$$

$$E_1 \quad = \quad 1 + bx_2^4 \tag{12}$$

$$E_2 \quad = \quad 1 + bx_3^2 \tag{13}$$

$$E_3 \quad = \quad x_4 + bx_5 \tag{14}$$

200    $$F_1 \quad = \quad a\sin^2(x_6) \tag{15}$$

[revised manuscript text omitted]